# Cryo-EM architecture of a near-native stretch-sensitive membrane microdomain

Jennifer M. Kefauver[1,7], Markku Hakala[2], Luoming Zou[1], Josephine Alba[3], Javier Espadas[2], Maria G. Tettamanti[1,2], Jelena Gajić[1,4], Caroline Gabus[1], Pablo Campomanes[3], Leandro F. Estrozi[5], Nesli E. Sen[1], Stefano Vanni[3,6], Aurélien Roux[2], Ambroise Desfosses[5] & Robbie Loewith[1✉]

Biological membranes are partitioned into functional zones termed membrane microdomains, which contain specific lipids and proteins[1–3]. The composition and organization of membrane microdomains remain controversial because few techniques are available that allow the visualization of lipids in situ without disrupting their native behaviour[3,4]. The yeast eisosome, composed of the BAR-domain proteins Pil1 and Lsp1 (hereafter, Pil1/Lsp1), scaffolds a membrane compartment that senses and responds to mechanical stress by flattening and releasing sequestered factors[5–9]. Here we isolated near-native eisosomes as helical tubules made up of a lattice of Pil1/Lsp1 bound to bio plasma membrane lipids, and solved their structures by helical reconstruction. Our structures reveal a striking organization of membrane lipids, and, using in vitro reconstitutions and molecular dynamics simulations, we confirmed the positioning of individual $PI(4,5)P_2$, phosphatidylserine and sterol molecules sequestered beneath the Pil1/Lsp1 coat. Three-dimensional variability analysis of the native-source eisosomes revealed a dynamic stretching of the Pil1/Lsp1 lattice that affects the sequestration of these lipids. Collectively, our results support a mechanism in which stretching of the Pil1/Lsp1 lattice liberates lipids that would otherwise be anchored by the Pil1/Lsp1 coat, and thus provide mechanistic insight into how eisosome BAR-domain proteins create a mechanosensitive membrane microdomain.

Membrane compartmentalization enables the spatio-temporal control of a variety of signalling events at the plasma membrane. Although the biological evidence for membrane compartmentalization is overwhelming[1,2,10], the determinants and the physical structure of the lipid organization within the membrane remain controversial. This is because almost all of the tools that are used to study membrane lipids also risk perturbing their behaviour within the membrane context[3,4].

In *Saccharomyces cerevisiae*, at least three plasma membrane compartments have been identified—the membrane compartment containing Pma1 (MCP), the membrane compartment containing Can1 (MCC) and the highly dynamic membrane compartment containing TORC2 (MCT)[5]—in addition to the patchwork organization of many other integral membrane proteins[11]. The MCC microdomains are randomly distributed membrane furrows about 300 nm long and 50 nm deep, scaffolded by a protein coat composed of the Bin–amphiphysin–Rvs (BAR) domain family protein Pil1 and its paralogue Lsp1, known as the eisosome[5,6,12]. MCC–eisosomes are relatively stable[13–15], and have been implicated in sensing and responding to plasma membrane stress: various stimuli, including hypo-osmotic shock, heat shock and mechanical pressure, cause eisosomes to flatten and release sequestered proteins to affect signalling or transport functions[5–7,12,16–18].

BAR-domain proteins are a large and diverse family of proteins that have physiological roles in membrane curvature sensing and/or induction[19–21]. Most of these proteins have a characteristic banana shape, with a membrane-binding surface that shows dense positive charge, enabling interaction with negatively charged lipid head-groups[19–22]. Although the features of BAR-domain proteins that mediate their function of sensing and/or generating curvature and their lipid-binding affinities in vitro are well-characterized, what remains unknown is how BAR scaffolding affects the organization of the complex mixture of lipids naturally found in cell membranes in vivo.

Cryo-electron microscopy (cryo-EM) is an emerging tool for the label-free study of membranes and protein–lipid interactions[23–25]. Beyond the wealth of data coming from structures of transmembrane proteins with bound lipids[24], studies have highlighted the potential of cryo-EM for studying lipids within the membrane. Variations in membrane thickness mediated by lipid composition and/or lipid–protein interactions have been observed by cryo-EM in liposomes in vitro[26,27], in reconstituted protein–lipid assemblies[28] and in in situ systems[29]. Moreover, patterned perturbations in membrane density mediated by lipid–protein interactions provide compelling examples of how this technique can be used to study lipids in molecular detail within their context[30,31].

[1]Department of Molecular and Cellular Biology, University of Geneva, Geneva, Switzerland. [2]Department of Biochemistry, University of Geneva, Geneva, Switzerland. [3]Department of Biology, University of Fribourg, Fribourg, Switzerland. [4]Department of Organic Chemistry, University of Geneva, Geneva, Switzerland. [5]Institut de Biologie Structurale, Université Grenoble Alpes, CEA, CNRS, IBS, Grenoble, France. [6]Swiss National Center for Competence in Research (NCCR) Bio-inspired Materials, University of Fribourg, Fribourg, Switzerland. [7]Present address: Nanomaterials and Nanotechnology Research Center (CINN), Spanish National Research Council (CSIC), El Entrego, Spain. ✉e-mail: Robbie.Loewith@unige.ch

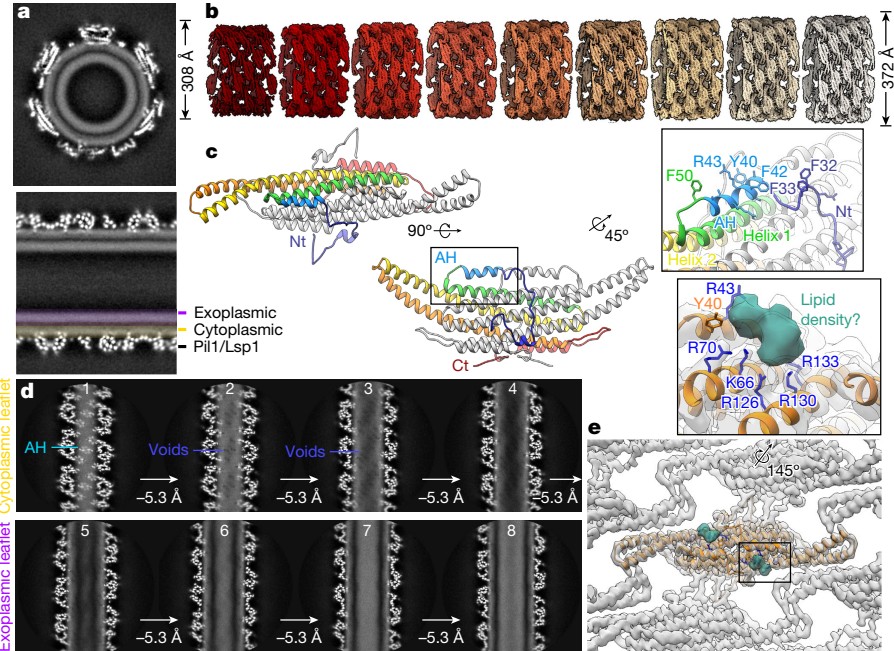

**Fig. 1 | Native-source eisosomes retain an unperturbed plasma membrane microdomain. a**, Central (top, transverse; bottom, sagittal) slices of a helical reconstruction of a native-source MCC–eisosome tubule with the membrane bilayer visible. **b**, Sharpened maps of nine helical structures of varying diameter. **c**, Model of the Pil1 dimer. Rainbow colouring on chain A from Nt (blue) to C terminus (Ct; red). Inset, magnified view of the AH. **d**, Series of one-pixel slices through the helical reconstruction of the native-source eisosome, separated by a depth of approximately 5.3 Å. Cyan label indicates AH (panel 1), violet labels indicate membrane voids (panels 2–3). The void pattern continues through the cytoplasmic leaflet (panel 4) but is absent in the exoplasmic leaflet (panels 5–8). **e**, Unassigned putative lipid density (sea green) in a deepEMhancer sharpened map localized to the charged pocket. Inset, charged residues coordinating the unassigned density.

## Native-source eisosome filaments

We isolated MCC–eisosomes from *S. cerevisiae* using a gentle purification procedure, preserving a lattice of untagged Pil1/Lsp1 structural proteins bound to a presumably intact plasma membrane bilayer in a near-native state, observable as a two-layer density in the protein tubule (Fig. 1a). Although MCC–eisosomes have been shown to have a furrow-like half-pipe structure in vivo[32,33], our isolated eisosome tubules seem to be closed, continuous helices of Pil1/Lsp1 proteins (Fig. 1a,b). Using helical reconstruction, we were able to resolve nine independent eisosome tubule structures with diameters ranging from 308 Å to 372 Å (within the range of diameters of curvatures observed in MCC–eisosomes in situ[34] (around 300–600 Å)), but with a nearly identical lattice pattern of Pil1/Lsp1 dimers in all structures (Fig. 1b, Extended Data Fig. 1a,b, Extended Data Table 1 and Supplementary Table 1).

Using a symmetry expansion and density subtraction strategy, we could refine our structures to a resolution of around 3.2 Å, which allowed us to build models of the paralogues Pil1 and Lsp1 (Fig. 1c and Extended Data Figs. 1c, 2a–c, 3a and 4a). We have chosen to base our interpretations on the model of a Pil1 homodimer owing to its essential role[32,35] and its predominance over Lsp1 in our sample (Extended Data Fig. 1d and Supplementary Data 1). With our improved resolution, we could characterize the lattice contact sites between Pil1 dimers in detail (Extended Data Fig. 4b–e).

## AHs associated with a pattern of membrane voids

Two novel structured regions of the protein were visible in our maps: (1) a folded N terminus (Nt) that forms lattice contact sites with the Nt of a neighbouring dimer, followed by (2) an Nt amphipathic helix (AH; residues 39–48) buried within the lipid density of the cytoplasmic leaflet of the bilayer that runs parallel to the BAR-domain helices of each Pil1/Lsp1 monomer (Fig. 1c, inset and Extended Data Fig. 4c). N-terminal AHs are a common feature of the N-BAR family of BAR-domain proteins (for example, endophilin and amphiphysin) and have been proposed to have crucial roles in the sensing and induction of curvature[21,36,37].

By making one-pixel slices through the unsharpened maps parallel to the axis of the eisosome tubule, we could clearly see the well-defined protein density of the AH buried in the cytoplasmic leaflet of the membrane, which is visible in these images as a uniform density of lower intensity (Fig. 1d, panel 1, teal arrow). At slices around 5 Å deeper, an array of small voids in the membrane density begins to appear just below the AH. This pattern continues throughout the cytoplasmic leaflet, producing a fence-like striation in the membrane (Fig. 1d, panels 2–4, violet arrows). This effect is asymmetric between the two leaflets: the pattern of voids is present only in the protein-bound cytoplasmic leaflet, whereas the density of the exoplasmic leaflet is homogeneous throughout the lateral slices (Fig. 1d, panels 5–8).

Visualizing the native-source membrane density at a high threshold to produce an inverted view of its topological features revealed small droplet-shaped pockets of higher resolution than the surrounding membrane density that are intercalated between the bulky side chains of the Pil1 AH (Extended Data Fig. 5a and Supplementary Video 1). We wondered whether these membrane voids could represent stably localized ergosterol molecules (see Supplementary Discussion).

## Lipid-binding pocket on the Pil1/Lsp1 dimer

Visualization of the electrostatic surface on the membrane-facing surface of the Pil1 dimer revealed two patches of intense positive charge adjacent to the AHs (Extended Data Fig. 1e). Within this pocket, an unassigned density is coordinated by several charged residues that have previously been proposed to be involved in PI(4,5)P$_2$ binding[33,38] (Fig. 1e). In vitro tubulation by Pil1 and Lsp1 is reportedly dependent on the presence of PI(4,5)P$_2$ in liposomes, and defects in PI(4,5)P$_2$ regulation cause changes in eisosome morphology in vivo[33,39,40]. In addition,

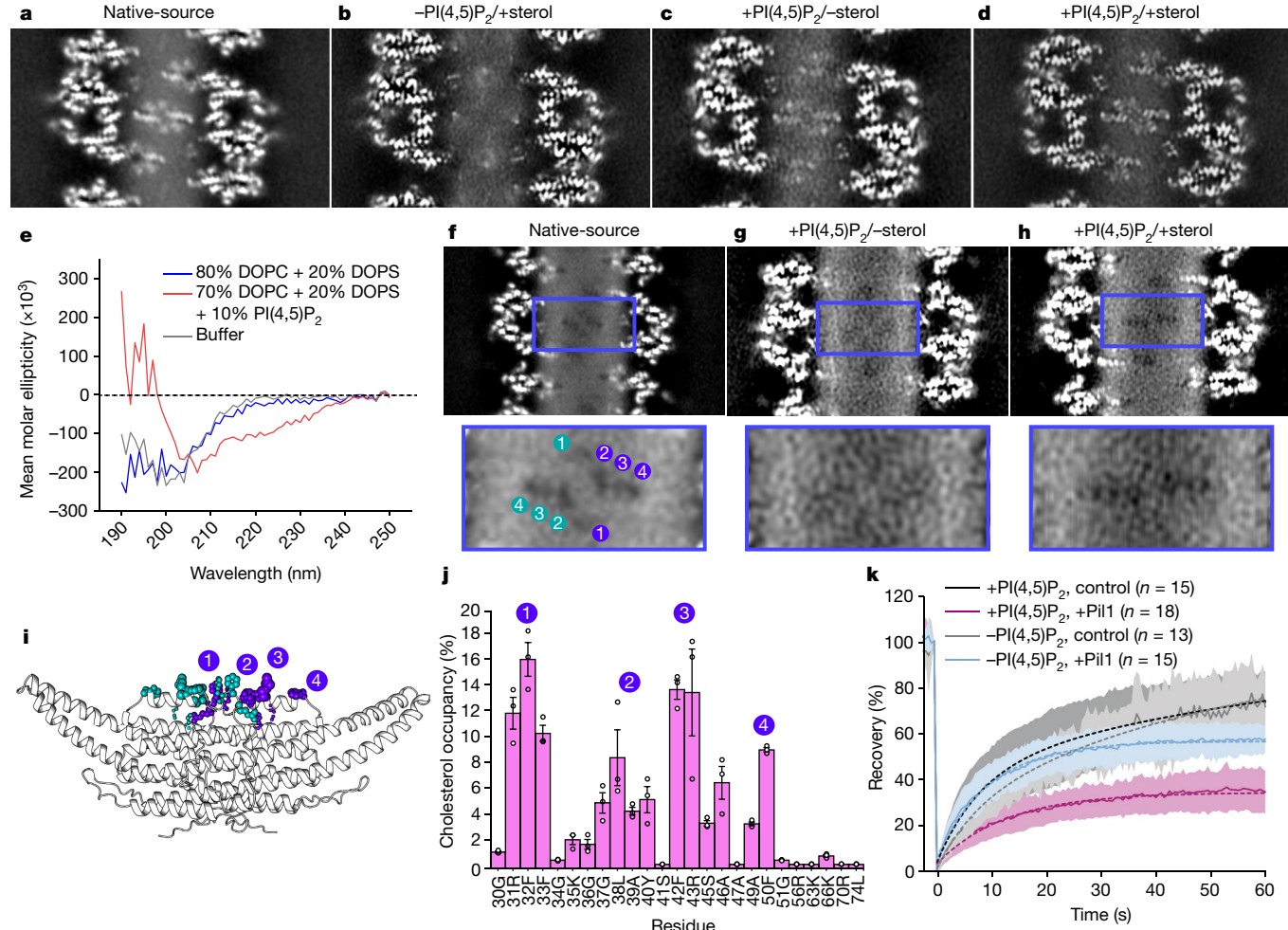

**Fig. 2 | Sterols are stabilized by the Pil1/Lsp1 AH within the MCC–eisosome membrane microdomain. a–d**, Parallel slice at maximum AH density of unsharpened maps of native-source (**a**), −PI(4,5)P$_2$/+sterol reconstituted (**b**), +PI(4,5)P$_2$/−sterol reconstituted (**c**) and +PI(4,5)P$_2$/+sterol reconstituted (**d**) eisosomes. **e**, CD spectra reflecting the folding of a synthetic Pil1 AH peptide in the presence of liposomes of the indicated compositions. Lines represent the mean of three experiments. **f–h**, Membrane void pattern within the cytoplasmic leaflet in native-source (**f**), +PI(4,5)P$_2$/−sterol reconstituted (**g**) and +PI(4,5)P$_2$/+sterol reconstituted eisosomes (**h**). Inset images: magnified view of membrane void region. Numbers indicate individual sterol dwell sites for native-source eisosomes (see **i**). **i**, CG MD snapshot highlighting residues with cholesterol headgroup occupancy. **j**, Average per-residue percentage occupancy of cholesterol at AH residues in the +PI(4,5)P$_2$/+sterol system in CG MD simulations (three replicas of 10 μs each), with peaks numbered at sterol dwell sites (see **i**). Error bars are s.e.m. **k**, FRAP of TF-cholesterol in samples with or without 1% PI(4,5)P$_2$. Solid lines indicate a mean of $n$ measured nanotubes with s.d. shown. Dashed lines indicate the fitted data.

Lsp1 has been shown to cluster PI(4,5)P$_2$ on giant unilamellar vesicles and prevent its lateral diffusion within the membrane[41]. Because of this well-documented structural and functional relationship between the eisosomes and PI(4,5)P$_2$, we speculated that this unassigned density could represent one or more coordinated PI(4,5)P$_2$ headgroups.

## Reconstitution of eisosomes with known lipids

To assign identities to the structural signatures we observe within the membrane, we chose to reconstitute eisosome filaments using lipid mixtures of a known composition with recombinantly expressed Pil1 protein. We tested several lipid mixtures (Extended Data Fig. 6a–c), ultimately using combinations of cholesterol and/or PI(4,5)P$_2$ with a mixture of phosphatidylcholine (DOPC), phosphatidylethanolamine (DOPE) and phosphatidylserine (DOPS) as a constant component. Our final reconstructions were made with three lipid mixtures: (1) 'minus PI(4,5)P$_2$/plus sterol' ('−PI(4,5)P$_2$/+sterol'), containing 30% cholesterol; (2) 'plus PI(4,5)P$_2$/minus sterol' ('+PI(4,5)P$_2$/−sterol'), containing 10% PI(4,5)P$_2$; and (3) 'plus PI(4,5)P$_2$/plus sterol' ('+PI(4,5)P$_2$/+sterol'), containing 10% PI(4,5)P$_2$ and 15% cholesterol (Supplementary Tables 2

and 3). We used helical reconstruction to solve several structures of varying diameters, helical parameters and resolutions (Extended Data Figs. 3b–d and 6d–g, Extended Data Table 1 and Supplementary Table 1).

For all three lipid mixtures, the overall architecture of the Pil1 lattice is similar to the native-source samples. However, in −PI(4,5)P$_2$/+sterol preparations, the AHs of the Pil1 dimers are not resolved (Fig. 2a,b and Extended Data Fig. 2d). This could be because the AH is more mobile in this lipid mixture or because it is not inserted into the membrane. The AH (H$_0$) of endophilin, for example, requires PI(4,5)P$_2$ for membrane penetration[42]. In the +PI(4,5)P$_2$/−sterol samples, the AH density is improved, and the AH density resolution is best and most resembles that of the native-source eisosomes in the +PI(4,5)P$_2$/+sterol samples, which suggests that PI(4,5)P$_2$ is crucial for AH stabilization (Fig. 2c,d and Extended Data Fig. 2e,f). To probe the PI(4,5)P$_2$-dependency of the AH insertion into the membrane, circular dichroism (CD) experiments with a synthesized Pil1 AH peptide and small unilamellar vesicles were performed. Only in the presence of PI(4,5)P$_2$ do we observe a clear helical folding of the peptide (Fig. 2e). Together, these observations suggest that the AH does not insert into the bilayer in the absence of PI(4,5)P$_2$.

## Membrane voids require PI(4,5)P$_2$ and sterol

One feature that was notably absent from the +PI(4,5)P$_2$/−sterol helices was the pattern of membrane voids in the cytoplasmic leaflet that we observe in the native-source eisosome filaments (Fig. 2f,g and Extended Data Fig. 5b,c). However, in the +PI(4,5)P$_2$/+sterol helices, we can again observe the void pattern, supporting the notion that this pattern results from the stable association of sterol molecules with the AH (Fig. 2h and Extended Data Fig. 5d,e). Coarse-grained (CG) molecular dynamics (MD) simulations were performed with tubules of an identical lipid composition to that of the +PI(4,5)P$_2$/+sterol mixture, and the number of contacts between lipid headgroups and residues of Pil1 was measured for each lipid in terms of the percentage of occupancy; that is, the percentage of frames in which any lipid–protein contact is formed (Extended Data Fig. 7a). Notably, sites of increased cholesterol occupancy corresponded to the locations of the pattern of holes in the lipid density, with peaks at four clusters of residues, each containing aromatic side chains: (1) residues 32F and 33F, (2) residues 37G, 38L, 39A and 40Y; (3) residues 42F and 43R; and (4) residue 50F (Fig. 2i,j and Extended Data Fig. 7b).

To directly test whether the voids represent sterol molecules, we reconstituted and solved structures of Pil1 tubules with lipid mixtures containing sterols that were brominated at the 7,8- position of the steroid ring (bromosterol) to add density to these molecules that can be observed in cryo-EM[30] (Extended Data Figs. 3e and 6h and Supplementary Data 2 and 3). In these +PI(4,5)P$_2$/+bromosterol reconstituted structures, the AHs are well-resolved, similar to the +PI(4,5)P$_2$/+sterol structures, suggesting that the bromosterols behave similarly to cholesterol in these structures (Extended Data Fig. 5e,f). Furthermore, the voids can be observed both at the plane of the AH and starting at a depth of around 8 Å from the bilayer midplane, corresponding approximately to the predicted location of C17 of cholesterol within the bilayer in MD simulations[43]. However, in the +PI(4,5)P$_2$/+bromosterol structures, in the slices ranging from a distance of around 11–12.5 Å from the bilayer midplane, which would correspond well with the bromination at C7(8) on the bromosterol molecule, the voids are interrupted by density, which strongly suggests that the brominated sterols are localized to the voids, and that the void pattern represents stabilized sterol molecules (Extended Data Fig. 5e,f).

To better understand the relationship between PI(4,5)P$_2$ binding and sterol dynamics, we reconstituted Pil1 scaffolds on preformed membrane lipid nanotubes to perform fluorescence recovery after photobleaching (FRAP) assays using TopFluor (TF)-cholesterol in lipid mixtures with or without 1% PI(4,5)P$_2$ (Supplementary Videos 2 and 3). For Pil1-scaffolded nanotubes without PI(4,5)P$_2$ (−PI(4,5)P$_2$/+sterol), there is a slight reduction in the mobile fraction of sterols, but this effect is pronounced in the presence of 1% PI(4,5)P$_2$ (+1% PI(4,5)P$_2$/+sterol), indicating that the immobilized fraction of cholesterol interacts strongly with Pil1 and/or other lipids bound to the protein in a PI(4,5)P$_2$-dependent manner (Fig. 2k and Supplementary Table 4).

## Lipid binding in reconstituted eisosomes

In the +PI(4,5)P$_2$/−sterol structures, a clear triangular density, which we fitted with an inositol-1,4,5-phosphate (IP$_3$) ligand to represent a PI(4,5)P$_2$ headgroup, interacts with the basic residues R126, K130 and R133 on each protomer in the binding pocket we had previously identified in the native-source structures (Fig. 3a, second panel and Extended Data Fig. 2e). Notably, these conserved residues were previously shown to be important for eisosome assembly in vivo and membrane binding in vitro[33,38]. This extra density, not present in the −PI(4,5)P$_2$/+sterol Pil1 filaments (Extended Data Fig. 2d), is likely to reflect a PI(4,5)P$_2$-binding site in the native-source eisosomes. To confirm the

interaction between PI(4,5)P$_2$ and Pil1, we checked TF-PI(4,5)P$_2$ diffusion in lipid nanotubes using FRAP assays, and observed a much slower recovery and a significantly increased immobile fraction of TF-PI(4,5)P$_2$ when Pil1 is bound, suggesting a strong interaction (Fig. 3b and Supplementary Table 4). We also measured lipid sorting coefficients of 1% TF-PI(4,5)P$_2$ using a fluorescence ratiometric comparison with a reference lipid (Atto647N DOPE). This revealed a relative accumulation of PI(4,5)P$_2$ in regions with Pil1 scaffolds (Fig. 3c and Extended Data Fig. 8a–c).

In the +PI(4,5)P$_2$/+sterol samples, a similar triangular density bound to R126, K130 and R133 was observed and fitted with an IP$_3$ ligand, comparable with the PI(4,5)P$_2$ headgroup in +PI(4,5)P$_2$/−sterol samples (Fig. 3a, third panel and Extended Data Fig. 2f). Lipid sorting values for PI(4,5)P$_2$ in the +1% PI(4,5)P$_2$/+sterol samples revealed the accumulation of PI(4,5)P$_2$ under Pil1 scaffolds, albeit to a lesser extent compared with +1% PI(4,5)P$_2$/−sterol samples, which is in line with the observed slight decrease in the intensity of this PI(4,5)P$_2$ density in the AH region of the +PI(4,5)P$_2$/+sterol samples (Fig. 3c and Extended Data Fig. 8d). In addition, in the CG MD simulations for both the +PI(4,5)P$_2$/−sterol and the +PI(4,5)P2/+sterol systems, PI(4,5)P$_2$ occupancy was increased at charged residues in the lipid-binding pocket, especially residues R43, Q60, R126, K130 and R133. The inclusion of cholesterol reduced occupancy for all these residues, complementing the lipid sorting observations (Fig. 3d and Extended Data Fig. 7c). In FRAP experiments, we observed a further reduction in PI(4,5)P$_2$ mobility under the Pil1 scaffold in samples with cholesterol, which suggests that, for the immobile fraction of PI(4,5)P$_2$, sterols might have a role in enhancing the Pil1–PI(4,5)P$_2$ interaction, perhaps by stabilizing the AH (Fig. 3b and Supplementary Table 4).

In the +PI(4,5)P$_2$/+sterol samples, we were surprised to observe an additional lipid density stabilized between the AH and the PI(4,5)P$_2$ headgroup. This density accommodates a phosphatidylserine (PS) lipid (DOPS is present in our lipid mixtures), with a large splay in the acyl tails. One acyl tail is visible up to C2 and the other is stabilized up to C10, including the double bond at the 9,10- position, and bent, coordinated by residues R43, K66 and R70 (Fig. 3a, third panel and Extended Data Fig. 2f). Consistent with reported in vitro and in vivo partitioning of PS with sterols[44–46], we observed TF-PS sorting, but not TF-PC or TF-PE (in which PC denotes phosphatidylcholine and PE denotes phosphatidylethanolamine) sorting, using lipid nanotubes with the +1% PI(4,5)P2/+sterol mixture, supporting the notion that our extra density is a PS lipid (Fig. 3c). CG MD simulations also suggest that DOPS exhibits a similar affinity to that of PI(4,5)P$_2$ to the charged lipid-binding pocket of Pil1, but with changes at several residues that are predicted to be involved in lipid binding in the presence of sterols (Extended Data Fig. 7c–e). Collectively, these data suggest that the inclusion of sterol in the lipid mixture increases the specificity of PI(4,5)P$_2$ and PS interactions with particular charged residues within the lipid-binding pocket.

To investigate microdomain formation by the Pil1 lattice, we used FRAP assays with TF-PC and TF-PE and found that a significant portion of each of these lipids is immobilized in the presence of Pil1 (Extended Data Fig. 8e,f). In the presence of both Pil1 and PI(4,5)P$_2$, although the immobile fraction remains similar, the dynamics of the mobile lipid fraction are decreased for TF-PC and TF-PE lipids (Extended Data Fig. 8e,f and Supplementary Table 4). However, the immobile fraction and the dynamics of the mobile lipid fraction for TF-PC and TF-PE are broadly similar in the presence or absence of cholesterol (Extended Data Fig. 8g,h), which suggests that it is the protein lattice itself along with the binding of PI(4,5)P$_2$ (and the resulting AH insertion and stabilization) that slows lipid dynamics in the membrane microdomain. CG simulations, despite their intrinsic faster diffusion owing to the smoother CG free energy landscape[47], also indicate that the presence of the Pil1 coat slows down lipid diffusion in the outer leaflet of the membrane tubule (Extended Data Fig. 7f).

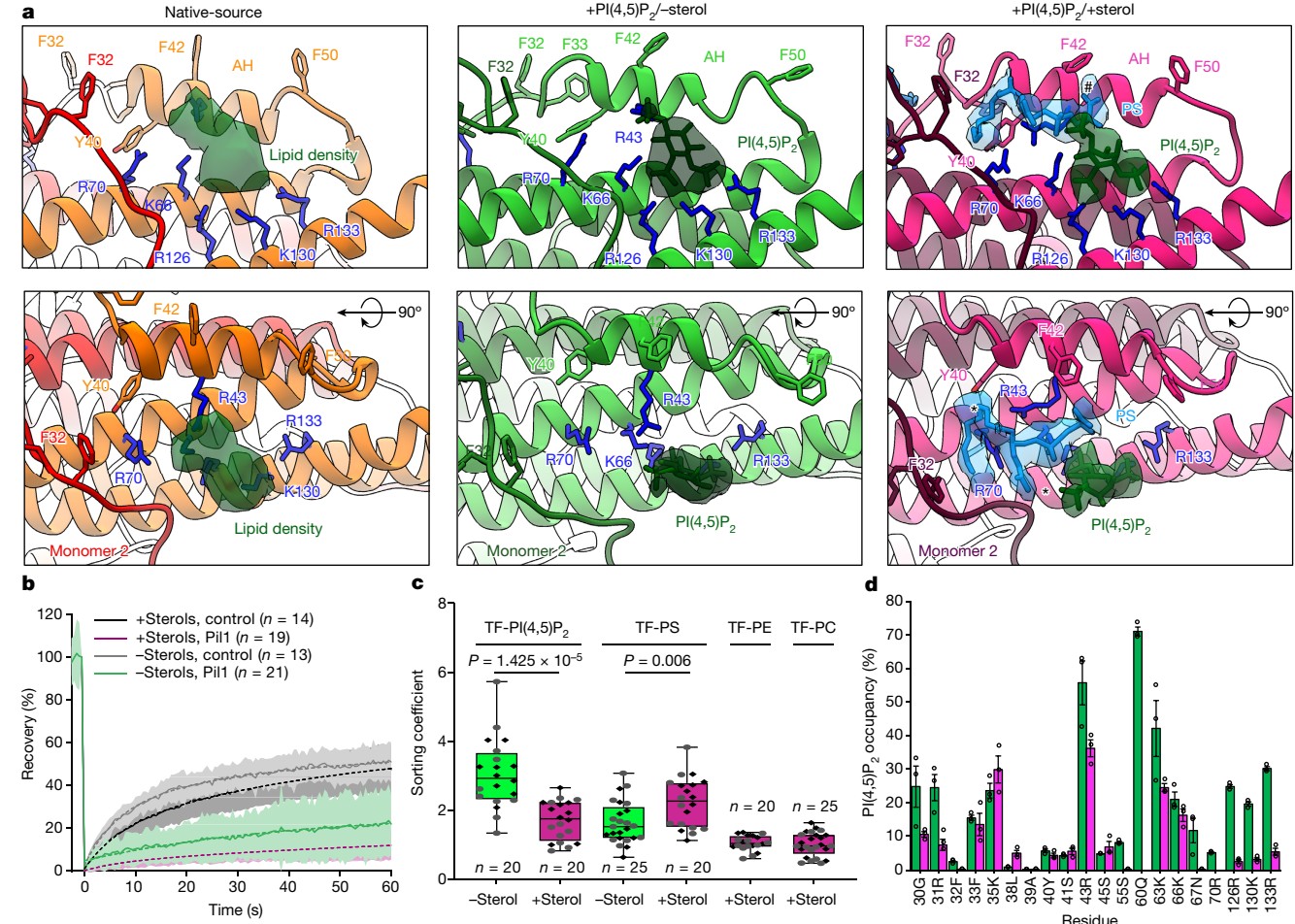

**Fig. 3 | Reconstitution of purified Pil1 with lipids of known composition enables the identification of structural signatures. a**, Lipid densities in lipid-binding pockets of native-source (orange; lipid density in dark green), +PI(4,5)P$_2$/−sterol reconstituted (lime green; PI(4,5)P$_2$ in dark green) and +PI(4,5)P$_2$/+sterol reconstituted (magenta; PI(4,5)P$_2$ in dark green and PS in dodger blue) eisosome maps. # indicates PS headgroup and * indicates two PS acyl tail chains in the +PI(4,5)P$_2$/+sterol reconstituted (magenta) model. **b**, FRAP of TF-PI(4,5)P$_2$ in +1% PI(4,5)P$_2$/−sterol (green) and +1% PI(4,5)P$_2$/+sterol (magenta) reconstituted Pil1 tubules. Solid lines indicate a mean of $n$ measured nanotubes with s.d. shown. Dashed lines indicate fitted data. **c**, Lipid sorting coefficients of TF-PI(4,5)P$_2$ and TF-PS in +1% PI(4,5)P$_2$/−sterol (green) and +1% PI(4,5)P$_2$/+sterol (magenta) reconstituted Pil1 tubules, as well as TF-PE and TF-PC in +1% PI(4,5)P$_2$/+sterol (magenta) reconstituted Pil1 tubules. Box plot elements are defined in the Methods. **d**, Average per-residue PI(4,5)P$_2$ lipid occupancy for residues less than 5 Å from the PI(4,5)P$_2$ headgroup with greater than 5% occupancy in CG MD simulations (three replicas of 10 µs each). Error bars are s.e.m.

## Physiological effects of lipid binding

To understand the in vivo relevance of the lipid binding we observe, we produced mutant yeast strains that were predicted to affect the binding of Pil1 to different lipid species on the basis of our structures and MD simulations (Fig. 4a and Supplementary Table 5). Specifically, we tagged endogenous *PIL1* with GFPenvy in an *LSP1* deletion background and then introduced mutations that are predicted to disrupt PI(4,5)P$_2$ binding (*pil1*[K130A/R133A]), PS binding (*pil1*[K66A/R70A]) or both PI(4,5)P$_2$ and PS binding (*pil1*[K66A/R70A/K130A/R133A]), as well as mutations to disturb sterol binding (*pil1*[F33A/Y40A/F42A/F50A]). We validated that lipid binding is indeed altered in these mutants using lipid sorting assays (for the mutants disrupting PS and PI(4,5)P$_2$ binding) and FRAP assays (for the mutant disrupting sterol binding) (Extended Data Fig. 9a,b).

Eisosome morphology and function was altered in these, as well as in two other *pil1* lipid-binding-pocket mutants (*pil1*[R43A] and *pil1*[R126A]) (Fig. 4b and Extended Data Fig. 9c). Strains expressing *pil1*[K66A/R70A], *pil1*[K130A/R133A] and, to a lesser degree, *pil1*[R43A], exhibited fewer eisosomes, but with an elongated morphology, whereas the *pil1*[R126A] and *pil1*[K66A/R70A/K130A/R133A] mutants displayed misshapen eisosomes and partial cytosolic mislocalization of the Pil1 protein. The *pil1*[F33A/Y40A/F42A/F50A]

sterol-binding-impaired mutant shows unusual rod-like eisosomes that ingress towards the centre of the cell (Fig. 4b and Extended Data Fig. 9d). To further investigate the role of these lipid-binding-pocket sites, we chose to add an endogenous mScarlet-I tag to the MCC–eisosome resident protein Nce102. Nce102 relocalizes from the MCC–eisosomes into the bulk membrane (or vice versa) after exposure to various membrane stressors, including osmotic shocks, inhibition of sphingolipid synthesis by the drug myriocin and sphingoid base treatment[7,48,49]. All of the lipid-binding-pocket mutants tested already showed a mislocalization of Nce102 in steady-state conditions, with almost no colocalization in the *pil1*[K66A/R70A/K130A/R133A] (PI(4,5)P$_2$- and PS-binding-impaired) and *pil1*[F33A/Y40A/F42A/F50A] (sterol-binding-impaired) mutants (Fig. 4c and Extended Data Fig. 9e).

To determine whether the disruption of lipid binding has any physiological consequences, we assessed the growth of our mutant strains under a variety of stress conditions. All of our Pil1 lipid-binding-pocket mutants exhibited resistance to myriocin, an inhibitor of sphingolipid biosynthesis (Fig. 4d and Extended Data Fig. 9f). This is in line with the previously described role of eisosomes in sphingolipid biosynthesis signalling[48,50] and, given the mislocalization of the proposed sphingolipid biosensor, Nce102 (ref. 49), we can speculate that sphingolipid

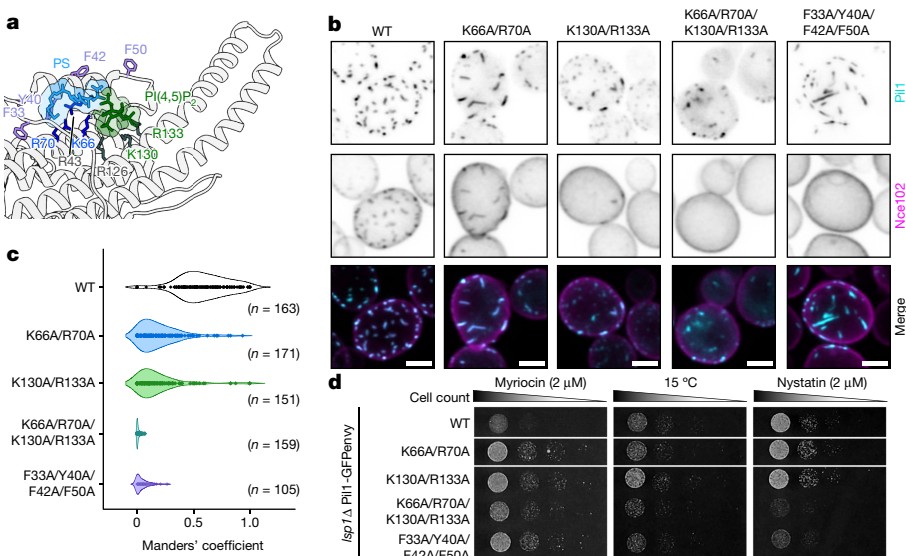

**Fig. 4 | Mutations that impair lipid binding affect the morphology and function of MCC–eisosomes in vivo. a**, Cartoon of the Pil1 lipid-binding pocket. Proposed sterol-binding residues are in violet, PS-binding residues in dodger blue and PI(4,5)P$_2$-binding residues in green. **b**, Eisosome morphology in *lsp1Δ* yeast expressing Pil1-GFPenvy with lipid-binding mutations and Nce102–mScarlet-I (summed *z*-stacks). The merge is summed stacks of Pil1-GFPenvy (cyan) and Nce102-mScarlet-I (magenta) signals. WT, wild type. Scale bars, 2 μm. **c**, Fraction of Nce102–mScarlet-I colocalizing with Pil1-GFPenvy lipid-binding-impaired mutants in single cells (Manders' M1 coefficient). The shaded area represents the probability density of the data. **d**, Growth assays of *lsp1Δ* yeast expressing Pil1-GFPenvy lipid-binding-pocket mutants. The tagged proteins in **b**–**d** are expressed from their endogenous locus.

biosynthesis is mildly upregulated in these myriocin-resistant mutants. We also observed that the *pil1*^K130A/R133A mutant had increased growth at a low temperature, relative to the control (Fig. 4d). Because cold resistance has been previously described for cells that lack the MCC–eisosome-localized PI(4,5)P$_2$ phosphatase Inp51, or Inp52 (ref. 51), it is possible that this cold resistance could be caused either by PI(4,5)P$_2$ dysregulation or by Inp51 or Inp52 mislocalization.

We also checked growth with nystatin, an antimycotic that interacts with free ergosterol in the plasma membrane to induce cell lysis[52,53]. The sterol-binding-impaired *pil1*^F33A/Y40A/F42A/F50A mutant and the *pil1*^R126A and *pil1*^K66A/R70A/K130A/R133A mutants that exhibited cytosolic mislocalization of the Pil1 protein were sensitive to nystatin, indicating a higher availability of free ergosterol at the plasma membrane (Fig. 4d and Extended Data Fig. 9f). However, none of the mutants showed resistance to the inhibitor of sterol synthesis atorvastatin[54], which suggests that the nystatin sensitivity is not due to enhanced ergosterol synthesis (Extended Data Fig. 9f). It is tempting to speculate that these mutations affect the lipid microdomain generated by Pil1, rendering a normally sequestered population of ergosterol available for nystatin binding, although it is also possible that a more general disorganization of ergosterol localization occurs owing to MCC–eisosome-dependent signalling dysregulation.

## 3DVA of the native-source eisosome lattice

With a clearer understanding of the identities of the lipids that produce the signatures we see within the membrane and their functional importance, we returned to the native-source eisosome dataset to check for variability in the lipid organization. We performed three-dimensional variability analysis (3DVA) using the symmetry-expanded and density-subtracted lattice particles. One component of our 3DVA was particularly noteworthy; we observed conformational flexibility within the Pil1 protein lattice that resembled a dynamic spring-like stretching and compression localized to the Nt contact sites (Supplementary Video 4). Because this Nt region is directly connected to the AH, we considered whether this lattice stretching could be transmitted to the lipid bilayer. We also saw that changes in the shape and size of the bound lipid density were synchronized with the stretching of the Nt contact sites in the visualization of the 3D variability.

To characterize the relationship between the Nt stretching and the changes in the lipid density, we extracted ten sets of non-overlapping particles along this variability dimension for further refinement, which allowed us to compare the most compact and most stretched lattice classes (Extended Data Fig. 3f,g and 10a,b). We also analysed the tubule of origin for each particle in these ten classes to determine the contribution of tubule diameter to this stretching component. Particles from every tubule diameter are distributed across all ten classes; however, particles from small-diameter tubules are overrepresented in the more compact classes, whereas those from large-diameter tubules are overrepresented in more stretched classes (Extended Data Fig. 10c,d). This suggests that the flexibility we observe arises from the protein lattice stretching to accommodate larger diameters of tubules.

In deepEMhancer sharpened maps, the lipid-binding pocket of the most compressed protein lattice class contained two clear densities: one triangular density that we could identify as a PI(4,5)P$_2$ headgroup with interactions at residues R126, K130 and R133; and the other smaller elongated density interacting with residue K66, which we had previously assigned as a PS lipid in the +PI(4,5)P$_2$/+sterol reconstituted samples, although its identity in the native plasma membrane cannot be definitively assigned owing to the complexity of its lipid composition relative to our reconstituted tubules (Fig. 5a and Extended Data Fig. 2g). By contrast, the lipid-binding pocket in the most stretched protein lattice class was unoccupied in the sharpened maps, suggesting that PI(4,5)P$_2$ and PS binding are disrupted upon lattice stretching (Fig. 5b and Extended Data Fig. 2h).

To further investigate the differences between these classes, we analysed slices through the membrane of the unsharpened maps (Fig. 5c,d and Extended Data Fig. 10e). Although the most compressed protein lattice class retained the pattern of membrane voids beneath the AH, indicating sterol binding, this pattern was gradually lost in the intermediate classes and was blurred in the most stretched protein lattice class, despite its slightly higher resolution (Fig. 5c,d, Extended Data Figs. 5g,h and 10f, Supplementary Videos 5 and 6 and Extended Data Table 1). However, no overall changes in membrane thickness or intensity were

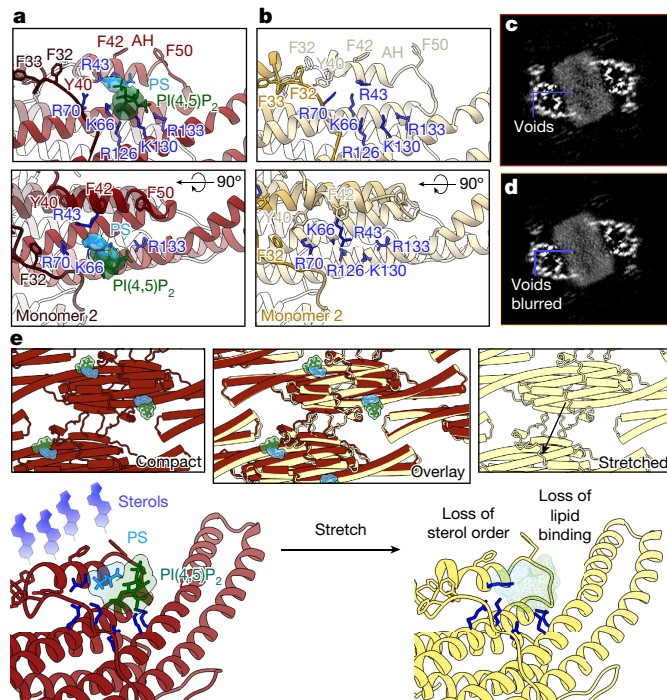

**Fig. 5 | Three-dimensional variability analysis reveals Pil1/Lsp1 lattice stretching and its effects on membrane organization within native-source MCC–eisosomes. a**, Well-defined two-part density (green for PI(4,5)P$_2$ and dodger blue for putative PS) occupying the charged pocket in a deepEMhancer sharpened map of the most compact protein lattice conformation (model in dark red). **b**, No clear density is observed in the charged pocket in a sharpened map of the most stretched protein lattice conformation (model in light yellow). **c**,**d**, Membrane void pattern, or lack thereof, in corresponding slices of unsharpened maps of the most compact protein lattice (**c**) and the most stretched protein lattice (**d**) conformation. **e**, Model illustrating that lattice stretching destabilizes lipid headgroup and sterol binding (compact model in dark red, stretched model in light yellow, PI(4,5)P$_2$ headgroup in green, PS headgroup in dodger blue and sterols in violet).

observed in radial angle profile plots of these two classes (Extended Data Fig. 10g). This implies that sterols are mobilized rather than redistributed between the leaflets in the stretched protein lattice class.

## Discussion

Together, our observations provide clear evidence that the eisosome scaffold proteins Pil1 and Lsp1 form a plasma membrane microdomain through their direct interactions with specific lipids. Our work provides a high level of detail into the organization and dynamics of plasma membrane lipids within this microdomain. Furthermore, our data allow us to propose a speculative model for how these MCC–eisosome microdomains sense and respond to mechanical stress.

Although the Pil1/Lsp1 lattice is capable of self-organization on the membrane surface in the absence of PI(4,5)P$_2$, we find that PI(4,5)P$_2$ is necessary for the stabilization of the AH within the cytosolic leaflet. When PI(4,5)P$_2$ is bound and the AH is inserted, semi-stable interactions of sterols, mostly with bulky side chains of the AH, are able to form and the mobility of the other lipids (and probably that of MCC–eisosome-resident proteins, such as Nce102) within the membrane microdomain is reduced. In the presence of both PI(4,5)P$_2$ and sterols, specific stable interactions between Pil1 and PS—including the acyl tails of PS—occur, which suggests that the PS acyl tail profile could have a role in these interactions. Notably, the stabilized lipid interactions we observe are limited to the cytoplasmic leaflet, showing clear membrane asymmetry in the intra-leaflet lipid dynamics.

Under mechanical stress, Pil1/Lsp1 lattice stretching could conceivably be communicated to the lipids in the cytoplasmic leaflet through the direct connection between the Nt lattice contact sites and the AH. In our native-source structures, we see that Pil1 lattice stretching is correlated with a mobilization of PI(4,5)P$_2$ and PS and a loss of the sterol patterning within the cytoplasmic leaflet (Fig. 5e), which implies an increased mobility of all of the lipids within the MCC–eisosome. Given that all of our Pil1 lipid-binding-pocket mutants—even those with mild morphology defects—exhibit a mislocalization of the tension-responsive protein Nce102, we propose that this lipid mobilization represents a general mechanism to free sequestered factors to initiate their signalling functions that are sensitive to membrane stress (see Supplementary Discussion for further details).

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

## Methods

### Yeast strains

Yeast strains used for endogenous protein expression and purification were constructed using classical recombination methods. Yeast strains with point mutations were constructed using CRISPR–Cas9-based methods. The strains used in this study are listed in Supplementary Table 5.

### Purification of native-source eisosomes

Eisosome tubules were isolated in the following manner. Using *S. cerevisiae* expressing a TAP-tag on the endogenous locus of the target of rapamycin complex 2 (TORC2) subunit Bit61, we performed a gentle purification procedure. Yeast were grown to an optical density at 600 nm ($OD_{600\,nm}$) of 6–8, collected by centrifugation at 6,000 rpm for 10 min, flash frozen in liquid nitrogen and stored at −80 °C. Cells were lysed by manual grinding with a pestle and mortar under liquid nitrogen, then resuspended by slow rotation at 4 °C in 1.5 volumes of extraction buffer containing CHAPS detergent at less than one-tenth of the critical micelle concentration[55] (50 mM PIPES pH 7, 300 mM NaCl, 0.5 mM CHAPS, 0.5 mM DTT plus 1 mM PMSF and 1× cOmplete protease inhibitor cocktail (−EDTA) (Roche)). Lysates were cleared by centrifugation at 12,000 rpm for 10 min, and supernatants were incubated with IgG-coupled Dynabeads M270 (Thermo Fisher Scientific) for 2 h at 4 °C. Beads were washed five times with wash buffer (50 mM PIPES pH 7, 300 mM NaCl, 1 mM CHAPS and 0.5 mM DTT) at 4 °C, then incubated with TEV protease (0.1 mg ml⁻¹) for 1 h at 18 °C. Eluate was collected at 4 °C then used immediately for cryo-EM grid preparation.

### Mass spectrometry of native-source eisosomes

Pil1 is a common contaminant in *S. cerevisiae* pull-downs[56] and the amount of Pil1/Lsp1 protein isolated by our methods is below the detection limit of our protein gels (Extended Data Fig. 1h), suggesting that our eisosome filaments are a contamination of our intended target. Nevertheless, the large tubulated structures that they form were salient features on the EM grid, enabling us to collect a sufficiently large dataset for structural determination through helical reconstruction. Ultimately, we were able to clearly assign protein identity with our structural data owing to the resolution we achieved, and to confirm the presence of Pil1 and Lsp1 in these preps using mass spectrometry (Extended Data Fig. 1d and Supplementary Data 1).

The native-source eisosome sample (around 0.5 µg) was separated on an SDS–PAGE gel and stained with Coomassie Brilliant Blue. The entire lane was cut into six pieces, destained (200 µl, 50 mM NH₄HCO₃ in 50% acetonitrile (ACN)), washed (200 µl, ACN), and dried for 30 min in SpeedVac in separate tubes. Dried gel fractions were rehydrated (200 µl, 10 mM DTT in 50 mM NH₄HCO₃) for 1 h at 56 °C shaking at 400 rpm. Liquid was removed, and samples were alkylated (200 µl, 50 mM iodoacetamide in 50 mM NH₄HCO₃) at room temperature for 45 min shaking at 400 rpm in dark. Samples were washed twice with 200 µl, 50 mM NH₄HCO₃ for 10 min shaking at 600 rpm and twice with 200 µl, ACN for 10 min shaking at 600 rpm. Samples were dried completely in SpeedVac, then rehydrated with cold trypsin + LysC stock solution (1 µg trypsin + 0.25 µg LysC per 100 µl 50 mM NH₄HCO₃) on ice. Once all the solution was absorbed, 50 mM NH₄HCO₃ was added to cover the gel pieces completely, and samples were kept on ice for 5 min. Then, samples were incubated o/n at 37 °C shaking at 400 rpm. Next day, the samples were centrifuged, the supernatant was collected into a new tube and peptide extraction was performed with 100 µl of 50% ACN + 5% formic acid at room temperature for 30 min shaking at 400 rpm. Samples were centrifuged, the supernatant was collected into a new tube and peptide extraction was repeated with the pellet once more. Supernatants from the same gels were combined and dried in SpeedVac to a final volume of approximately 40 µl. Fifty microlitres of water was added and samples were desalted with C18 columns. Liquid

chromatography with tandem mass spectrometry was performed by nanoflow reverse-phase liquid chromatography (EASY-nLC 1000, Thermo Fisher Scientific) coupled to an Orbitrap Fusion Lumos Tribrid mass spectrometer (Thermo Fisher Scientific). Raw data files were analysed using MaxQuant v.1.6.6.0 using default parameters for false discovery rate (FDR), fixed and variable modifications.

Because our samples are likely to contain a mixture of both Pil1 and Lsp1 proteins and their sequence conservation is very high, we could not conclusively differentiate between these two proteins in our structures (Extended Data Figs. 2c and 4a; 73% identity, *MUSCLE*). Analysis of the mass spectrometry data of our native-source preparations yielded an average intensity ratio of 3.1:1 Pil1:Lsp1 peptides (Extended Data Fig. 1d and Supplementary Data 1).

### Plasmids

*PIL1* was cloned into the pCoofy6 vector (a gift from S. Suppmann, Addgene plasmid 43990) as described previously[57] with the following primers: LP1-Sumo3-Pil1-fwd 5′-GTGTTCCAGCAGCAGACCGGTGG AATGCACAGAACTTACTCTTTAAG, LP2-ccdB-Pil1-rev 5′-CCCCAG AACATCAGGTTAATGGCGTTAAGCTGTTGTTTGTTGGGGAAG, LP2-ccdB-fwd 5′-CGCCATTAACCTGATGTTCTGGGG and LP1-Sumo3-rev 5′-TCCACCGGTCTGCTGCTGGAACAC. After PCR amplification using Q5 High-Fidelity 2X Mastermix (M0492, New England Biolabs), gel-purified DNA fragments were assembled using RecA recombinase (M0249, New England Biolabs). A plasmid containing mCherry was a gift from A. Michelot. The mCherry was cloned into the C terminus of Pil1 with the following primers: LP1-Pil1-linker-mCherry-fwd 5′-TCTCT TCCCCAACAAACAACAGCTGAGCTCGCTGCAGCAATGGTGAG, LP2-mCherry-rev 5′-TGGTGCTCGAGTGCGGCCGCAAGCCTAGTTTC CGGACTTGTACAGCTC, LP1-Pil1-rev 5′-AGCTGTTGTTTGTTGGGGAA GAGAC and LP2-pCoofy-vector-fwd 5′-GCTTGCGGCCGCACTCGA GCACCAC. The DNA fragments were amplified and assembled as above. *PIL1* mutant DNA was PCR-amplified from genomic DNA extracted from mutant yeast strains using LP1-Sumo3-Pil1-fwd primer and LP2-mCherry-Pil1-rev primer 5′-GCTCACCATTGCT GCAGCGAGCTCAGCTGTTGTTTGTTGGGGAAGAGAC. The vector containing 6×His-Sumo3 and mCherry was linearized with LP2-mCherry-FWD primer 5′-GAGCTCGCTGCAGCAATGGTGAGC and LP1-Sumo3-rev primer. The PCR amplification was performed as described above. Gel-purified DNA fragments were assembled using NEBuilder HiFi DNA Assembly Cloning Kit (New England Biolabs). The sequence of each plasmid was verified by Sanger sequencing (Microsynth).

### Purification of Pil1 protein

Recombinant Pil1 and Pil1–mCherry were expressed in BL21(DE3) pLysS (200132, Agilent Technologies) in auto-induction LB medium (AIMLB0210, Formedium) overnight at 20 °C. Cells were lysed in lysis buffer (20 mM HEPES, pH 7.4, 150 mM KCl, 2 mM MgAc and 30 mM imidazole) supplemented with 1% Triton X-100, 1 mM PMSF and cOmplete protease inhibitor cocktail (5056489001, Roche) by sonication on ice. Proteins were first purified with a HisTrap Fast Flow column (GE17-5255-01, Cytiva) in the Äkta Pure system (Cytiva) using a gradient of imidazole from 30 mM to 500 mM. Proteins were subsequently dialysed overnight with 20 mM HEPES, pH 7.4, 75 mM KCl and 2 mM MgAc buffer, and further purified with a HiTrap Q sepharose HP column (17115401, Cytiva). A KCl gradient from 75 mM to 500 mM was used to elute the protein. To cleave off the Sumo3 tag, SenP2 protease at a final concentration of 30 µg ml⁻¹ was then added and the protein was dialysed with 20 mM HEPES, pH 7.4, 150 mM KoAc and 2 mM MgAc buffer overnight. Finally, protein samples were cleaned with a Superdex 200 Increase 10/300 GL column (GE28-9909-44, Cytiva) equilibrated with 20 mM HEPES, pH 7.4, 150 mM KoAc and 2 mM MgAc buffer. Proteins were concentrated to 20–25 µM, snap-frozen with liquid nitrogen and stored at −80 °C.

Recombinant Pil1–mCherry mutants R126A and K130A/R133A were expressed in BL21(DE3)pLysS (200132, Agilent Technologies) in auto-induction LB medium (AIMLB0210, Formedium) with kanamycin overnight at 20 °C. Recombinant Pil1–mCherry mutants R43A and K66A/R70A were expressed in Rosetta2 (DE3) pLysS (71397, Novagen) in LB medium with kanamycin and chloramphenicol at 37 °C for 4 h followed by overnight induction at 18 °C with 0.1 mM isopropyl-β-ᴅ-thiogalactopyranoside (IPTG). Recombinant Pil1–mCherry mutant F33A/Y40A/F42A/F50A was expressed in Rosetta2 (DE3) pLysS (71397, Novagen) in LB medium with kanamycin and chloramphenicol and grown at 37 °C to an $OD_{600 nm}$ of 0.6–0.8. Next, 1 mM IPTG was added, followed by growth for 3 h at 37 °C. The induced cells were collected by centrifugation and resuspended in lysis buffer (20 mM HEPES, pH 7.4, 150 mM KCl, 2 mM MgAc, 30 mM imidazole, 0,15% CHAPS, 1 µg ml$^{-1}$ DNase and 1 µg ml$^{-1}$ lysozyme), supplemented with protease inhibitors (1 mM PMSF and cOmplete EDTA-free protease inhibitor cocktail (73567200, Roche)). Cells were lysed using an Emulsiflex system (Avestin) and cleared by centrifugation at 15,000 rpm for 45 min at 4 °C. The soluble fraction was purified using a HisTrap Fast Flow crude column (1752801, Cytiva) on an Äkta Explorer-HPLC (GE Healthcare). The protein was washed with the same buffer and elutated with a gradient of imidazole from 30 mM to 500 mM in the same buffer. The purest fractions were desalted on a HiPrep Desalting column (17508701, Cytiva) against 20 mM HEPES, pH 7.4, 150 mM KCl, 2 mM MgAc and 30 mM imidazole. To cleave off the Sumo3 tag, SenP2 protease at a final concentration of 50 µg ml$^{-1}$ was then added and the reaction was performed in the cold room (4–6 °C) overnight. Cleaved protein was separated from the tag, the protease and the contaminants by reapplication to the HisTrap column. Finally, protein samples were applied on a Superdex 200 prep 16/600 (28989335, Cytiva) equilibrated with 20 mM HEPES, pH 7.4, 150 mM KoAc and 2 mM MgAc buffer. Proteins were concentrated to 15 µM for R43A mutant and 38 µM for the others, aliquoted, snap-frozen in liquid nitrogen and stored at –80 °C.

**Reconstitution of Pil1 tubules**

The lipids used in this study are listed in Supplementary Table 2. The selection of the subset of lipid compositions used for this study was based on our ability to observe tubulation that was sufficiently robust for cryo-EM studies. The variations in conditions we tried were combinations of the following variables: (1) ±20% DOPE; (2) 0.5% versus 2% versus 10% PI(4,5)P$_2$; (3) brain PI(4,5)P$_2$ versus 18:1 PI(4,5)P$_2$; (4) cholesterol versus ergosterol; (5) combination of PI(4,5)P$_2$ with 15% versus 30% cholesterol. Although these mixtures do not capture the full complexity of the native plasma membrane, using these reconstitutions, we were able to make several salient observations.

To reconstitute Pil1 tubules using large unilamellar vesicles (LUVs) for cryo-EM, lipids were mixed in chloroform to a final concentration of 3.8 mM with the desired molar ratios. Chloroform was evaporated under argon gas flow and subsequently for three hours in a 30 °C vacuum oven. A lipid film was hydrated in reaction buffer (20 mM HEPES, pH 7.4, 150 mM KoAc and 2 mM MgAc), subjected to 10 cycles of freeze-thaw and extruded through a 200-nm-pore-sized polycarbonate filter (Cytiva) using a mini-extruder (Avanti Polar Lipids). Lipid compositions used in cryo-EM studies are listed in Supplementary Table 3. To produce samples for cryo-EM studies, a mixture of 15–20 µM recombinant Pil1 and around 2 mg ml$^{-1}$ LUVs was incubated at 30 °C for one hour before freezing.

The vast majority of the conditions tested yielded some amount of tubulation (except DOPC:DOPS alone), but we found that the most robust tubulation occurred in mixtures containing 20% DOPE, 10% brain PI(4,5)P$_2$ and cholesterol, rather than ergosterol. We were surprised that ergosterol gave us slightly less robust tubulation, because it is the main sterol species in yeast (they do not produce cholesterol). Nevertheless, cholesterol recapitulates the void pattern that we observed in the native-like eisosomes, which almost certainly contain ergosterol, and

not cholesterol. We also confirmed using FRAP assays that no significant differences in either TF-PI(4,5)P$_2$ or TF-PS fluorescence recovery were observed when 30% cholesterol was replaced with 30% ergosterol (Extended Data Fig. 6a and Supplementary Table 4). This suggests that the structural features of eisosome proteins that enable sterol coordination are likely to be conserved across species.

To reconstitute Pil1 tubules on preformed membrane nanotubes for fluorescence microscopy experiments, supported lipid films over silica beads were formed from multilamellar vesicles by mixing lipids in chloroform to a final concentration of 1 mg ml$^{-1}$ and evaporating the solvent as described previously[58]. In brief, lipid films were hydrated using 5 mM HEPES, pH 7.4 buffer. Multilamellar vesicles were then mixed with silicon dioxide microspheres (Corpuscular 140256-10 or Sigma-Aldrich 904384) and dried for 30 min in a 30 °C vacuum oven. The imaging chamber was prepared by attaching a sticky-Slide VI 0.4 (Ibidi, 80608) on a 24 × 60-mm microscope cover glass. Sample chambers were passivated for 10 min with 2 g l$^{-1}$ bovine serum albumin solution and subsequently washed several times with reaction buffer (20 mM HEPES, pH 7.4, 150 mM KoAc and 2 mM MgAc). Lipid-coated silica beads were then hydrated by adding a small amount of beads in the sample chamber and allowing beads to roll through the chamber. Several different lipid compositions were used in these experiments (See Supplementary Table 3). Then, 0.01 mol% of Atto647N DOPE was added to each lipid mixture to visualize nanotubes and as the reference fluorescent lipid for measuring lipid sorting coefficients.

DO- lipids were used for all these studies despite their low frequency in the yeast plasma membrane because their melting temperature is –18 °C. PO- or saturated lipids are more prone to phase changes during cooling. To confirm that DO- lipids recapitulate the behaviour in the eisosome filaments of the more physiological PO- lipids, we performed MD simulations and FRAP assays replacing DOPC:DOPE:DOPS 30:20:20 with POPC:POPE:POPS 30:20:20. Lipid occupancy trends were broadly similar in these MD simulations, and no significant differences were observed for either TF-PI(4,5)P$_2$ or TF-PS in the FRAP assays (Extended Data Figs. 6b,c and 7g and Supplementary Table 4).

**Cryo-EM grid preparation and data collection**

Five microlitres of fresh sample was applied to untreated lacey carbon film on copper mesh grids (Jena Bioscience X-170-CU400), blotted for 3-4 s, then re-applied, blotted for 2–4 s (second blot) and finally plunge-frozen in a Leica GP2 plunge system at 18 °C, 90% humidity.

Native-source eisosome filaments were imaged by targeted acquisition using SerialEM with a 300 kV Titan Krios fitted with a Gatan K2 Quantum direct electron detector (Heidelberg). A total of 2,827 movies were collected, each with a total dose of 40 e$^-$ Å$^{-2}$, a target defocus range of –0.8 to –1.8 µm and a pixel size of 1.327 Å (105,000× magnification).

Reconstituted Pil1 filaments were imaged using EPU v.2.14 software with a 300 kV Titan Krios and a Falcon 4 direct detector (DCI Lausanne). Three datasets were collected: (1) Pil1 + 'minus PI(4,5)P$_2$' liposomes (21,386 movies); (2) Pil1 + 'minus cholesterol' liposomes (22,960 movies); and (3) Pil1 + 'PI(4,5)P$_2$/cholesterol' liposomes (22,408 movies). For each movie, a total dose of 50 e$^-$ Å$^{-2}$, a target defocus range of –0.6 to –1.8 µm and a pixel size of 0.83 Å (96,000× magnification) was used.

**Cryo-EM data processing**

The pipeline for cryo-EM data processing is outlined in Extended Data Figs. 1 and 6.

For native-source eisosomes, movies were aligned using Motion-Cor2[59], and CTF correction was completed using Gctf v.1.06[60]. Filaments were handpicked using manual picking in RELION v.2.1.0. Two-dimensional (2D) classification was run iteratively in RELION v.2.1.0 to sort particles into clean sets of similar diameter and helical arrangement. In our raw images and 2D class averages, we noted a large variation in tubule diameters (Extended Data Fig. 1f,g). For particles in each clean RELION 2D class, power spectra were summed by class,

then manually sorted into identical 'types'. Helixplorer-1[61] was used to estimate helical parameters, which were used with particles from each 'type' for 3D auto-refinement with helical parameters in RELION. All helix types were then corrected for handedness and aligned along the $D$ symmetry axis (using $C$ symmetry worsened resolution). A mask was generated in RELION covering the central third of the helix and a final round of helical refinement was completed either with the mask (to optimize resolution) or unmasked (to be used for particle subtraction). Resolution estimates for masked maps are based on gold standard FSC values with a 0.143 cut-off on post-processed maps with the one-third mask used for refinement, a manually chosen initial threshold and auto-b-factor calculation.

To improve resolution and enable 3D classification of the Pil1/Lsp1 dimers, we used a symmetry expansion and density subtraction strategy (Extended Data Fig. 1c). This allowed us to merge lattice pieces from the nine helical structures into an expanded dataset. Symmetry expansion and density subtraction on native-source filaments was completed using unmasked maps from the final iteration of 3D auto-refinement. Helical parameters for each helix type were used for symmetry expansion, except using $C$ symmetry instead of $D$ symmetry to produce dimer particles. A mask for density subtraction was generated in Chimera v.1.16 through the addition of two zone maps: (1) an 8-Å zone using models of a central Pil1/Lsp1 dimer and the six dimers with which it shares lattice contact sites; and (2) a spherical zone of 60 Å centred on the AHs of the Pil1/Lsp1 dimer. This initial mask was extended with a soft edge, then used for density subtraction and reboxing of the particles in RELION v.3.1.3. Density-subtracted particles were used to reconstruct a volume and particle set for all helix 'types' and the reconstructed volumes were imported into cryoSPARC v.4.1.2 for further processing. Homogenous refinement was completed with all particles with the reconstructed volume as the initial model using $C2$ symmetry. This map was used for refinement of the Pil1 and Lsp1 native-source models (see 'Model building' for details). This map was then symmetry expanded in $C2$ for 3DVA.

For reconstituted Pil1 filaments, data processing was completed in cryoSPARC v.4.1.2. Movies were processed with CryoSPARC Live v.3.2.2, using patch motion correction and patch CTF estimation. Filaments were picked using Filament Tracer, then cleaned and sorted using iterative rounds of 2D classification. Clean classes were used to calculate average power spectra, which were then manually sorted into identical 'types'. Helixplorer-1 was used to estimate helical parameters, which were used with particles from each 'type' for helical refinement. After initial refinement, all helix types were corrected for handedness and aligned along the $D$ symmetry axis. A mask on the central third of the helix was created in RELION v.3.1.3 and used for an additional round of helical refinement. A final round of helical refinement with non-uniform refinement enabled was used to improve the resolution in the lipid-binding pocket for the best resolved maps from the $+PI(4,5)P_2/-$sterol and $PI(4,5)P_2/+$sterol datasets (cryoSPARC v.4.4.0). These maps were used for real-space refinement of the Pil1 lattice ($-PI(4,5)P_2/+$sterol reconstituted), Pil1 lattice ($+PI(4,5)P_2/-$sterol reconstituted) and Pil1 lattice ($+PI(4,5)P_2/+$sterol reconstituted) models (see 'Model building' for details). Resolution estimates are based on gold standard FSC values with a 0.143 cut-off using an optimized mask automatically generated during refinement.

To identify dimensions of continuous heterogeneity in the symmetry-expanded and density-subtracted dataset, we used 3DVA, which enables both the resolution and the visualization of flexible movements within cryo-EM datasets[62]. The analysis completed with five components requested. Manual inspection revealed one component that exhibited obvious lattice stretching. This component was used for 3D variability display in intermediate mode with ten non-overlapping frames used to generate particle subsets. Each particle subset was then used for masked local refinement with a mask covering the central dimer (generated in Chimera using the Pil1/Lsp1 dimer with a 10-Å zone

and extended with a soft edge in RELION). These maps were used for refinement of the Pil1 compact and stretched (near-native) models (see 'Model building' for details). Sharpening with deepEMhancer[63] was used to improve the resolution of lipid headgroups in the lipid-binding pocket. Resolution estimates are based on gold standard FSC values with a 0.143 cut-off using an optimized mask automatically generated during refinement.

Parallel slice images were made in Fiji v.1.54f using the Reslice tool without interpolation. Three-dimensional intensity plots were made in Fiji using the 3D surface plot tool on the slice with maximum sterol void intensity using identical display conditions on both maps. Radial angle profile plots were made in Fiji using the Radial Profile Extended plug-in with an angle of 40°, corresponding to the size of the sphere used in density subtraction to include the lipid density under the central dimer.

Figures were made in Chimera v.1.16 or ChimeraX v.1.5. Three-dimensional sterol void visualization figures were produced by displaying the membrane density at a high threshold, so that the membrane density appears as a solid tubule, and applying a Gaussian filter of 1.2 s.d. to smoothen the topological features of the membrane density. Next, a zone map of around 12 Å from residues of the AH of dimer models was generated and then coloured by local resolution (in Å). The zone map provides a cut-out window into the inside of the membrane density adjacent to the AH, producing an inverted view of the topological features within the membrane density (Extended Data Fig. 5a–c,g,h).

## Model building

Structure predictions for Pil1 and Lsp1 from the AlphaFold database (https://alphafold.ebi.ac.uk/) were used as starting models, with the C-terminal region removed, starting from residue 275, for which no density was observed. Iterative rounds of model building, performed in Coot v.0.8.9.2, and real-space refinement, performed in PHENIX v.1.20-4459, were completed until no improvement in the model was observed. The model quality and fit to density were performed using PHENIX v.1.20-4459.

Ligand constraints for inositol 2,4,5-triphosphate, phosphatidylserine and phosphoserine were produced using phenix.elbow. Refinements with ligands were performed with these ligand constraints. For lipid headgroup ligands in native-source eisosome compact dimer model, ligands refined in the Pil1 $+PI(4,5)P_2/+$sterol reconstituted map were placed in the in the deepEMhancer sharpened native-source eisosome compact map, then adjusted in Coot v.0.8.9.2 with rigid body fitting.

An electrostatic potential map of the model surface was calculated using the Coulombic potential function of ChimeraX v.1.5.

## CD spectrometry

Small unilamellar vesicles with a lipid composition of either 80 mol% DOPC + 20 mol% DOPS or 70 mol% DOPC + 20 mol% DOPS + 10 mol% brain $PI(4,5)P_2$ were produced by mixing lipids in the desired ratios in chloroform, evaporating chloroform under argon gas and finally drying in a vacuum oven. Lipids were hydrated in 5 mM Na-phosphate buffer pH 7.4, followed by thorough sonication in water bath. C-terminally aminated Pil1 amphipathic peptide (GKGGLAYSFRRSAAGAFGPEL) was synthesized by GenScript and peptide was dissolved in 5 mM Na-phosphate buffer pH 7.4. CD spectra were measured using the peptide at a final concentration of 50 µM in Na-phosphate buffer either with or without small unilamellar liposomes at a final concentration of 250 µg ml$^{-1}$. Measurements were performed with a Jasco J-815 circular dichroism spectrophotometer using a high precision quartz cuvette with a 1-mm light path. Spectra were recorded between 190 nm and 250 nm with 1-nm increments and a scanning speed of 10 nm per min. The molar ellipticity was calculated by using Spectra Manager Analysis software v.2.14.02 (Jasco) and alpha helical character was assessed using previously described methods[64]. We note that the use of small unilamellar vesicles causes a high scattering effect especially in the wavelength

range below 200 nm. Each measurement was performed three times and we show the mean of these measurements.

### Lipid diffusion measurements with FRAP
Lipid nanotubes were prepared as described above and 200 nM of Pil1–mCherry was incubated with nanotubes for 30 min. FRAP experiments were performed with an Olympus IX83 wide-field microscope equipped with an Olympus Uapo N 100× 1.49 oil objective and an ImageEM X2 EM-CCD camera (Hamamatsu). The system was controlled by the Visiview v.4.4.0.11 software (Visitron Systems). Five frames were captured before a small protein-coated region was bleached. Subsequently, the recovery of fluorescence intensity was measured by capturing images every 500 ms for 1–2 min (see Supplementary Methods for a description of the analysis).

### Measurements of lipid sorting coefficients
Lipid nanotubes were prepared as described earlier and incubated with 200–400 nM of Pil1–mCherry for 30 min until protein scaffolds were formed and visible by fluorescence microscopy. Imaging was performed using an inverted spinning disk microscope assembled by 3i (Intelligent Imaging Innovation) consisting of a Nikon Eclipse C1 base and a 100× 1.3 NA oil immersion objective. Fluorescence microscopy images were collected using SlideBook software v.6.0.22 (Intelligent Imaging Innovations). Lipid sorting experiments performed with Pil1 mutants were performed in the same manner as those performed with the WT protein (see Supplementary Methods for a description of the analysis).

### MD simulations
In accordance with the experimental models, three systems with different lipid compositions were modelled and simulated. CG MD simulations of Pil1 tubule interacting with the tubule membranes were performed, in duplicates.

The tubule membranes were built using the BUMPy tool v.1.1[65]. The systems, without the protein, were solvated with water and minimized using the steepest descent algorithm[66]. Four equilibration steps were performed, as follows. (1) a first equilibration with a time step of 5 fs was run for 10 ns, imposing position restraints with a force constant (fc) of 300 kJ per mol per nm$^2$ on the lipid tails, to allow the formation of membrane pores to equilibrate the lipid and water content between the tubule lumen and the external region. (2) A second equilibration step of 5 ns was performed using the previous settings but increasing the fc to 500 kJ per mol per nm$^2$ and the time step to 10 fs. (3) A third equilibration step was run for an additional 10 ns after the removal of position restraints to allow the closure of the pores. The Berendsen barostat and the v-rescale thermostat[67,68] (with a temperature of 303 K) were used.

From the cryo-EM structure of Pil1 tubule (21 dimers), CG mapping was performed using Martinize2[69], imposing an elastic network within the dimers. The CG protein model was then manually positioned around the tubule membrane using VMD v.1.9[70]. The final system was solvated with water beads and neutralized adding Na$^+$ and Cl$^-$ ions. Each system was minimized and equilibrated in seven steps. (1) A first equilibration with a time step of 5 fs was run for 10 ns, imposing position restraints with a fc of 300 kJ per mol per nm$^2$ on the lipid tails, to allow water pore formation. The Berendsen barostat[67] was applied to all the direction, with $\tau_p = 5$, and the v-rescale thermostat was used, setting the temperature at 303 K (ref. 68). (2) A second equilibration step of 5 ns was performed using the previous setting but increasing the fc to 500 kJ per mol per nm$^2$, and the time step to 10 fs. (3) A third equilibration step was run for another 10 ns, increasing the fc to 1,000 kJ per mol per nm$^2$, to maintain the waterpores open and to allow for the solvent equilibration. (4) Starting from the fourth step, the fc on the tails was progressively reduced to slowly induce a slowly closure of the pores. Finally, a fc of 500 kJ per mol per nm$^2$ was applied, to run an equilibration

of 5 ns. (5) An equilibration decreasing the fc at 300 kJ per mol per nm$^2$ was performed for another 5 ns. (6) An equilibration removing the fc on the lipid tails was performed to allow the complete closure of waterpores. (7) A final equilibration step of 10 ns was run without restraints, increasing the time steps to 20 fs.

For systems containing cholesterol, the equilibration procedure was extended including an additional first equilibration step, with a reduced time step of 2 fs.

For production, the Parrinello–Rahman barostat[71] was used with $\tau_p = 12$. For each system, two replicates of approximately 10 μs were performed. The simulations were performed using GROMACS v.2021.5[66] and the Martini3 force field[72,73].

To compute lipid occupancy, the PyLipID python package was used[74]. The analysis was performed selecting the headgroups of lipids. The values were averaged over time and over the dimers.

### Synthesis of bromosterols
Bromosterol was synthesized by adding halogen to the double bond as previously described[75]. Br$_2$ (45 μl, 0.9 eq, 0.87 mmol) was added dropwise to a solution of 400 mg (1 eq, 1 mmol) ergosterol in 40 ml CH$_3$Cl. The mixture was stirred on ice in the dark for 30 min and a 10% solution of Na$_2$S$_2$O$_3$ was added. The organic layer was separated from the aqueous layer and the latter was extracted with CH$_2$Cl$_2$ and dried over Na$_2$SO$_4$. The solvent was removed under reduced pressure and the product was purified by flash chromatography. The brominated ergosterol was stored as a powder at −20 °C in the dark. Compounds were verified by nuclear magnetic resonance (Supplementary Data 2 and 3)

### Spot assays
Saturated overnight yeast cultures (30 °C, SC medium) were diluted to an OD$_{600 nm}$ of 0.1 in the morning and grown into mid log phase (OD$_{600 nm}$ = 0.5–0.8). Log phase cells were diluted to OD$_{600 nm}$ 0.1, and a tenfold dilution series was spotted onto SC medium plates containing treatment substances, or vehicle. Plates were incubated at 30 °C, except low (15 °C) temperature plates, and imaged when differences were most apparent (typically after 40 h for nystatin, 48 h for controls and atorvastatin, 72 h for myriocin and 168 h for 15 °C). Substance stocks used in this study: myriocin (Sigma M1177) 2.5 mM in MeOH, nystatin (Sigma 475914) 50 mM in DMSO and atorvastatin (Sigma PHR1422) 20 mM in DMSO.

### Fluorescence microscopy
Logarithmically growing overnight yeast cultures (30 °C, SC medium) were diluted and grown to an OD$_{600 nm}$ of 0.6. For fluorescence live-cell microscopy, cells were loaded into a Concanavalin coated flow chamber (Ibidi μ-Slides VI 0.4 ibiTreat). Microscopy was performed at room temperature with a Zeiss LSM 980 microscope with Airyscan 2, using a 63× 1.4 NA oil immersion objective with Zeiss Zen 3.3.89.0008 (blue edition) software. Images were taken as z-series to generate 2D SUM projections.

For determining colocalization between Pil1–GFP and Nce102–mScarlet-I, cells were first segmented using Cellpose v.2.0[76]. Cells that were intersected by the image borders, and cells with an Nce102–Scarlet signal below the fixed threshold (thresholded area = 0) for calculating the Manders' colocalization coefficient M1 (fraction of Nce102–Scarlet overlapping with Pil1–GFP) were excluded. Manders' coefficients of single cells were obtained by analysis of 3D stacks in Fiji v1.54f, using the BIOP version of the JACoP plug-in[77] with fixed manual thresholds for Pil1–GFP and Nce102–mScarlet-I, and graphs were generated with Origin Pro 2022 v.9.9.0.225 (OriginLab).

### Statistics and reproducibility
Native-source protein purifications were repeated more than 20 times ($n > 20$), yielding similar results in Coomassie stained gels (Extended Data Fig. 1h) and a varying density of visible tubules in negative-stain

and/or cryo-EM micrographs. Reconstituted samples for EM studies were repeated at least three times for each lipid composition with a similar degree of tubulation observed in negative-stain and/or cryo-EM micrographs. For MD simulations, the sample size was determined empirically, considering the time necessary for equilibration of the lipids. For our systems, that is about 10 µs. Two replicates were performed for each lipid system. The FRAP data were combined from three individual experiments, with individual measurements from these experiments pooled together for the analysis. The raw data are available upon reasonable request. For lipid sorting coefficients, in all conditions $N = 2$, meaning that the experiments were repeated two different days (whereas $n$ refers to the number of independent tested nanotubes). To pool all data points from the two different days together, we performed the Wilcoxon–Mann–Whitney non-parametric test between data points obtained each day for each condition to ensure that there were no batch differences between the experimentation days. Statistical significance was determined with the two-sample $t$-test after testing the normality of the data distributions (all conditions following normal distribution at 0.01 tested by Shapiro–Wilk, Kolmogorov–Smirnov and Anderson–Darling normality tests). Box plot elements for lipid sorting data are defined as follows. The box indicates the interquartile range (IQR) from Q1 (25%) to Q3 (75%) quartiles. The bottom and top whiskers show from Q1 and Q3 quartiles to the minimum and maximum data points, respectively. The horizontal line shown inside the box indicates the median, black rhombuses show data points obtained at day 1 and grey circles show data points obtained for day 2 (Fig. 3c and Extended Data Fig. 9a). Yeast growth assays and microscopy were repeated at least three times on different days, yielding similar results (Fig. 4b and Extended Data Fig. 9c,d). For calculating Manders' overlap coefficient, microscopy data from several days were pooled to analyse at least 100 cells per mutant.

## Reporting summary

Further information on research design is available in the Nature Portfolio Reporting Summary linked to this article.

## Data availability

Mass spectrometry data have been deposited to the ProteomeXchange Consortium via the PRIDE partner repository with the dataset identifier PXD050326. Sharpened maps used for model refinement and all associated helical maps and deepEMhancer sharpened maps have been deposited in the Electron Microscopy Data Bank (https://www.ebi.ac.uk/emdb/) under the following accession codes: eisosome native-source (EMD-18307), Pil1 −PI(4,5)P$_2$/+sterol reconstituted (EMD-18308), Pil1 +PI(4,5)P$_2$/−sterol reconstituted (EMD-18309), Pil1 +PI(4,5)P$_2$/+sterol reconstituted (EMD-18310), Pil1 +PI(4,5)P$_2$/+bromosterol reconstituted (EMD-19822), eisosome native-source compact (EMD-18311) and eisosome native-source stretched (EMD-18312). Raw micrographs of native-source eisosome samples have been deposited in the Electron Microscopy Public Image Archive (EMPIAR) database under the accession code EMPIAR-12053. The starting model for building the Pil1 model was acquired from the AlphaFold Protein Structure Database (https://alphafold.ebi.ac.uk/) using the Uniprot accession number P53252 (PIL1_YEAST). All models have been deposited in the Protein Data Bank (https://www.rcsb.org/): Pil1 lattice (native-source) (PDB 8QB7), Lsp1 lattice (native-source) (PDB 8QB8), Pil1 lattice (−PI(4,5)P$_2$/+sterol reconstituted) (PDB 8QB9), Pil1 lattice (+PI(4,5)P$_2$/−sterol reconstituted) (PDB 8QBB), Pil1 lattice (+PI(4,5)P$_2$/+sterol reconstituted) (PDB 8QBD), Pil1 lattice compact (native-source) (PDB 8QBE), Pil1 dimer compact with lipid headgroups (native-source) (PDB 8QBF) and Pil1 lattice stretched (native-source) (PDB 8QBG). Lipid diffusion, lipid sorting and yeast cell biology data are provided at https://doi.org/10.26037/yareta:ubja4xykqzfjbhcfwmg7sgj2x4. Raw gels are provided in Supplementary

Data 4. All other data supporting the findings of this study are provided in this manuscript. Source data are provided with this paper.

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

**Acknowledgements** This work was supported by grants from the H2020 Marie Curie Actions IF-2020-101026765-MEMTOR (to J.M.K.); EMBO postdoctoral fellowships ALTF 703-2020 (to M.H.) and ALTF 989-2022 (to J.E.); the Swiss National Supercomputing Centre (to S.V.) under project IDs s1251, s1132 and s1221; the Swiss National Science Foundation (SNSF) under project CRSII5_189996 METEORIC (to R.L., A.R., S.V. and J.A.); the European Research Council (ERC) AdG TENDO (to R.L.) and the SNSF under project 310030_207754 (to R.L.). A.R. and R.L. acknowledge additional support from the Republic and Canton of Geneva. This work used the EM facilities at EMBL Heidelberg (iNext) as well as the Dubochet Center for Imaging (DCI) Geneva with assistance from Y. Sadian and A. Howe. Further cryo-EM data collection and initial image processing were performed at the Dubochet Center for Imaging Lausanne (a joint initiative from EPFL, UNIGE, UNIL and UNIBE) with the assistance of A. Myasnikov, B. Beckert, S. Nazarov, I. Mohammed and E. Uchikawa. Image processing was also performed at the Grenoble Instruct-ERIC Center (ISBG; UAR 3518 CNRS CEA-UGA-EMBL) with support from the French Infrastructure for Integrated Structural Biology (FRISBI; ANR-10-INBS-05-02) and GRAL, a project of the University Grenoble Alpes graduate school (Ecoles Universitaires de Recherche) CBH-EUR-GS (ANR-17-EURE-0003) within the Grenoble Partnership for Structural Biology. The IBS Electron Microscope facility is supported by the Auvergne Rhône-Alpes Region, the Fonds Feder, the Fondation pour la Recherche Médicale and GIS-IBiSA. We thank R. Puschmann, A. Bergman, L. Tafur and M. Prouteau for help with mass spectrometry, molecular biology, biochemistry and sample transport; M. Kaksonen and A. Picco for providing facilities and technical support for FRAP imaging and image analysis; and A. Frost, F. Moss III and N. Unwin for discussions.

**Author contributions** J.M.K. and R.L. designed the project. J.M.K. designed experiments, processed EM data, built and refined structural models, interpreted results and prepared

the manuscript. M.H. prepared reconstituted samples and designed and interpreted CD and FRAP experiments. L.Z. optimized and prepared native-source samples and produced CRISPR mutants. J.A., P.C. and S.V. designed, analysed and interpreted MD simulations. J.A. performed MD simulations. J.E. designed and interpreted lipid sorting experiments. M.G.T. performed and interpreted growth assays and fluorescence microscopy. J.G. synthesized brominated ergosterol. C.G. expressed and purified lipid-binding-pocket mutant proteins. L.F.E. coded tools to assist in sorting native-source eisosome data. N.E.S. analysed mass spectrometry data. R.L. supervised the project, S.V. supervised MD simulations, A.R. supervised in vitro reconstitution studies and A.D. supervised EM data processing. All authors discussed the results and commented on the manuscript.

**Competing interests** The authors declare no competing interests.

**Additional information**
**Correspondence and requests for materials** should be addressed to Robbie Loewith.

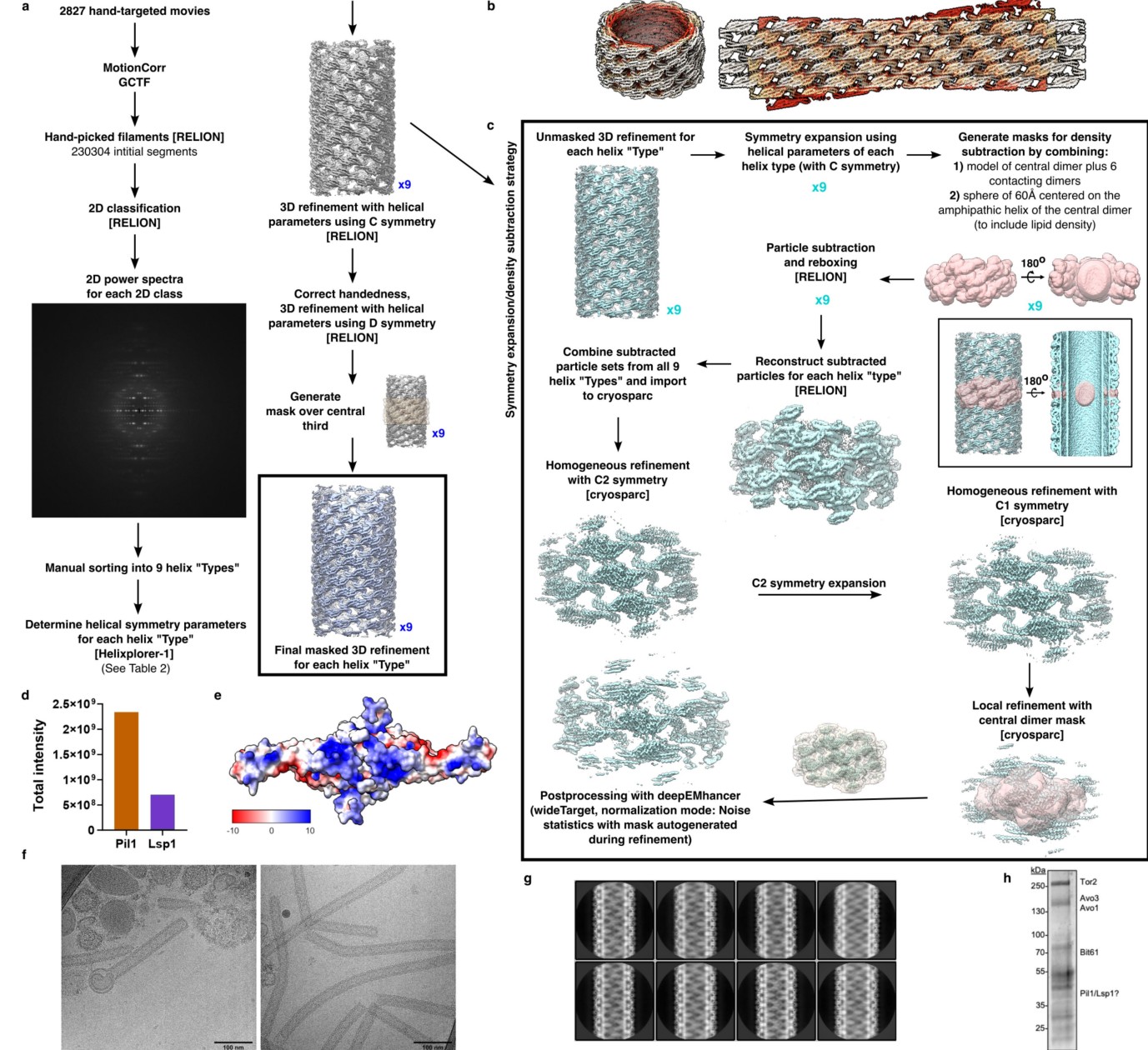

**Extended Data Fig. 1 | Purification and cryo-EM data processing of native-source eisosome filaments. a**, Helical reconstruction data processing strategy. **b**, Unrolled and aligned helical structures of native-source eisosome filaments of different diameters show nearly identical lattice pattern. **c**, Data processing strategy for symmetry expansion of helical reconstructions **d**, Total intensity of Pil1 and Lsp1 peptides in mass spectrometry analysis. Intensity ratio of Pil1:Lsp1 is 3.1:1. **e**, Electrostatic surface prediction of Pil1 model with potentials ranging from −10 kcal*mol⁻¹$e^{-1}$ (red) to +10 kcal*mol⁻¹$e^{-1}$ (blue). **f**, Two representative aligned micrographs with native-source MCC–eisosome

tubules and other putative contaminants visible. 2827 micrographs were collected for this dataset. **g**, Example 2D class averages with varying filament diameters. **h**, Representative Coomassie staining of protein gel of Bit61-TAP purification of MCC–eisosome tubules. Clear bands for TORC2 complex components Tor2, Avo3, Avo1 and Bit61-TAP are visible. The faint band at ~40 kDa is likely to correspond to Pil1/Lsp1. This protein purification was repeated n > 20 times yielding similar results. See Supplementary Data 4 for raw gel image.

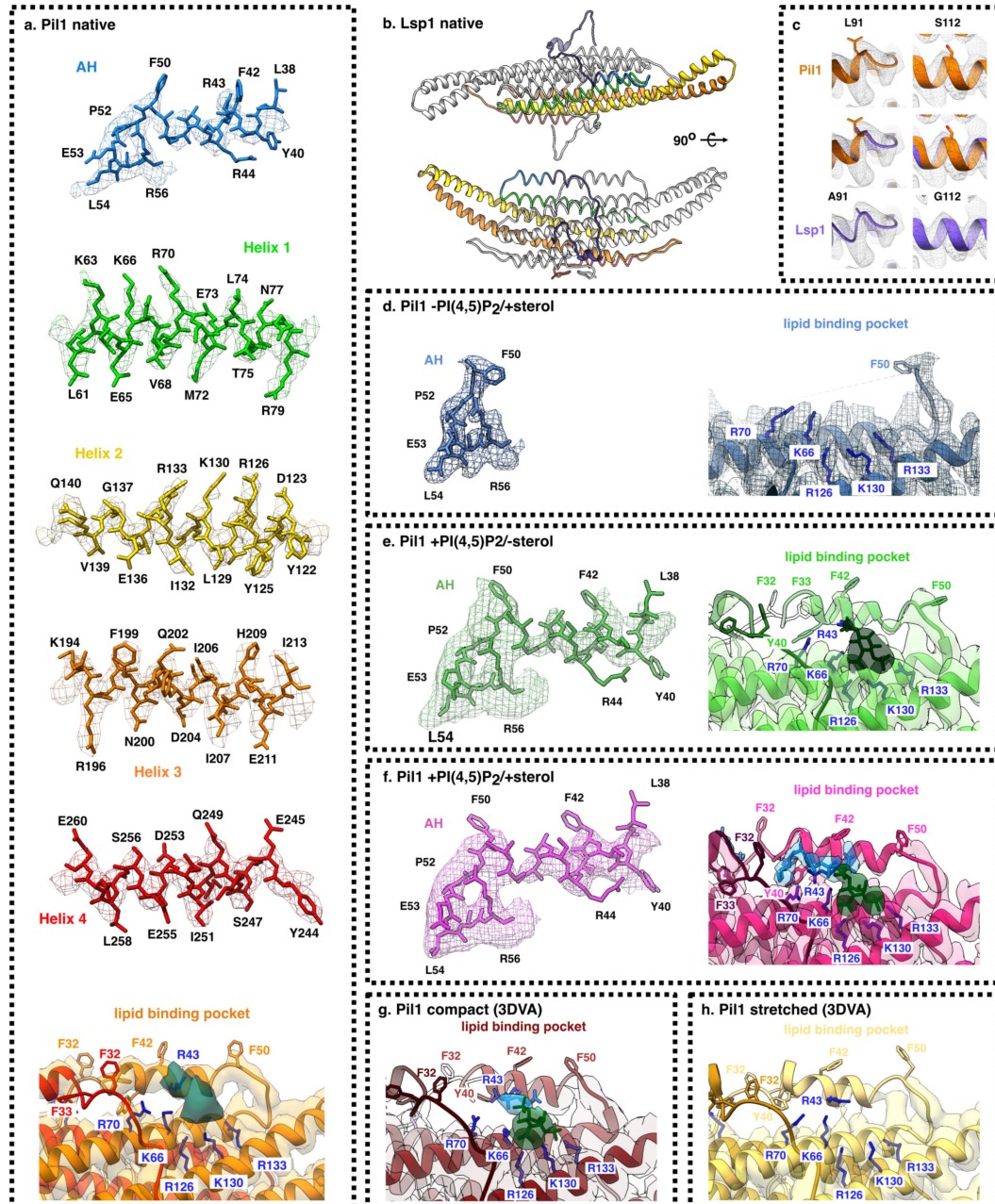

**Extended Data Fig. 2 | Map-to-model fit. a**, Pil1 native-source eisosome map-to-model fit and lipid-binding pocket bound to unassigned lipid density (green). **b**, Lsp1 model. **c**, Fit comparison of divergent residues in Pil1 and Lsp1. **d**, −PI(4,5)P$_2$/+sterol reconstituted map-to-model fit of unresolved AH and unoccupied lipid-binding pocket. **e**, +PI(4,5)P$_2$/−sterol reconstituted map-to-model fit of AH and lipid-binding pocket bound to PI(4,5)P$_2$ headgroup (dark green). **f**, +PI(4,5)P$_2$/+sterol reconstituted map-to-model fit of AH and lipid-binding pocket bound to PI(4,5)P$_2$ (green) and phosphatidylserine (PS; dodger blue) lipids. **g**, Most compact protein lattice class of 3D variability analysis (3DVA) map-to-model fit of lipid-binding pocket bound to PI(4,5)P$_2$ (green) and putative PS (dodger blue) headgroups. **h**, Most stretched protein lattice class of 3DVA map-to-model fit of unoccupied lipid-binding pocket.

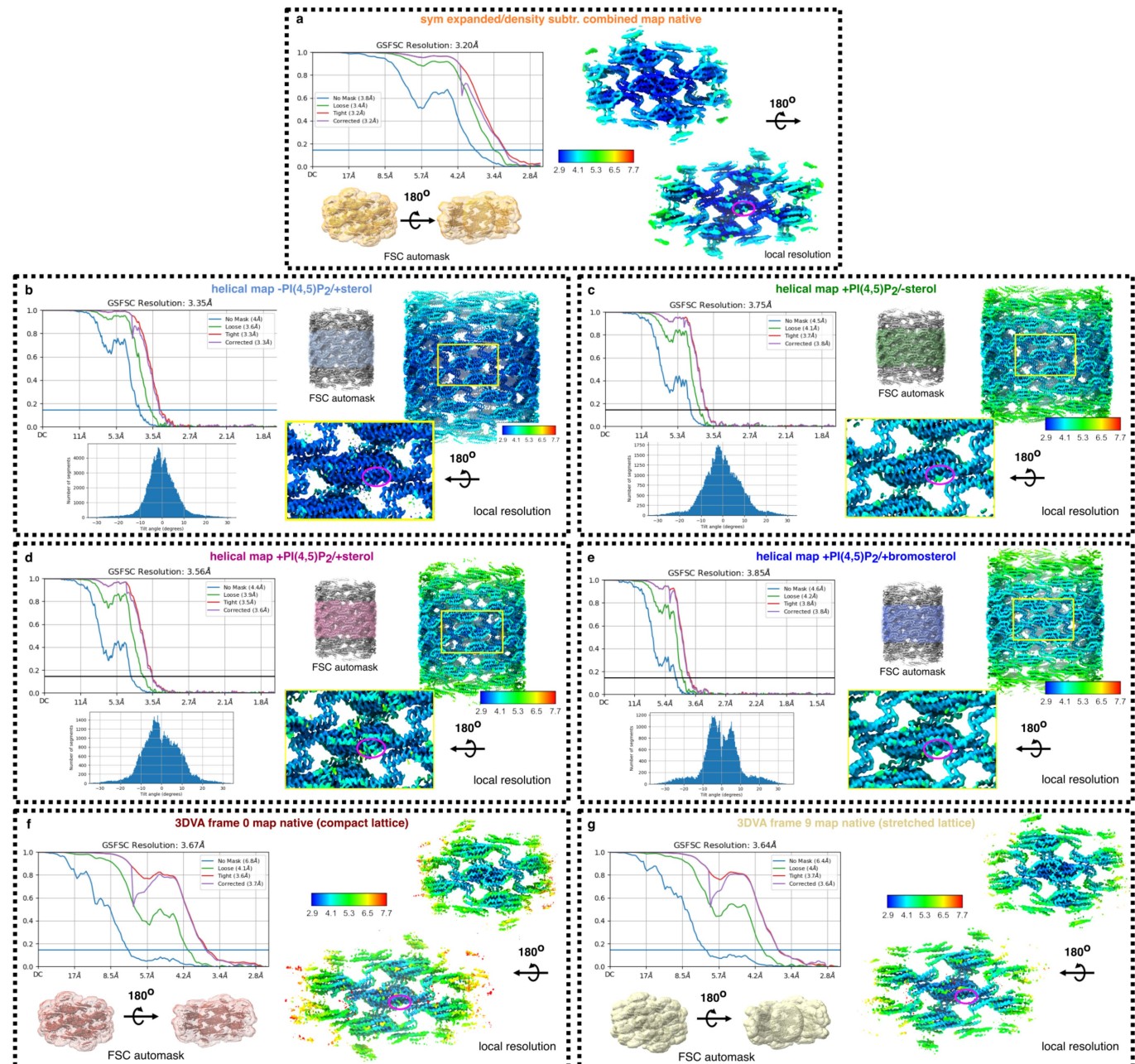

**Extended Data Fig. 3 | Map quality and local resolution. a**, Gold standard Fourier shell correlation (GSFSC) plots, auto-generated mask for average resolution determination at FSC 0.143, and local resolution map for symmetry-expanded native-source map. Magenta circle highlights lipid-binding pocket. **b–e**, GSFSC plots, helical symmetry error plot, auto-generated mask for average resolution determination at FSC 0.143, and local resolution map for helical maps of −PI(4,5)P$_2$/+sterol (**b**), +PI(4,5)P$_2$/−sterol (**c**), +PI(4,5)P$_2$/+sterol (**d**) and +PI(4,5)P2/+bromosterol (**e**) reconstituted Pil1 tubules. Magenta circles highlight lipid-binding pocket. **f,g**, GSFSC plots, auto-generated mask for average resolution determination at FSC 0.143, and local resolution map for 3DVA frame 0 (most compact) (**f**) and 3DVA frame 9 (most stretched) (**g**) refined maps. Magenta circles highlight lipid-binding pocket.

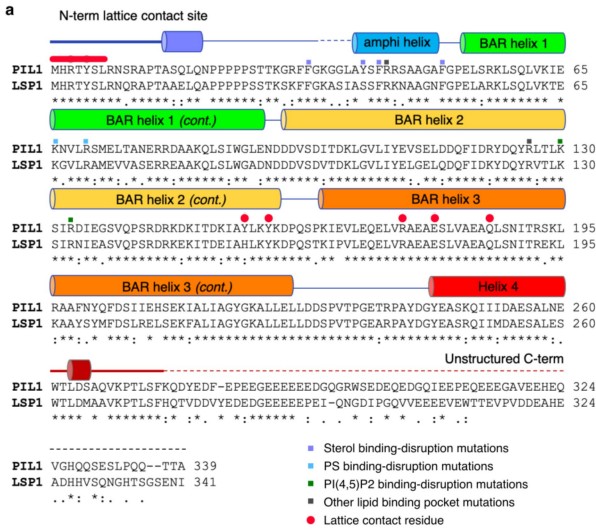
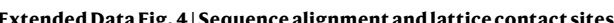
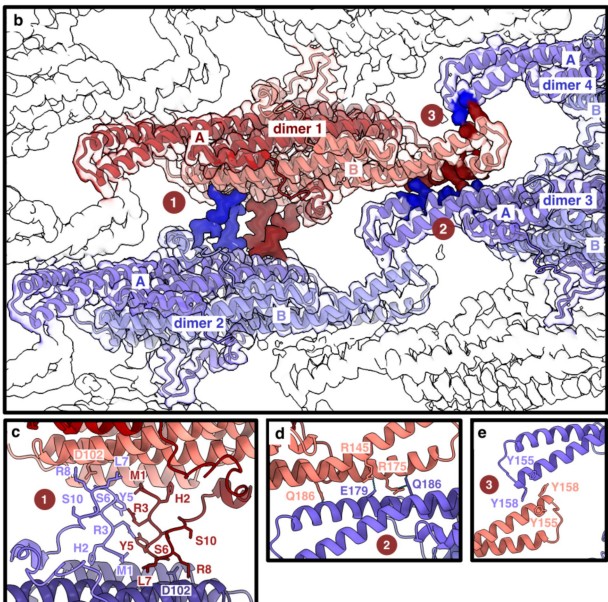

**Extended Data Fig. 4 | Sequence alignment and lattice contact sites.**
**a**, Sequence alignment of *S. cerevisiae* Pil1 and Lsp1 (*MUSCLE*) with domain architecture illustrated (dotted lines indicate unstructured regions in native-source MCC–eisosome model). Violet squares indicate sterol-binding residues, dodger squares indicate PS-binding residues, green squares indicate PI(4,5)P₂-binding residues, grey squares indicate other lipid-binding-pocket residues, red circles indicate residues that form lattice contacts. **b**–**e**, Lattice contact sites between Pil1 dimers. Previous nanometre-resolution helical reconstructions of reconstituted Pil1 and Lsp1 proteins revealed a lattice pattern that could be fitted with the Lsp1 crystal structure, albeit with unaccounted density at the lattice contact sites[33]. Three regions of contact between the central dimer (red/salmon) and its neighbours (blue/light violet) are clear in our structures (**b**). The first site of contact is a short stretch of interactions between the well-folded,

domain-swapped N-terminus (Nt) of monomer A (res1-8) in the central dimer (red) and the equivalent Nt stretch (res1-8) of monomer B in neighbouring dimer 2 (light violet) (**c**), including residue S6 which previously shown to be phosphorylated by Pkh1 and important for eisosome assembly in combination with other phosphorylated residues[78,79]. The remaining two contact sites are localized to the BAR-domain tips, previously shown to be flexible in crystallographic studies[38]. A stretch of electrostatic interactions between residues 171–186 on helix 3, as well as residue 145 on helix 2, of the BAR domain in monomer A of the central dimer (salmon) and the equivalent residues from monomer B of dimer 3 (blue) forms the second contact site (**d**). A hydrophobic interaction between Y155 at the tip of BAR-domain helix 2 on monomer A of the central dimer (salmon) with Y158 on monomer B of dimer 4 (light violet), and vice versa, forms a third contact (**e**).

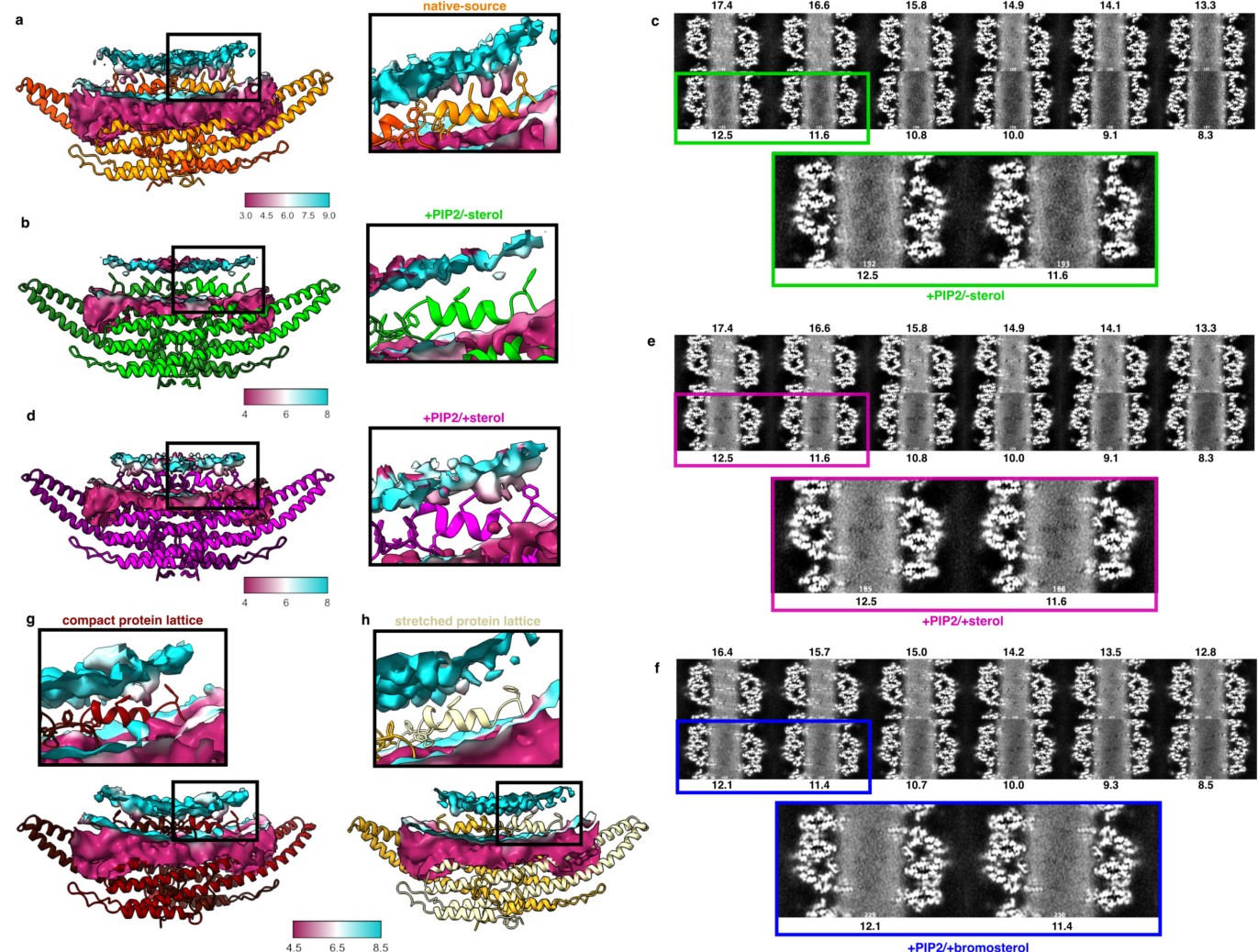

**Extended Data Fig. 5 | Sterol void visualization. a**, Membrane density from native-source samples visualized at high threshold to display an inverted topological surface of the cytosolic leaflet, coloured by local resolution (in angstrom). Position of sterol voids, corresponding to droplet-shaped pockets of higher local resolution (magenta), is shown relative to the residues of the AH within the cytosolic leaflet (inset). **b**, Membrane density from +PI(4,5)P₂/−sterol reconstituted samples visualized at high threshold (as in **a**). Droplet-shaped pockets corresponding to sterol voids are not observed in the +PI(4,5)P₂/−sterol zone map (inset). **c**, Sequence of one-pixel slices through the protein-bound leaflet of +PI(4,5)P₂/−sterol samples. Numbers above or below slices indicate distances in angstrom from the bilayer midplane. Coloured boxes highlight slices -11–12.5 Å from the midplane bilayer and are presented as zoomed inset below. **d**, Membrane density from +PI(4,5)P₂/+sterol reconstituted samples visualized at high threshold (as in **a**,**b**). Position of sterol voids, corresponding

to droplet-shaped pockets of higher local resolution (magenta), is shown relative to the residues of the AH within the cytosolic leaflet. These pockets can be clearly observed in +PI(4,5)P₂/+sterol zone maps (inset). **e**,**f**, Sequence of one-pixel slices through the protein-bound leaflet of +PI(4,5)P₂/+sterol (**e**) and +PI(4,5)P₂/+bromosterol (**f**) reconstituted samples. Numbers above or below slices indicate distances in angstrom from the bilayer midplane. Coloured boxes highlight slices -11–12.5 Å from the midplane bilayer (in which voids are interrupted by density in the +PI(4,5)P₂/+bromosterol eisosomes) and are presented as zoomed inset below. **g**,**h**, Membrane density from native-source compact protein lattice (**g**) and stretched protein lattice (**h**) maps visualized at high threshold to display an inverted topological surface of the cytosolic leaflet, coloured by local resolution (in Å). Sterol voids are visible within the cytosolic leaflet of the compact protein lattice (**g**, inset) but not the stretched protein lattice (**h**, inset) zone maps.

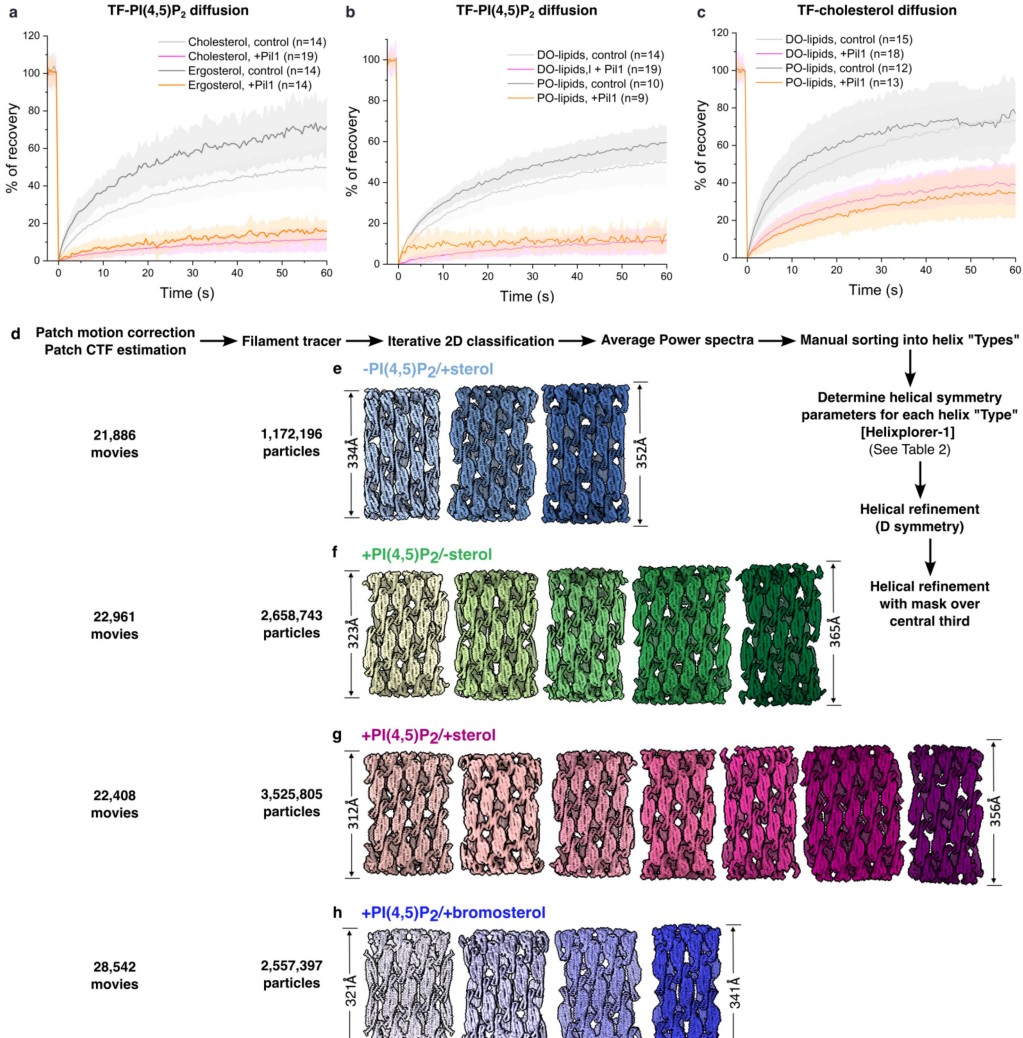

**Extended Data Fig. 6 | Validation and cryo-EM data processing of reconstituted Pil1 filaments with lipid mixtures of known composition.** **a**, FRAP of TF-PI(4,5)P2 lipids in cholesterol- and ergosterol-containing lipid mixtures in the presence or absence of Pil1. **b,c**, FRAP of TF-PI(4,5)P2 (**b**) and TF-cholesterol (**c**) lipids in DO- or PO- lipid mixtures in the presence or absence of Pil1. In **a**–**c**, solid lines indicate a mean of n number of measured nanotubes with standard deviation shown. **d**, Helical reconstruction data processing strategy for reconstituted Pil1 tubules. **e**–**h**, All helical reconstructions from −PI(4,5)P$_2$/+sterol (**e**), +PI(4,5)P$_2$/−sterol (**f**), +PI(4,5)P$_2$/+sterol (**g**) and +PI(4,5)P$_2$/+bromosterol (**h**) reconstituted Pil1 tubules.

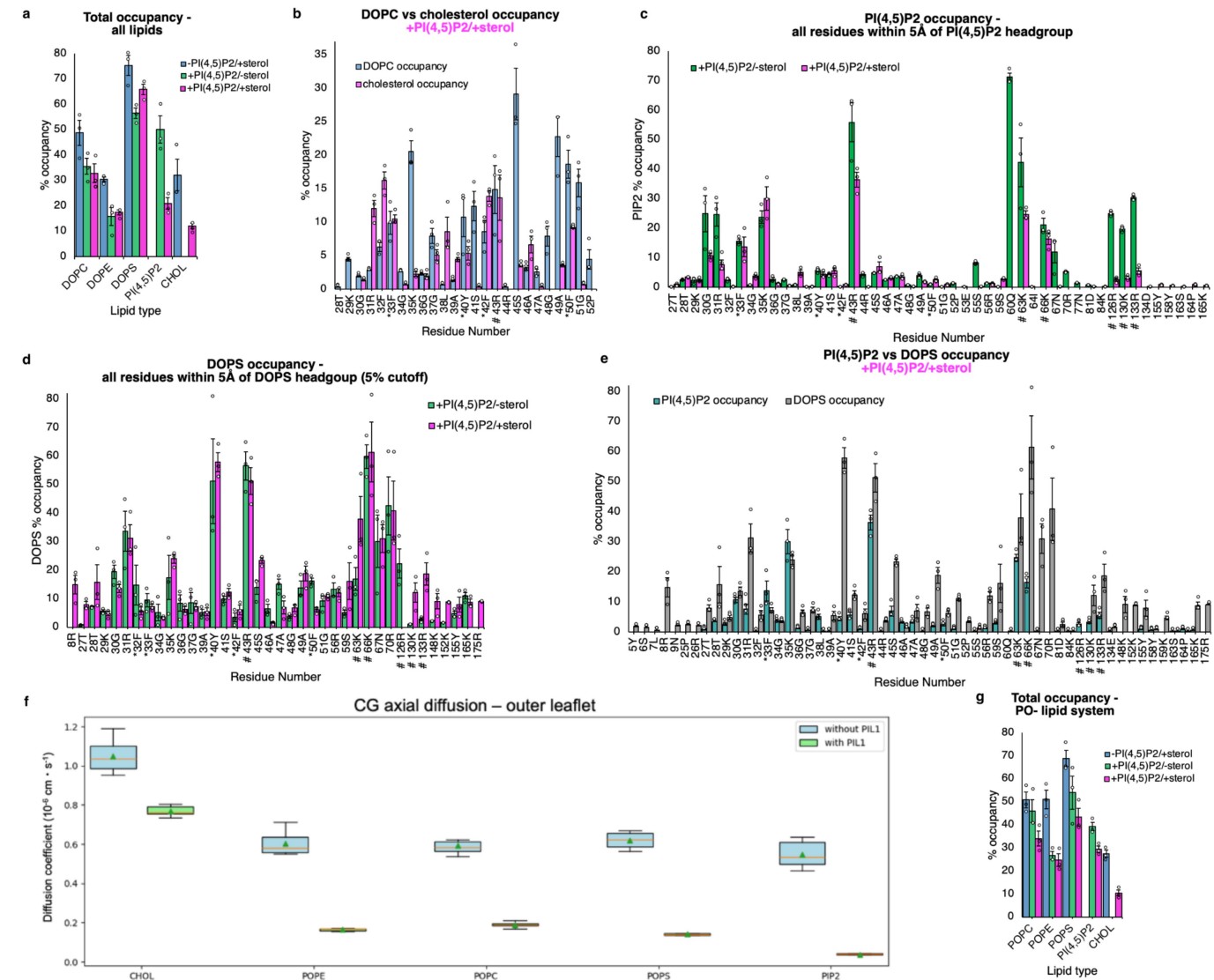

**Extended Data Fig. 7 | CG MD simulations. a**, Total occupancy per lipid for all lipids in each CG MD system. Total occupancy was averaged across the 3 replicas, error bars are s **b**, Comparison of DOPC occupancy and cholesterol occupancy in the +PI(4,5)P₂/+sterol system for AH region residues <5 Å from DOPC and/or sterol headgroup, revealing the specificity of the sterol occupancy over the DOPC occupancy at the bulky side chains of the AH. **c**, PI(4,5)P₂ occupancy reported for all residues <5 Å from PI(4,5)P₂ headgroups. **d**, DOPS lipid occupancy for residues <5 Å from DOPS headgroup with >5% occupancy in CG MD simulations. **e**, Comparison of PI(4,5)P₂ occupancy and DOPS occupancy in the +PI(4,5)P₂/+sterol system for residues <5 Å from PI(4,5)P₂ and/or DOPS headgroup. In **b**–**e**, values represent averages of per-residue occupancy computed along three replicas of 10 μs each, error bars are SEM. **f**, Lipid

diffusion coefficients from CG simulations with PO- lipids. The values reported are the 1D axial diffusion of lipids in the outer leaflet of the tubes, with (green) and without (cyan) the Pil1 coat. Diffusion coefficients were computed from 5 replicas of 2 μs. Box plots elements are defined as follows: Centre line is the median, box limits are 25% to 75% lower and upper quartiles, whiskers extend from the box to the minimum and maximum data points lying within 1.5x interquartile range (IQR), and green triangle indicates the mean value. **g**, Total occupancy per lipid for all lipids in each CG MD system using PO- lipids instead of DO- lipids. * indicates sterol-binding residues and # indicates charged predicted lipid-binding-pocket residues. Total occupancy was averaged across the 3 replicas, error bars are SEM.

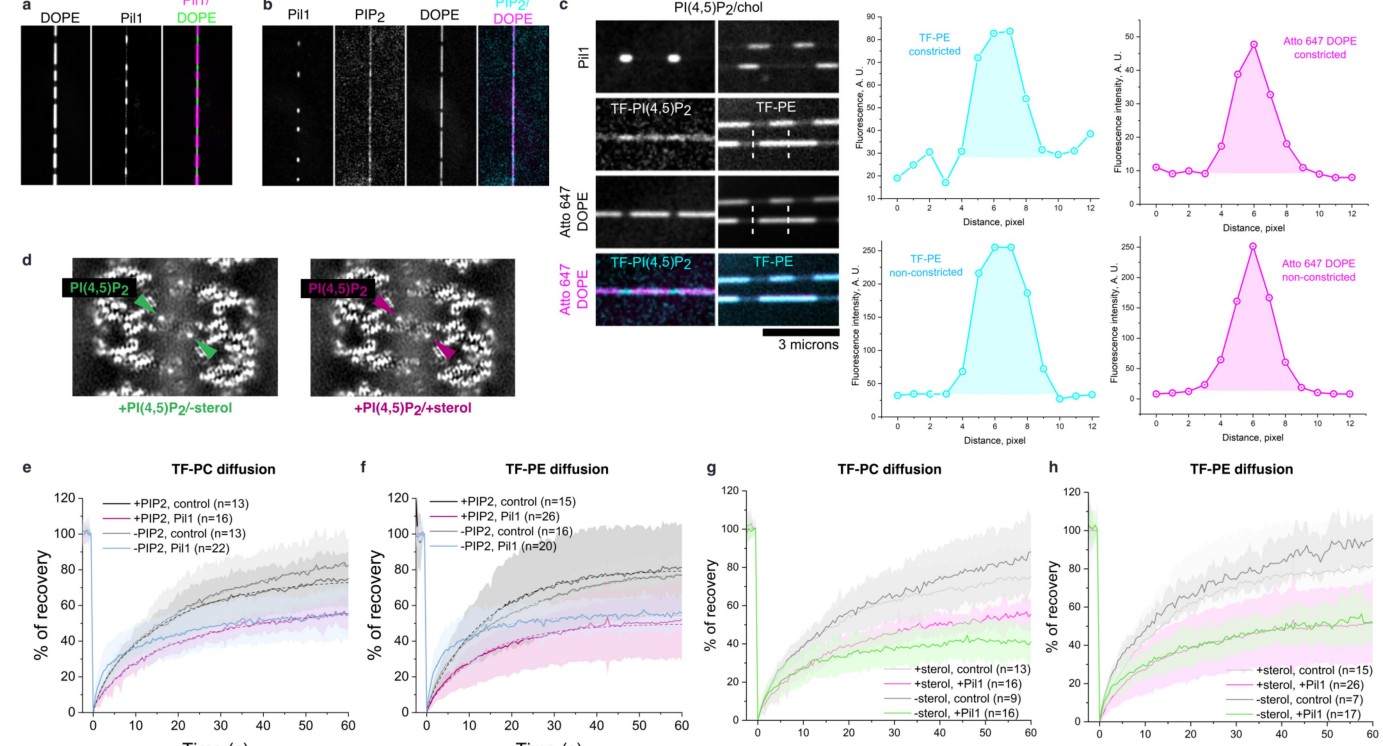

**Extended Data Fig. 8 | Lipid sorting coefficients and +PI(4,5)P₂/−sterol FRAP assays. a**, Example lipid nanotube with DOPE-Atto647N and Pil1−mCherry bound. Constriction of the nanotube reduces the intensity of the labelled lipid fluorescence where Pil1−mCherry is bound. **b**, Example lipid nanotube with Pil1−mCherry, fluorescent TF-PI(4,5)P₂, and fluorescent DOPE-Atto647N. Constriction of the nanotube reduces the intensity of the labelled lipid fluorescence where Pil1−mCherry is bound. Increased sorting of TF-PI(4,5)P2 relative to DOPE-Atto647N is observed at sites of Pil1 nanotube constriction. **c**, Fluorescence plot profiles of the lipid of interest and reference lipid used to extract the integrated fluorescence densities (shading under the curve) to measure lipid sorting coefficients. **d**, AH slice through unsharpened maps of

+PI(4,5)P₂/−sterol (left panel) and +PI(4,5)P₂/+sterol (right panel), with arrows indicating presumed PI(4,5)P₂ density in each reconstruction (green and lilac, respectively). **e,f**, FRAP of TF-PC (**e**) and TF-PE (**f**) in control samples without protein and with Pil1 in −PI(4,5)P₂/+sterol (blue) and +1% PI(4,5)P₂/+sterol (magenta) lipid nanotubes. Solid lines indicate a mean of n number of measured nanotubes with standard deviation shown. Dashed lines indicate the fitted data. **g,h**, FRAP of TF-PC (**g**) and TF-PE (**h**) in control samples without protein and with Pil1 in +1% PI(4,5)P₂/−sterol and +1% PI(4,5)P₂/+sterol lipid nanotubes. Solid lines indicate a mean of n number of measured nanotubes with standard deviation shown.

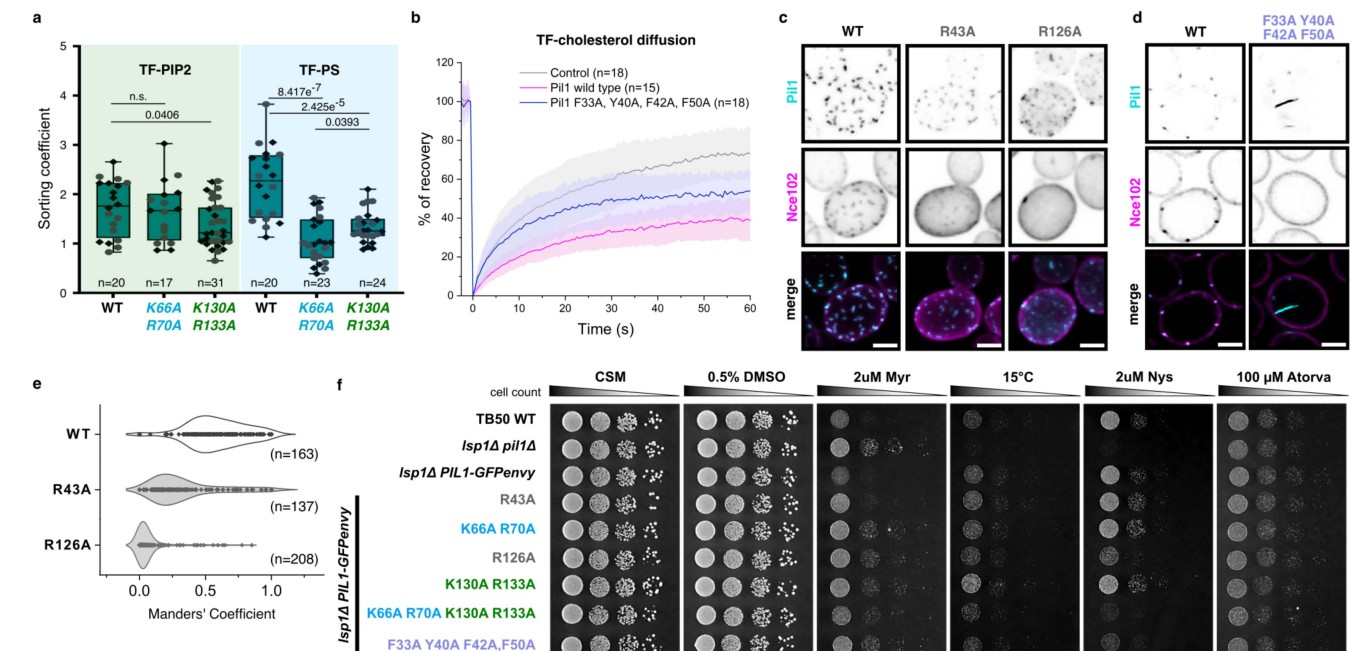

**Extended Data Fig. 9 | In vivo lipid-binding-pocket mutants. a**, Lipid sorting coefficients of TF-PI(4,5)P2 and TF-PS for PS-binding-impaired mutant (pil1-K66A/R70A) and PI(4,5)P2-binding-impaired mutant (pil1-K130A/R133A) in +1% PI(4,5)P2/+sterol lipid nanotubes. A significant decrease in $PI(4,5)P_2$ sorting is observed in the PI(4,5)P2-binding impaired mutant relative to the WT (WT vs K130A p = 0.04058), but not in the PS-binding-impaired mutant (WT vs K66A p = 0.71233, K66A vs K130A p = 0.15098). PS sorting is impaired in both the PS- and the PI(4,5)P2-binding impaired mutants (WT vs K66A p = 8.417e-7, WT vs K130A p = 2.425e-5, K66A vs K130A p = 0.03931). Box indicates interquartile range (IQR) from Q1 (25%) to Q3 (75%) quartiles. Lower and upper whiskers show from Q1 and Q3 quartiles to minimum and maximum data points, respectively. Horizontal line shown inside the box indicates the median [Statistical significance: p-values obtained applying two-sample t-test with all conditions following normal distribution at 0.01 tested by Shapiro–Wilk, Kolmogorov–Smirnov, and Anderson–Darling normality tests. n is the number of independent tested nanotubes, N = 2 for all conditions, being the number of experimental repetitions. Black rhombuses show data points obtained at day 1 and grey circles show data points obtained for day 2]. **b**, FRAP of TF-cholesterol for sterol-binding-impaired mutant (Pil1-F33A/Y40A/F42A/F50A), with and without Pil1 protein in +1% PI(4,5)P2/+sterol lipid nanotubes. The mobile fraction of sterols is increased in the sterol-binding-impaired mutant, confirming the reduction of sterol binding by this mutant. Solid lines indicate a mean of n number of measured nanotubes with standard deviation shown. **c**, Eisosome morphology in *lsp1Δ* yeast expressing Pil1-GFPenvy lipid-binding-pocket mutation variants and Nce102–mScarlet-I (summed stacks). Merge represents summed stacks of Pil1-GFPenvy (cyan) and mScarlet-I (magenta) signals. **d**, Central confocal slice of *lsp1Δ* yeast expressing Pil1-GFPenvy variants and Nce102–Scarlet-I highlighting cytosolic ingression of eisosomes in sterol-binding-impaired mutant Pil1^[F33A/Y40A/F42A/F50A]. **e**, Thresholded fraction of Nce102–mScarlet-I that colocalizes with indicated Pil1-GFPenvy lipid-binding-pocket mutants in single cells (Manders' M1 colocalization coefficient). The shaded area represents the probability for data points of the population to take on this value. **f**, Growth of serial dilutions of *lsp1Δ* cells expressing Pil1-GFPenvy lipid-binding-pocket mutants on Complete Supplement Mixture media (CSM) with the indicated treatments. (DMSO: dimethylsulfoxide, Atorva: atorvastatin, Nys: nystatin, Myr: myriocin). All tagged/mutant proteins in **c–f** are expressed from their endogenous locus. Scale bars, 2 µm.

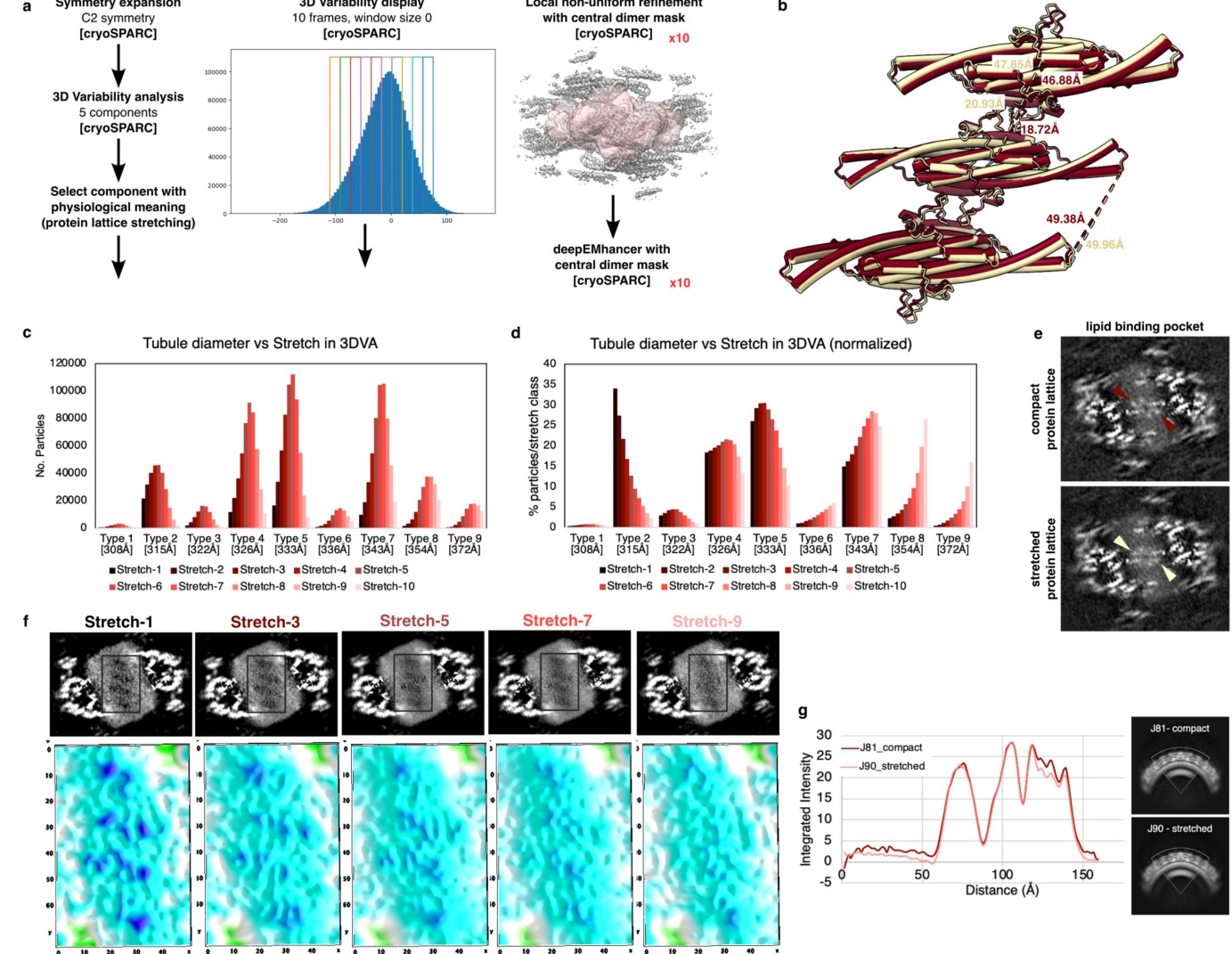

**Extended Data Fig. 10 | 3D variability analysis. a**, 3DVA data processing strategy. **b**, Alignment of most compact protein lattice class (dark red) and most stretched protein lattice class (yellow) comparing measured distances between 3 different regions on neighbouring dimers within the lattice. Differences in distance between dimer regions range from 0.5–2.2 Å. **c**, Number of particles from each helical reconstruction type found in each 3DVA stretch class (Stretch-1=most compact protein lattice, Stretch-10=most stretched protein lattice). **d**, Percent of particles within each 3DVA stretch class (Stretch-1=most compact protein lattice, Stretch-10=most stretched protein lattice) deriving from each helical reconstruction type. **e**, One-pixel slices through unsharpened maps illustrating the lipid-binding pocket and the sterol voids of the most compact protein lattice and most stretched protein lattice 3DVA classes with the lipid-binding-pocket region indicated (most compact protein lattice: red arrows, most stretched protein lattice: yellow arrows). **f**, 3D surface intensity plot of sterol void slice from 5 intermediate classes along the 3DVA component exhibiting protein lattice stretch, ranging from Stretch-1 (compact protein lattice) through Stretch-9 (stretched protein lattice). **g**, Radial angle profile plots of bilayer within most compact protein lattice and most stretched protein lattice classes.

## Extended Data Table 1 | Cryo-EM data collection, refinement and validation statistics

| | #1 Pil1 lattice (native) | #2 Lsp1 lattice (native) | #3 Pil1 lattice (-PIP2/+sterol reconstituted) | #4 Pil1 lattice (+PIP2/-sterol reconstituted) | #5 Pil1 lattice (+PIP2/+sterol reconstituted) | #6 Pil1 lattice (+PIP2/+bromosterol) | #7 Pil1 lattice compact (native) | #8 Pil1 compact dimer with lipids headgroups (native) | #9 Pil1 lattice stretched (native) |
|---|---|---|---|---|---|---|---|---|---|
| | (EMD-18307) | | (EMD-18308) | (EMD-18309) | (EMD-18310) | (EMD-19822) | (EMD-18311) | | (EMD-18312) |
| | (PDB 8QB7) | (PDB 8QB8) | (PDB 8QB9) | (PDB 8QBB) | (PDB 8QBD) | | (PDB 8QBE) | (PDB 8QBF) | (PDB 8QBG) |
| **Data collection and processing** | | | | | | | | | |
| Magnification | 10500x | 10500x | 96000x | 96000x | 96000x | 165000x | 105000x | 105000x | 105000x |
| Voltage (kV) | 300 | 300 | 300 | 300 | 300 | 200 | 300 | 300 | 300 |
| Electron exposure (e–/Å²) | 40.00 | 40.00 | 50 | 50 | 50 | 50 | 40 | 40 | 40 |
| Defocus range (μm) | -0.8 to -1.8 | -0.8 to -1.8 | -0.6 to -1.8 | -0.6 to -1.8 | -0.6 to -1.8 | -0.6 to -1.8 | -0.8 to -1.8 | -0.8 to -1.8 | -0.8 to -1.8 |
| Pixel size (Å) | 1.33 | 1.33 | 0.83 | 0.83 | 0.83 | 0.7121 | 1.327 | 1.327 | 1.327 |
| Symmetry imposed | Symmetry expanded, C1 | Symmetry expanded, C1 | Helical, D1 | Helical, D3 | Helical, D1 | Helical, D1 | Symmetry expanded, C1 | Symmetry expanded, C1 | Symmetry expanded, C1 |
| Rise (Å) | - | - | 5.044 | 14.547 | 5.41 | 5.33 | - | - | - |
| Twist (deg.) | - | - | 136.5 | 83.250 | 133.60 | 133.61 | - | - | - |
| Initial particle images (no.) | 1211972 | 1211972 | 1172196 | 2685743 | 3525805 | 2557397 | 1211972 | 1211972 | 1211972 |
| Final particle images (no.) | 1211972 | 1211972 | 176005 | 85456 | 77414 | 56475 | 63118 | 63118 | 77457 |
| Map resolution (Å) | 3.20 | 3.20 | 3.35 | 3.75 | 3.56 | 3.85 | 3.67 | 3.67 | 3.64 |
| FSC threshold | 0.143 | 0.143 | 0.143 | 0.143 | 0.143 | 0.143 | 0.143 | 0.143 | 0.143 |
| Map resolution range (Å) | 2.972-42.269 | 2.972-42.269 | 1.762-29.519 | 1.754-37.804 | 1.754-46.832 | 1.490-57.484 | 3.302-61.571 | 3.302-61.571 | 3.261-61.394 |
| **Refinement** | | | | | | | | | |
| Initial model used (PDB code) | Pil1 (P53252) AlphaFold2 database | - | - | - | - | | - | - | - |
| Model resolution (Å) | 3.3 | 3.5 | 3.3 | 3.8 | 3.5 | | 3.9 | 3.7 | 3.8 |
| FSC threshold | 0.143 | 0.143 | 0.143 | 0.143 | 0.143 | | 0.143 | 0.143 | 0.143 |
| Map sharpening B factor (Å²) | -106 | -106 | -101.9 | -101.3 | -91.1 | | -99.6 | -99.6 | -94.6 |
| **Model composition** | | | | | | | | | |
| Non-hydrogen atoms | 30058 | 29750 | 27874 | 30394 | 30814 | | 30058 | 4364 | 30058 |
| Protein residues | 3794 | 3794 | 3500 | 3794 | 3794 | | 3794 | 542 | 3794 |
| Ligands | 0 | 0 | 0 | I3P: 14 | IP3:14 P5S:14 | | 0 | IP3:2 SEP:2 | 0 |
| **B factors (Å²)** | | | | | | | | | |
| Protein | 7.58/145.95/75.53 | 40.94/191.65/92.81 | 36.10/208.72/84.74 | 74.76/233.48/116.33 | 61.79/189.51/97.32 | | 73.61/172.32/104.67 | 73.61/172.32/104.84 | 63.31/156.79/96.07 |
| Ligand | - | - | - | 128.10/128.10/128.10 | 98.09/117.01/108.60 | | - | 109.94/109.94/109.94 | - |
| **R.m.s. deviations** | | | | | | | | | |
| Bond lengths (Å) | 0.002 | 0.003 | 0.002 | 0.003 | 0.002 | | 0.003 | 0.004 | 0.004 |
| Bond angles (°) | 0.508 | 0.65 | 0.467 | 0.546 | 0.476 | | 0.67 | 1.002 | 0.699 |
| **Validation** | | | | | | | | | |
| MolProbity score | 1.47 | 1.59 | 1.38 | 1.71 | 1.55 | | 1.63 | 1.75 | 1.67 |
| Clashscore | 8.68 | 11.82 | 6.96 | 7.23 | 6.71 | | 13.07 | 14.73 | 12.9 |
| Poor rotamers (%) | 0.43 | 0 | 0 | 2.56 | 1.71 | | 0.85 | 1.07 | 0 |
| **Ramachandran plot** | | | | | | | | | |
| Favored (%) | 98.51 | 99.26 | 99.19 | 99.26 | 99.63 | | 98.14 | 97.77 | 97.77 |
| Allowed (%) | 1.49 | 0.74 | 0.81 | 0.74 | 0.37 | | 1.86 | 2.23 | 2.23 |
| Disallowed (%) | 0 | 0 | 0 | 0 | 0 | | 0 | 0 | 0 |

# Reporting Summary

## Statistics

For all statistical analyses, confirm that the following items are present in the figure legend, table legend, main text, or Methods section.

| n/a | Confirmed | |
|---|---|---|
| ☐ | ☒ | The exact sample size (*n*) for each experimental group/condition, given as a discrete number and unit of measurement |
| ☐ | ☒ | A statement on whether measurements were taken from distinct samples or whether the same sample was measured repeatedly |
| ☐ | ☒ | The statistical test(s) used AND whether they are one- or two-sided *Only common tests should be described solely by name; describe more complex techniques in the Methods section.* |
| ☐ | ☒ | A description of all covariates tested |
| ☐ | ☒ | A description of any assumptions or corrections, such as tests of normality and adjustment for multiple comparisons |
| ☐ | ☒ | A full description of the statistical parameters including central tendency (e.g. means) or other basic estimates (e.g. regression coefficient) AND variation (e.g. standard deviation) or associated estimates of uncertainty (e.g. confidence intervals) |
| ☐ | ☒ | For null hypothesis testing, the test statistic (e.g. *F*, *t*, *r*) with confidence intervals, effect sizes, degrees of freedom and *P* value noted *Give P values as exact values whenever suitable.* |
| ☒ | ☐ | For Bayesian analysis, information on the choice of priors and Markov chain Monte Carlo settings |
| ☒ | ☐ | For hierarchical and complex designs, identification of the appropriate level for tests and full reporting of outcomes |
| ☒ | ☐ | Estimates of effect sizes (e.g. Cohen's *d*, Pearson's *r*), indicating how they were calculated |

*Our web collection on statistics for biologists contains articles on many of the points above.*

## Software and code

Policy information about availability of computer code

| | |
|---|---|
| Data collection | SerialEM (Nexperion) and EPU 2.14 (ThermoFisher) for EM data collection. BUMPy v1.1 (doi: 10.1021/acs.jctc.8b00765) and GROMACS version 2021.5 (doi.org/10.5281/zenodo.4457591) for CG MD simulations . Slidebook 6.0.22 (Intelligent Imaging) for Spinning disk confocal microscopy experiments and lipid sorting measurements. Visiview v4.4.0.11 (Visitron Systems GmbH) for FRAP experiments. Zeiss Zen 3.3.89.0008, Blue edition for cell imaging. Spectra Manager version 2.14.02 (Jasco) for CD spectrometry. |
| Data analysis | EM Data: cryoSPARC Live v3.2.2, MotionCor2, Gctf v1.06, RELION v2.1.0 and v3.1.3, Helixplorer-1, cryoSPARC v4.1.2 and v4.4.0 for data processing. Chimera v1.16, ChimeraX v1.5 and Fiji v1.54f, for visualization and figure preparation. Coot 0.8.9.2, phenix.elbow (https://phenix-online.org/documentation/reference/elbow.html), deepEMhancer (https://github.com/rsanchezgarc/deepEMhancer) and PHENIX 1.20-4459 for model building and refinement. Lipid sorting assays: Fiji v1.54f, OriginPro 2022 v9.9.0.225 (OriginLab Corp.) and GraphPad Prism 10.1.1 CG MD simulations: GROMACS version 2021.5 (doi.org/10.5281/zenodo.4457591); PyLipID package (doi.org/10.1021/acs.jctc.1c00708); Python version 3; VMD version 1.9.4 (doi: 10.1016/0263-7855(96)00018-5) and Martinize2 (https://github.com/marrink-lab/vermouth-martinize) Mass spec analysis: MaxQuant v1.6.6.0 Fluorescence microscopy analysis: Fiji v1.54f, JACoP BIOP version (https://github.com/BIOP/ijp-jacop-b), Cellpose v2.0, and Origin Pro 2022 v9.9.0.225 CD spectrometry: Spectra Manager Analysis version 2.15.05 (Jasco) and Origin Pro 2022 v9.9.0.225 FRAP analysis: Fiji v1.54f and Origin Pro 2022 v9.9.0.225 |

For manuscripts utilizing custom algorithms or software that are central to the research but not yet described in published literature, software must be made available to editors and reviewers. We strongly encourage code deposition in a community repository (e.g. GitHub). See the Nature Portfolio guidelines for submitting code & software for further information.

## Data

Policy information about availability of data

All manuscripts must include a data availability statement. This statement should provide the following information, where applicable:

- Accession codes, unique identifiers, or web links for publicly available datasets
- A description of any restrictions on data availability
- For clinical datasets or third party data, please ensure that the statement adheres to our policy

Mass spectrometry data were deposited to the ProteomeXchange Consortium via the PRIDE partner repository with the dataset identifier PXD050326.

Sharpened maps used for model refinement and all associated helical maps and deepEMhancer sharpened maps have been deposited in the Electron Microscopy Data Bank (https://www.ebi.ac.uk/emdb/) under the following accession codes: Eisosome native-source (EMD-18307), Pil1 -PIP2/+sterol reconstituted (EMD-18308), Pil1 +PIP2/-sterol reconstituted (EMD-18309), Pil1 +PIP2/+sterol reconstituted (EMD-18310), Pil1 +PIP2/+bromosterol reconstituted (EMD-19822), Eisosome native-source compact (EMD-18311), Eisosome native-source stretched (EMD-18312). Raw micrographs of native-source eisosome samples have been deposited in the Electron Microscopy Public Image Archive (EMPIAR) database under the accession code EMPIAR-12053.

The starting model for building the Pil1 model was acquired from the AlphaFold Protein Structure Database (https://alphafold.ebi.ac.uk/) using the Uniprot accession number P53252 (PIL1_YEAST) and Q12230 (LSP1_YEAST). All models have been deposited in the Protein Data Bank (https://www.rcsb.org/): Pil1 lattice (native-source) (PDB 8QB7), Lsp1 lattice (native-source) (PDB 8QB8), Pil1 lattice (-PIP2/+sterol reconstituted) (PDB 8QB9), Pil1 lattice (+PIP2/-sterol reconstituted) (PDB 8QBB),  Pil1 lattice (+PIP2/+sterol reconstituted) (PDB 8QBD), Pil1 lattice compact (native-source) (PDB 8QBE), Pil1 dimer compact with lipid headgroups (native-source) (PDB 8QBF), and Pil1 lattice stretched (native-source) (PDB 8QBG).

Lipid diffusion, lipid sorting and yeast cell biology data are provided at DOI:10.26037/yareta:ubja4xykqzfjbhcfwmg7sgj2x4. Raw gels are provided in Supplementary data. All other data supporting the findings of this study are provided in this manuscript. Source data are provided with this paper.

## Research involving human participants, their data, or biological material

Policy information about studies with human participants or human data. See also policy information about sex, gender (identity/presentation), and sexual orientation and race, ethnicity and racism.

| | |
|---|---|
| Reporting on sex and gender | n/a |
| Reporting on race, ethnicity, or other socially relevant groupings | n/a |
| Population characteristics | n/a |
| Recruitment | n/a |
| Ethics oversight | n/a |

Note that full information on the approval of the study protocol must also be provided in the manuscript.

# Field-specific reporting

Please select the one below that is the best fit for your research. If you are not sure, read the appropriate sections before making your selection.

☒ Life sciences ☐ Behavioural & social sciences ☐ Ecological, evolutionary & environmental sciences

For a reference copy of the document with all sections, see [nature.com/documents/nr-reporting-summary-flat.pdf](http://nature.com/documents/nr-reporting-summary-flat.pdf)

# Life sciences study design

All studies must disclose on these points even when the disclosure is negative.

| | |
|---|---|
| Sample size | EM data: Sizes of cryoEM datasets were determined by microscope availability, efficiency of image collection strategy, and number of tubules visible per micrograph (dependent on sample). For native structures, manual targeting produced a dataset of 2827 movies, sufficiently large to produce 9 helical maps and merged lattice map of good resolution. For reconstituted structures, large (>20,000 movies) datasets were collected, and final particle number after classification was sufficient to achieve good resolution.<br>CG MD simulations: Simulations of 10us were performed in triplicate, thus producing an aggregated time of 30us per system.  Sample size was chosen to allow membrane equilibration of multicomponent lipid tubules, as in PMID: 36624348.<br>FRAP assays: Sample sizes were chosen based on previous publications in the field (PMID: 24055060 and PMID: 35858336)<br>Lipid sorting assays: Sample size was chosen based on previous publications on lipid and/or protein sorting experiments using membrane in vitro reconstitutions (PMID:19304798, PMID:20160074, and PMID: 31757972).<br>Cellular microscopy and growth assays: The samples sizes were selected based on previous studies with similar methodologies (PMID: 37902009 and PMID: 29976762). |

| | |
|---|---|
| Data exclusions | EM data: Micrographs and particles of poor quality were excluded as part of the process of EM data analysis to reach good resolution.<br>CG MD simulations: residues >5 angstroms in distance from lipid headgroups were not included in the analysis. Lipid headgroup occupancy lower than 5% was excluded for analysis of DOPS occupancy in +PIP2/-sterol vs +PIP2/+sterol system comparisons and in DOPS vs PIP2 occupancy comparison in the +PIP2/+sterol system.<br>FRAP assays: No data was excluded.<br>Lipid sorting assays: No data was excluded.<br>Cellular microscopy: Segmented cells that where intersected by the image borders, and cells that featured Nce102-Scarlet signal below the fixed threshold (thresholded area = 0) for calculating Manders' co-localisation coefficients were excluded. |
| Replication | EM data: For each dataset, at least 3 and up to 9 different helical structures were solved from independent sets of particles within the same dataset. Each helical structure within a sample dataset, while varying in diameter and/or helical parameters, exhibited an identical lattice pattern and similar structural features (e.g. lipid headgroups, sterol voids) in regions of interest for our study.<br>CG MD simulations: Three replicas were performed for each system. In addition, as a control, a single replica was performed for the +PIP2/+sterol system composition without the protein. Results were similar across replicas.<br>FRAP assays: repeated three times and in each repetition at least 3 different nanotubes were measured. Results were similar across replicas.<br>Lipid sorting assays: repeated two times and in each repetition at least 8 different nanotubes were measured. Results were similar across replicas.<br>Cellular microscopy: Performed at least three times, on different days, from fresh cultures, yielding similar results. Data from at least two different days was pooled for each strain for overlap analysis. Results were similar across replicas.<br>Yeast growth assays: Performed at least three times, on different days, from fresh cultures, of which at least once from a different streak out from glycerol stock, yielding similar results. Results were similar across replicas. |
| Randomization | EM data: CryoEM data analysis involves inherent randomization: tubules are randomly oriented within the micrographs, particles are divided into random halfsets and independently processed, and resolution estimates are based on the independently processed half maps.<br>FRAP assays and lipid sorting assays: Randomization was not applied because measuring multiple samples at the same time was not possible. Each sample had to be prepared individually. Samples for each experiment were prepared in a similar manner.<br>Cellular microscopy and growth assays: Yeast strains of known identity were required and used for these assays. |
| Blinding | EM data: Blinding was not possible because samples of known composition were required.<br>CG MD simulations: Blinding was not done because systems of known composition were required.<br>FRAP assays: Blinding was not possible because the difference between samples without protein and with proteins was significant upon protein binding.<br>Lipid sorting assays: Blinding for group allocation or data analysis was not done since this study relies on studying difference lipid behaviors under Pil1 helical scaffolds<br>Cellular microscopy and growth assays: Blinding was not done as it would not affect the outcome of these experiments. |

# Reporting for specific materials, systems and methods

We require information from authors about some types of materials, experimental systems and methods used in many studies. Here, indicate whether each material, system or method listed is relevant to your study. If you are not sure if a list item applies to your research, read the appropriate section before selecting a response.

## Materials & experimental systems

| n/a | Involved in the study |
|---|---|
| ☒ | Antibodies |
| ☒ | Eukaryotic cell lines |
| ☒ | Palaeontology and archaeology |
| ☒ | Animals and other organisms |
| ☒ | Clinical data |
| ☒ | Dual use research of concern |
| ☒ | Plants |

## Methods

| n/a | Involved in the study |
|---|---|
| ☒ | ChIP-seq |
| ☒ | Flow cytometry |
| ☒ | MRI-based neuroimaging |

