## [Peer Review File · Nature]

Manuscript Title: CryoEM architecture of a near-native stretch-sensitive membrane microdomain

Reviewer Comments & Author Rebuttals

Reviewer Reports on the Initial Version:

Referee #1 (Remarks to the Author):

This most impressive structural analysis describes a peripheral protein lattice in a quasi-native state. This lattice, which defines one of the 3 subdomains of the yeast plasma membrane, the eisosome, is formed by a complex of BAR domain-containing proteins. The electron microscopy analysis allows the authors to not only define the protein structural part but also the lipid organization underneath. The data are compelling and demonstrate that the lipids in the cytosolic leaflet are not randomly distributed but, instead concentrate or are excluded, owing to specific or weak interactions with the protein lattice. The three lipids that are affected by this protein-induced organization are PIP2, cholesterol and PS. Furthermore, when the lattice is under mechanical constraint, the associated lipids tend to diffuse away. No doubt that this study will raise the interest of numerous cell biologists and biochemists. Addressing the following points might improve the ms.

#1. Phosphatidylserine and its acyl chain profile. The ability of phosphatidylserine to interact with cholesterol is a very interesting property in the context of the cytosolic leaflet of the plasma membrane, considering the high density of these lipids in this leaflet. However, cholesterol oxidase accessibility to cholesterol, which is a very sensitive assay, show strong variability for the strength of interactions between cholesterol and the different acyl chain species of PtdSer (PMID: 25663704). 18:0-18:1 PtdSer (SOPS) but not other PtdSer species such as 16:0/18:1 (POPS), 18:1/18:1 (DOPS) or 16:0/18:2 (PLPS) shield cholesterol. Interestingly, the acyl chain profile of PS changes along the secretory pathway. PS species with one saturated and one monounsaturated acyl chains become predominant at the plasma membrane at the expense of di monounsaturated species (PMID: 10459010). Yet, in both their reconstitution and in their molecular dynamic simulations, the authors use DOPS, which is a rather rare lipid species in yeast. More generally, the systematic use of C18:1-C18:1 lipids in reconstitution experiments (supplementary Table 3) is at odds with the major species found in yeast, especially at the plasma membrane.

Because lipid organization under the BAR domain lattice is the main finding of this study, it would be more convincing to use lipid mixes that are as close as possible to what is physiologically found. I understand that this request might be overwhelming and I am not asking for a complete revision including numerous new lipid species, but if the authors could repeat a few experiments and simulations with POPS or SOPS, that would be great. Finally, it would be very informative to conduct a lipidomic analysis of the native eisosome purification if the amount of material is compatible with MS/MS sensitivity.

#2. Cholesterol – amphipathic helix interactions. The sentence “multiple examples of sterol/amphipathic helix interactions have been previously demonstrated” is puzzling if not awkward. First, the interaction of amphipathic helices with membranes varies tremendously

depending on the chemistry of the helix; some are sensitive to lipid electrostatic, to lipid unsaturation, or to curvature. However, cholesterol sensitivity does not sound a recurrent theme in amphipathic helix – membrane interaction. For BAR family proteins, it has been found that the presence or the absence of cholesterol does not change their ability to bend membranes (PMID: 32649209) or that cholesterol can decrease it (PMID: 32878944). Second, the single reference given by the authors is not appropriate. The MD simulations they refer to is not of the Osh4 transporter but of the amphipathic helix of phosphatidylinositol 4-phosphate 5-kinase (PIP5K) (PMID: 31402097 = ref 56). For Osh4, the situation is complex, because Osh4 not only interacts with sterol-containing membranes through an amphipathic helix, but also extracts sterol (PMID: 22162133). I would suggest the authors to correct their statement, tone down their general conclusion or provide other examples of clear-cut amphipathic helices interacting with sterols.

That said, I am not questioning the interaction of the protein lattice with sterols, which is one of the most important results of this work (the striking voids seen underneath the amphipathic helices in the native eisosome filaments as well as in the artificial tubes containing sterols, but not in the absence of sterol). One additional clue for a favorable interaction between the Pil1 lattice and sterols is given by FRAP measurements on nanotubes containing the fluorescence sterol Top fluor cholesterol. I wonder whether this molecule is suitable for such measurements. For other lipids, e.g. PIP2, the Topfluor moiety replaces a fraction of the acyl chain, which is fine. This replacement is minor in the case of Top-fluor cholesterol and so the chemistry of the probe is quite different from the chemistry of the lipid of interest. Whether the exact partitioning of Topfluor cholesterol in the membrane underneath the amphipathic helix is as exquisite as that of ergosterol is an open issue. Would it be possible to use the naturally fluorescent sterols DHE or cholestane trienol to see whether they tend to concentrate close to the Pil1 lattice? I acknowledge that these fluorescent analogs are not very stable, so imaging might be tricky, but I think it is worth trying.

#3. The gentle procedure for eisosome isolation involves the use of the detergent CHAPS at a concentration of 0.5 mM, well below its CMC. Interestingly, the chemistry of CHAPS is quite close to sterol, but with additional charged groups. How do the authors exclude that the detergent could contribute to some of the density (the polar head) or lack of density (the void) observed?

#4. General positioning of this study versus the long-standing question of lipid domains. At the end of the manuscript, the authors nicely make a parallel between their model and other protein lipid complexes at the plasma membrane, notably those including some mechanosensitive channels, as well as caveolins. In contrast, I found the reminder to the lipid domain hypothesis in the introduction of the paper less adapted. First, the present study is in fact a most intelligent investigation of a serendipitous observation; “our eisosome filaments are a contamination of our intended target”. The authors plan was not to solve the issue of domain formation, was it? Most importantly, they show that the Pil1 lattice is remarkably robust and can form on various artificial membranes even without some key lipids (- sterol or -PIP2 conditions). They demonstrate a mechanism by which a protein lattice concentrates specific lipids rather than a lipid domain that preexists by virtue of preferential interaction between some lipid classes or species, which would subsequently concentrate proteins. I would suggest that the author remain more elusive in the introduction and not refer to detergent resistant domains as well as liquid order domains, which at the end are very different to what is actually shown here; the idea being to not to create shortcuts in the naive

reader's reasoning.

Minor points

#5. I got sometimes confused by the words 'mobilization' vs 'sequestration' Are they antonyms of synonyms?

#6. Figure 3. One panel should be labelled "B". In addition, the Bar plot shows in the X axis the same labels, whereas in the legend it is written "+1% PI(4,5)P2/-sterol" and "+1% PI(4,5)P2/+sterol". Please clarify

#7. Figure Extended Data 9. It seems to me that the legends for panel E and D have been inverted.

Referee #2 (Remarks to the Author):

General comments

In this manuscript, Kefauver and colleagues aim to understand in molecular detail membrane sculpting function of BAR-domain containing proteins and their ability to organize lipid microdomains in general. To this end, they have chosen one of the best characterized membrane microdomains, the MCC in the yeast plasma membrane, which is scaffolded on the cytosolic side by the eisosome, a hemitubular assembly of BAR domain proteins Pil1 and Lsp1.

For the first time, they isolated eisosomes in a "native-like" form, attached to tubules composed of the yeast plasma membrane, i.e. in a state very close to that in situ. Using cryoelectron microscopy, they solved the structure of these tubular eisosomes with unprecedented resolution, allowing them to identify amino acid residues in the Pil1 sequence that are responsible for binding individual Pil1 dimers to the cylindrical network whose curvature can vary over a previously predicted range depending on membrane tension. Benefiting from the extraordinary resolution they achieved and combining their cryoEM data with molecular dynamics simulations, the authors were additionally able to determine the likely binding sites of specific lipids, phosphatidylinositol 4,5-bisphosphate, phosphatidylserine and ergosterol, to the eisosome. Lipid binding was then characterized in detail on in vitro reconstituted eisosomes that tubulated liposomes prepared from lipid mixtures of different compositions.

Finally, by combining the data describing the flexible structure of the eisosome and the way lipids bind to it, the authors were able to propose a mechanism by which lipids could be released into the surrounding membrane when the BAR domain proteins-composed lattice deforms due to changes in membrane tension. This is something that, for sterols in particular, has been suggested repeatedly before, albeit so far on the basis of only indirect evidence. And, it strongly supports speculation that it could be the release of sterols from membrane microdomains such as MCC/eisosome that underlies the observed rapid, ATP-independent adaptation of the plasma membrane to a range of external stimuli.

I believe that this is why the authors' findings will be of interest to a wide range of readers of different backgrounds and specializations. I am convinced that the data presented in the manuscript are sufficiently robust to justify the conclusions drawn by the authors. The abstract is written clearly

and in a form that makes the manuscript accessible and attractive to a wide readership. Previous works are appropriately and in a balanced way cited throughout the whole text.

Specific comment

It is not explained why cells expressing a TAP-tagged TORC2 subunit Bit61 were used to isolate native-like eisosomes. If the isolation of the eisosome was just a by-product obtained by serendipity, as suggested by the authors' mention of Pil1 protein as a frequent contaminant of the yeast pull-downs, this should be clearly stated in the text.

Minor comments

1. lines 47-8: "The MCC microdomains, also known as eisosomes..." This formulation is misleading and should be rephrased. Many authors favor a clear distinction between the eisosome and the MCC - while the eisosome is a membrane-associated protein complex, the MCC is a membrane microdomain, i.e. a region of the plasma membrane organized and shaped by the eisosome.
2. lines 56, 57: Judging by the references used, some of which rely only to integral membrane proteins accumulated within MCC, I would suggest using "MCC/eisosome" instead of "eisosome" here in the spirit of the previous comment. It is not just a matter of name: when, for example, reference #5 discusses changes in the distribution of Nce102, these are primarily related to the composition of the membrane microdomain, not to the structure of the membrane-associated complex formed by BAR domain proteins.
3. line 78: "Here we have isolated intact membrane microdomain scaffolded in helical tubules..." In this form, the statement seems to lack internal consistency. As the authors themselves state below (lines 109 and following), the intact membrane microdomain takes the form of a furrow, not a tubule. So if tubulated, the membrane microdomain does not appear to be literally intact. Tubulation, however, is a spontaneous reshaping of the MCC microdomain that can be expected after thawing of the ground portions of the membrane during the described sample preparation procedure. So I agree with the authors that the membrane microdomain is as little disrupted as possible under the given conditions. In this sense, I recommend a change of wording here. The realization of the alternative, which the authors also mention (lines 115-18), that the eisosomes were tubulated already *in vivo*, seems rather unlikely. By the way, the first argument in the list lacks reference to the presence of tubular eisosomes in Sur7-family deletion strains. If data from a mutant strain of *Candida albicans* lacking Sur7 is cited here, this argument should be removed from the list because the aberrant invaginations observed in this mutant contain the cell wall and thus do not represent membrane tubules comparable to the structures described in this study.
4. line 120: A wide range of readers is expected in Nature. I'm not sure if everyone has encountered the use of the term "2D classes" in this context. Neither the text here nor the Figure legend is helpful in this regard; a brief explanation would suffice.
5. lines 293-5: A reference to "an additional small density" is missing here. Please add the reference to "Ext Data 9E, bottom panel" at the end of the sentence. Consider adding another arrow to the Fig Ext Data 9E.
6. Figure 1E: The numbers below the scale depicting the color coding of the electrostatic surface prediction should be explained or omitted.
7. Figure 3A: Lipid headgroup density for PS marked in grey is virtually invisible unless the image is greatly magnified. Would it be possible to use another color for it?
8. Figure 3B: i) the letter B marking this image is placed incorrectly; ii) labels on the horizontal axis

look incomplete

9. Fig Ext Data 9: The legends for images D and E are interchanged.

Referee #3 (Remarks to the Author):

Many cell biologists and structural biologists believe that the nano-scale organization of lipids in biological membranes will play an important role in many cell membrane mechanisms. Outside of simulations, however, it been very difficult to study this organization in biologically relevant, protein-bound membranes. In this manuscript, Kefauver and colleagues have studied the organization of proteins and lipids in purified eisosomes – specialized membrane compartments in yeast that play a poorly-described role in responding to membrane stress. They have isolated eisosomes from cells and they have reconstituted eisosome-like structures in vitro from pure components. They have obtained structures of eisosomal proteins bound to membranes by cryoEM, have measured lipid mobility in reconstituted systems, and have performed molecular dynamics simulations of protein-lipid interactions. Based on these experiments they conclude that Pil1 forms a protein lattice on the eisosome membrane, and that the protein organizes and arranges PI(4,5)P2, PS and sterols at the nano-scale within the lipid membrane, thereby altering lipid mobility and presumably the mobility of other membrane protein and lipid components.

These are detailed insights into a specialized yeast membrane domain but their implications are broader. I find this the clearest experimental demonstration that peripheral membrane proteins can organize specific lipids at the nanometer scale (beyond the well-described direct lipid binding), and that this lipid organization can be regulated by changes in the protein (here the stretching of the lattice). In my opinion this is an important step towards understanding the role of lipid organization in cellular function and should be of high and broad interest to cell biologists and structural biologists.

Overall, this is an exciting study that appears to have been well designed and executed. There some weaknesses that should be addressed prior to publication.

1. The described loss of sterol ordering in the stretched classes is based on blurring of the voids in the most extended class. This is, however, not very convincing. Can the attempt at quantification in Extended figure 10 be improved, perhaps including intermediate stretch classes?
2. The sorting data for PS is used to support its assignment as a direct binder, but there is nothing that works as a negative control. Presumably the authors have done sorting analysis for the other top-fluor lipids they have analysed? These should be added to figure 3B and would provide such a negative control. While I agree that this density is most likely to be PS in the simple in vitro system, might there be other candidates in the eisosomes? I think it is too strong language to describe the PS density as a “structural signature”.
3. The evidence that PIP2 is required for amphipathic helix insertion (rather than amphipathic helix immobilization), seems to be based on a difference in average lipid distance distributions of about 0.5Å in the CG MD simulations. This is a very small difference – is it enough to describe as partial

detachment from the membrane? Given that the lattice remains stably membrane associated in the absence of PIP2, and that PIP2 appears to coordinate residues in both the AH and the body of the protein, is it not more likely that it instead helps immobilize the helix? The authors could also add some discussion of how PIP2 may be stabilizing the helix.

4. In general, there is some lack of consistency about where the authors think changes in lipid mobility reflect specific interactions with the protein and where they reflect changes in bilayer organization, eg by the sterols. One way to clarify this would be to present FRAP data for the other top-fluor lipids in the minus sterol conditions, to assess the impact of the ordered sterols on overall lipid mobility.

5. The CG MD simulations suggest that the sterols are positioned via interactions with bulky sidechains in the AH. The in vitro system provides the opportunity to test this directly by repeating the reconstitution with protein containing mutations at specific amino acids in the AH. These experiments would complement the data on the mutated yeast strains. They would provide additional support for the assignment of the voids as sterols.

6. I agree that the voids are likely to represent sterols, but the assignment is not completely conclusive since removing sterols from the bilayer may broadly alter lipid-lipid and lipid-protein interactions. I do not think further experimentation is needed here, but suggest here addition of a sentence or two to make clear that a direct assignment is not easily achievable. Alternatively, the authors could consider lipid cross-linking experiments, or experiments using labelled lipids.

7. In my opinion there should be much more prominent citation and discussion of the Moss et al paper (ref 54), and the Unwin paper (ref 55) both of which address the organization of lipids within membrane bound lipid tubes.

Minor issues:

1. The abstract implies that the lipids could be assigned based on the native eisosome structure, and that this assignment was verified using the invitro reconstitutions and the MD. In fact, convincing assignments cannot be made based on the eisosome structure and require the in vitro reconstitutions. Please edit the abstract appropriately.

2. The “voids” assigned to sterols are illustrated by a section parallel to the lipid plane. The figure needs magnified views of these voids, and also orthogonal views, so that their positions relative to the AH could be more easily interpreted.

3. Have the authors tried other methods to estimate what fraction of protein in the sample is Pil1 and what fraction Lsp1?

4. Other lipid compositions were tested for reconstitution, but only a subset were used for analysis – on what basis were these subsets selected, and can anything be learned from the others?

5. The comment that the extra PIP2 density is not present in the -PIP2 filaments needs to be qualified by the statement that the AH which forms part of the binding pocket is also missing in the structure.

6. 3B panel label is missing and all columns have the same label on the x axis.

7. Remarkably, the authors have used “remarkably” six times in the manuscript.

Referee #4 (Remarks to the Author):

Kefauver et al present structural insights into the organization of proteins associated with eisosomes in yeast. The major claim is that the reported structures represent a 'native' and 'stretch-activated' membrane domain. The basis for this claim is that the tubules analyzed by cryoEM appear in micrographs of a preparation from cells, rather than being reconstituted from isolated proteins. The structures are interesting and similar to those previously observed (and confirmed here) with isolated proteins. There are some potentially interesting speculations about the origin of unusual 'voids' in the membrane near amphipathic helices and possible PIP2 densities, though these are weakly supported and of unclear biological relevance.

This is a frustrating and disappointing paper. In principle, there could have been interesting insights here: the eisosome is poorly understood, membrane scaffolds have interesting organizations, and structural insights into lipid organization are few and far between. Unfortunately, the many far-reaching claims here are presented with little and weak evidence. Several are detailed below, but the most egregious is that these structures are in no way 'native' and there is no evidence for stretch-sensitivity. Much of the data are over-interpreted, making the major conclusions largely unsupported.

Major issues:

1. As explicitly claimed by the title, abstract, and keywords, the core advance of this manuscript is resolution of the structure of a 'native' membrane microdomain. There are several issues with this claim, but the most glaring is that the structures reported are not 'native'. This is clear from the first paragraph of the results, which makes the more reasonable – though still unsupported - claim that the tubules analyzed are 'native-like' or 'near-native'. An obvious way that the structures are not 'native' is that they are derived with detergent. Another is that they are tubes, while eisosomes in situ are not. The speculations connecting these tubes to cellular structures are guesses with no evidence behind them.
2. It is unclear to this reviewer and seemingly to the authors themselves what is meant by native, near-native, and native-like. The former has a commonly used definition that is clearly inappropriate here. The latter have no specific meaning, but it remains unclear what the authors would intend, considering they provide no evidence to support which features of the isolated structures are near-native and in which ways they may be native-like. This also extends to the claim of 'intact' plasma membrane, which is again obviously incorrect under the common definition of 'intact'.
3. There are also myriad issues with claimed "stretch sensitivity", which appears closer to conjecture than data. (1) I do not know what "3D variability analysis" is, there are no references, and minimal explanation in the methods section. It is impossible to critically evaluate this approach based on the information in the manuscript. (2) it is impossible to understand what "spring-like stretching" (line 373) or "changes in shape and size of bound lipid density" (line 375) the reader is meant to observe from SuppVideo3. The legend provides minimal explanation. (3) all isolated tubules are at equilibrium, there is no possibility for mechanical stretch, thus assigning "stretched" and "compressed" to different tubules presupposes something that doesn't exist in this experiment. It is inappropriate to use phrases like "lattice stretching" without any evidence thereof. (4) the morphing between various classes is essentially a guess, it is unclear how much data or manual input goes into

these types of analyses and they appear ripe for over-fitting and over-interpretation

4. It seems odd to base the entire manuscript on a contaminant of an irrelevant pulldown with no detectable signal for the supposed proteins being resolved. Without any biochemical evidence of purity or even protein identity in the 'native' samples, the potential for artifacts (eg from contaminants) or over-fitting seems high. For example, the various tubules used to suggest 'stretch' may instead have different isoforms, interacting proteins, post-translational modifications, etc etc etc.

5. The evidence of sterol-dependence of the 'voids' is weak. Fig2F-H appears to show much stronger differences between the 'native' and the two reconstituted samples - why are reconstituted structures more poorly resolved than the isolated? The presence of 'voids' is not quantified in any way, preventing meaningful comparison. It is a trivial observation that the hydrophobic face of a membrane-embedded AH is interacting more frequently with sterol than are the charged residues on the other side. This is not evidence of sterol recruitment, only of those residues being in a lipid environment. A more interesting analysis would be whether sterol is enriched relative to PC.

6. Fig 2K purports to show that 60-70% of cholesterol is immobilized on the minute timescale in the +Pil1+PIP2 condition. This is physically implausible: e.g. the MD simulations would likely reveal that all lipids are completely mobile on the usec timescale. Same issue extends to the FRAP experiments in Fig 3. There are also issues with interpretations of the FRAP data: immobile fraction and diffusion rate are two independent parameters in FRAP experiments and it is unclear which the authors refer to when they claim that "the membrane is less diffusive".

7. Fig 3B is confusingly presented. First, its not easy to understand what this experiment is. Why is there no lipid under the protein in ExtFig9A? Second, the x-axis appears to have all the same labels, so compared conditions arent clear. What do the n refer to, number of nanotubes? If so, those should not be used as independent samples.

8. The logic of the myriocin result is unclear. How does it make sense that mutants with a defect in lipid binding are more myriocin resistant? More generally, the physiological experiments are underwhelming, with few (if any) meaningful defects reported despite mutating quite a few of the residues supposedly important for lipid interactions. Along that line, none of the mutants described in Fig 4 should be called "lipid binding impaired" because there is no evidence provided about their lipid binding. The authors obviously know this, correctly referring to them as "predicted to affect binding" on line 323.

Other issues:

- a. Sentence starting on line 53 is oddly phrased: the function described does not sound mysterious, but rather quite clear.
- b. Some citations are a bit sloppy for this level of journal: eg Shimshick & McConnell say nothing about the plasma membrane
- c. The novelty and impact of the manuscript are framed with respect to membrane microdomains. However, eisosomes are in most ways entirely unlike the controversial PM domains referenced: they are stable, easily visualized, scaffolded by specific proteins, etc. Rather, they are effectively peripheral membrane protein assemblies, like ESCRTs, caveolae, clathrin cages. Structures of many of these have been reported. Thus, framing the novelty around lipid microdomains is misleading.
- d. And why were the reconstitutions and simulations done with cholesterol rather than ergosterol?

Author Rebuttals to Initial Comments:

Referees' comments:

Referee #1 (Remarks to the Author):

Expertise #1: Membrane biophysics

This most impressive structural analysis describes a peripheral protein lattice in a quasi-native state. This lattice, which defines one of the 3 subdomains of the yeast plasma membrane, the eisosome, is formed by a complex of BAR domain-containing proteins. The electron microscopy analysis allows the authors to not only define the protein structural part but also the lipid organization underneath. The data are compelling and demonstrate that the lipids in the cytosolic leaflet are not randomly distributed but, instead concentrate or are excluded, owing to specific or weak interactions with the protein lattice. The three lipids that are affected by this protein-induced organization are PIP₂, cholesterol and PS. Furthermore, when the lattice is under mechanical constraint, the associated lipids tend to diffuse away. No doubt that this study will raise the interest of numerous cell biologists and biochemists. Addressing the following points might improve the ms.

We thank the reviewer for their positive feedback. We hope that this updated version of the manuscript, especially our improved resolution of the bound PS lipid in the reconstituted structures and our new reconstituted structures with bromosterols will address these concerns.

#1. Phosphatidylserine and its acyl chain profile. The ability of phosphatidylserine to interact with cholesterol is a very interesting property in the context of the cytosolic leaflet of the plasma membrane, considering the high density of these lipids in this leaflet. However, cholesterol oxidase accessibility to cholesterol, which is a very sensitive assay, show strong variability for the strength of interactions between cholesterol and the different acyl chain species of PtdSer (PMID: 25663704). 18:0-18:1 PtdSer (SOPS) but not other PtdSer species such as 16:0/18:1 (POPS), 18:1/18:1 (DOPS) or 16:0/18:2 (PLPS) shield cholesterol. Interestingly, the acyl chain profile of PS changes along the secretory pathway. PS species with one saturated and one monounsaturated acyl chains become predominant at the plasma membrane at the expense of di monounsaturated species (PMID: 10459010). Yet, in both their reconstitution and in their molecular dynamic simulations, the authors use DOPS, which is a rather rare lipid species in yeast. More generally, the systematic use of C18:1-C18:1 lipids in reconstitution experiments (supplementary Table 3) is at odds with the major species found in yeast, especially at the plasma membrane.

We thank the reviewer for their comments. We chose to use DO- lipids despite their low frequency in the yeast plasma membrane because their melting temperature is -18°C. PO- or saturated lipids are more prone to phase changes during cooling. We wanted our lipid mixtures to be in a completely disordered state at the start, as our main goal at the outset of the reconstitutions was to check the effects of including PI(4,5)P₂ and cholesterol on the lipid “voids” in cryo-EM structures. The coordination of PS in the +PIP₂/+sterol reconstitutions was a surprise discovery for us, though we did observe a similar density in the native-source samples after we did 3D variability analysis in the compact lattice state.

Because lipid organization under the BAR domain lattice is the main finding of this study, it would be more convincing to use lipid mixes that are as close as possible to what is physiologically found. I understand that this request might be overwhelming and I am not asking for a complete revision including numerous new lipid

species, but if the authors could repeat a few experiments and simulations with POPS or SOPS, that would be great.

We thank the reviewer for this suggestion. We repeated FRAP experiments and MD simulations in the +PIP2/+sterol conditions with a base POPC:POPE:POPS lipids. For both assays, trends remained broadly similar. These data are included in Ext Data Figs 7F and 8J-K and Supplementary Table 5.

We also went back to our +PIP2/+sterol structures and reprocessed them with the newly implemented non-uniform refinement for helical reconstruction in cryoSPARC v4.4.0. Using this procedure, we were able to improve local resolution in the lipid binding pocket, enabling the visualization of partial tail chains of the bound DOPS molecule. While our resolution is not sufficient to unequivocally place the sn1 and sn2 tails, we can see that while one acyl tail is ordered to C2, the other stabilized in a bent position by interactions with residues R43, R70, and perhaps K66 of Pil1 up to C10 (including where the 9,10- double bond is located in the sn2 tail of DO-, PO-, and SO- lipids). While of course this reconstitution was made with DOPS, this orientation allows for the possibility that PS lipid stabilization would likely be similar for DOPS, POPS, and SOPS lipids. These data are now included in Fig 3A and Ext Data Fig 2F and described in the main text as follows:

“In the “+PI(4,5)P₂/+sterol” samples, we were surprised to observe an additional lipid density stabilized between the AH and the PI(4,5)P₂ headgroup. This density accommodates a phosphatidylserine (PS) lipid (DOPS is present in our lipid mixtures), with a large splay in the acyl tails. One acyl tail is visible up to C2 and the other is stabilized up to C10, including the double bond at the 9,10- position, and bent, coordinated by residues R43, K66, and R70 (Fig 3A, third panel, and Extended Data Fig 2F).” (line 246-251)

“In the presence of both PI(4,5)P₂ and sterols, specific stable interactions between Pil1 and PS occur, including its acyl tails, suggesting the PS acyl tail profile could play a role in these interactions.” (line 378-381)

Finally, it would be very informative to conduct a lipidomic analysis of the native eisosome purification if the amount of material is compatible with MS/MS sensitivity.

We had also hoped we could do this analysis; however, as shown in the negative stain EM images of our preps in Rebuttal Fig 1, our native-source samples contain a large number of lipidic contaminants relative to the amount of tubules that we can isolate. These contaminants would most likely overwhelm the results of a lipidomic analysis, making them difficult to interpret. This problem is compounded by the low amount of material we can produce with our native-source preps.

Rebuttal Fig 1. Lipidic contaminants in native-source preps. Negative stain EM montage illustrating frequency of tubules (green arrows) relative to lipidic contaminants (folded micelle shapes with dark staining).

#2. Cholesterol – amphipathic helix interactions. The sentence “multiple examples of sterol/amphipathic helix interactions have been previously demonstrated” is puzzling if not awkward. First, the interaction of amphipathic helices with membranes varies tremendously depending on the chemistry of the helix; some are sensitive to lipid electrostatic, to lipid unsaturation, or to curvature. However, cholesterol sensitivity does not sound a recurrent theme in amphipathic helix – membrane interaction. For BAR family proteins, it has been

found that the presence or the absence of cholesterol does not change their ability to bend membranes (PMID: 32649209) or that cholesterol can decrease it (PMID: 32878944). Second, the single reference given by the authors is not appropriate. The MD simulations they refer to is not of the Osh4 transporter but of the amphipathic helix of phosphatidylinositol 4-phosphate 5-kinase (PIP5K) (PMID: 31402097 = ref 56). For Osh4, the situation is complex, because Osh4 not only interacts with sterol-containing membranes through an amphipathic helix, but also extracts sterol (PMID: 22162133). I would suggest the authors to correct their statement, tone down their general conclusion or provide other examples of clear-cut amphipathic helices interacting with sterols.

We agree that the wording of this sentence is awkward. We have changed it and added additional references to avoid the implication that cholesterol-AH interactions are a recurring theme, but rather specify that a few choice examples in the literature provide interesting parallels to our observations: “*Examples of sterol/AH interactions have been previously observed [Daum, et al., 2016, J Struct Biol; Martyna, et al., 2020, J Phys Chem B; Rahman, et al., 2022, J Mol Biol].*” (line 127-128)

We also thank the reviewer for pointing out our error with regard to the MD simulations in the Nishimura, *et al.* reference; indeed, the MD simulations utilize the AH peptide of PIP5K and the manuscript text has been corrected: “*Notably, MD simulations with the AH of PIP5K predict that cholesterol fills packing defects near aromatic side chains and wedges between the acyl chains of poly-unsaturated PI(4)P [Nishimura, et al., 2019, Mol Cell].*” (line 128-130). In the corrected form, we think including this reference is quite useful for interpreting our results, as we observe the sterol voids adjacent to bulky side chains just as they do in their simulations with PIP5K AH.

That said, I am not questioning the interaction of the protein lattice with sterols, which is one of the most important results of this work (the striking voids seen underneath the amphipathic helices in the native eisosome filaments as well as in the artificial tubes containing sterols, but not in the absence of sterol). One additional clue for a favorable interaction between the Pil1 lattice and sterols is given by FRAP measurements on nanotubes containing the fluorescence sterol Top fluor cholesterol. I wonder whether this molecule is suitable for such measurements. For other lipids, e.g. PIP2, the Topfluor moiety replaces a fraction of the acyl chain, which is fine. This replacement is minor in the case of Top-fluor cholesterol and so the chemistry of the probe is quite different from the chemistry of the lipid of interest. Whether the exact partitioning of Topfluor cholesterol in the membrane underneath the amphipathic helix is as exquisite as that of ergosterol is an open issue. Would it be possible to use the naturally fluorescent sterols DHE or cholestane trienol to see whether they tend to concentrate close to the Pil1 lattice? I acknowledge that these fluorescent analogs are not very stable, so imaging might be tricky, but I think it is worth trying.

We thank the reviewer for this suggestion. Owing to its challenging fluorescence properties including a very low quantum yield as well as excitation and emission maxima of 324 nm and 370 nm, respectively, DHE is a very challenging molecule for imaging, and our attempts were not successful.

However, we were able to reconstitute Pil1 tubules in the presence of brominated sterols [Moss, *et al.*, 2023, NSMB] and solve structures of these tubules to $\sim 3.9\text{\AA}$ resolution. Using ergosterol as the starting product, we were able to brominate this molecule across the double bond at the 7,8- position, adding extra density at approximately the midpoint of the sterol ring structure.

When we compare our reconstituted structures of +PIP2/+bromo-sterol lipid mixture with the +PIP2/-sterol and the +PIP2/+sterol reconstituted structures using the one-pixel parallel slice visualization strategy, we can see that the bromo-sterol structures resemble the +PIP2/+sterol structures in that their amphipathic helices are well-resolved, and the voids can be observed at the plane of the amphipathic helix, as well as starting at the depth of $\sim 8\text{\AA}$ from the bilayer midplane, corresponding approximately to the predicted location of C17 of

cholesterol within the bilayer in MD simulations [Smondryev&Berkowitz, 1999, Biophys J (PMID: 10512828)]. However, in the bromo-sterol structures, in the slices ranging from $\sim 11-12\text{\AA}$ distance from the bilayer midplane, which would correspond quite well with C7(8) where the bromination is located on the bromo-sterol molecule, the voids are “filled”. We think that this result strongly supports the idea that the voids represent individual (or perhaps a few) sterol molecules, while nicely controlling for the incorporation of our altered sterol molecules (amphipathic helix remains well resolved, and void pattern is retained, except at the depth of the bromination). These data have been included in Ext Data Fig 5E-F and the following text has been added to the manuscript to describe this result:

“To directly test whether the voids represent sterol molecules, we reconstituted and solved structures of Pil1 tubules with lipid mixtures containing sterols that were brominated at the 7,8- position of the steroid ring (bromosterol) to add density to these molecules that can be observed in cryoEM [Moss, 2023, NSMB] (Extended Data Figs 3E and 6E and Supplementary Data 2&3). In these +PI(4,5)P₂/+bromosterol reconstituted structures, the AHs are well-resolved, similar to the +PI(4,5)P₂/+sterol structures, suggesting that the bromosterols behave similarly to cholesterol in these structures (Extended Data Fig 5E-F). Furthermore, the voids can be observed at both the plane of the AH and starting at the depth of $\sim 8\text{\AA}$ from the bilayer midplane, corresponding approximately to the predicted location of C17 of cholesterol within the bilayer in MD simulations⁴⁸. However, in the +PI(4,5)P₂/+bromosterol structures, in the slices ranging from $\sim 11-12.5\text{\AA}$ distance from the bilayer midplane, which would correspond well with the bromination at C7(8) on the bromo-sterol molecule, the voids are interrupted by density, strongly suggesting the brominated sterols are localized to the voids, and that the void pattern represents stabilized sterol molecules (Ext Data Fig 5E-F).” (line 189-202)

#3. The gentle procedure for eisosome isolation involves the use of the detergent CHAPS at a concentration of 0.5 mM, well below its CMC. Interestingly, the chemistry of CHAPS is quite close to sterol, but with additional charged groups. How do the authors exclude that the detergent could contribute to some of the density (the polar head) or lack of density (the void) observed?

We acknowledge that the inclusion of CHAPS (even at 0.1-0.2x CMC) in the buffers for the native-source eisosome purification could affect the appearance of the sterol voids in these samples. We did attempt to purify

Rebuttal Fig 2. Detergent-free native-source eisosome preps. Two exemplar raw micrographs from screening data on native-source eisosomes isolated without detergent. Clear filaments are visible, but also many lipidic contaminants. The majority of images from this screening data were contaminants; images with filaments represented <5% of the images collected.

native-source eisosome filaments from yeast in the absence of detergent, and we were still able to obtain clear images of filaments, demonstrating that the inclusion of detergent is not necessary for the formation of the tubules. However, in the absence of detergent, the preps contained many more lipidic contaminants and the images we could collect were overwhelmed by these contaminants precluding the collection of a dataset sufficiently large for structural studies. (see Rebuttal Fig 2)

To exclude the possibility that CHAPS detergent was responsible for the voids, CHAPS was not used during the purification or reconstitution of recombinant Pil1 for structural studies. During purification of these proteins, 1% Triton-X (a non-ionic polyethylene glycol derivative) was used to lyse cells and, after a HisTrap step and a Sepharose size exclusion step, the final dialysis buffer contained 20 mM HEPES, pH 7.4, 150 mM KoAc, and 2 mM MgAc. Pil1 protein from the same prep was used for reconstitution of the +PIP2/-sterol and the +PIP2/+sterol tubules. The voids were absent in the +PIP2/-sterol sample and present in the +PIP2/+sterol sample in equivalent positions to those in the native-source sample. Furthermore, the sterol dwell positions predicted by the MD simulations correspond well with the void positions observed in both the native-source and reconstituted +PIP2/+sterol sample. Our new bromosterols results also strongly support the notion that the voids indeed represent sterol molecules.

As to whether CHAPS contributes to the polar headgroups, these reconstituted samples with no CHAPS also showed clear density for a PI(4,5)P₂ headgroup, at sufficient resolution to unequivocally assign the headgroup identity and position. The well-resolved density in the reconstituted samples is interacting with identical side chains to the native-source samples, so we can conclude that the density we observe in the native-source samples is most likely a PI(4,5)P₂ headgroup.

#4. General positioning of this study versus the long-standing question of lipid domains. At the end of the manuscript, the authors nicely make a parallel between their model and other protein lipid complexes at the plasma membrane, notably those including some mechanosensitive channels, as well as caveolins. In contrast, I found the reminder to the lipid domain hypothesis in the introduction of the paper less adapted. First, the present study is in fact a most intelligent investigation of a serendipitous observation; “our eisosome filaments are a contamination of our intended target”. The authors plan was not to solve the issue of domain formation, was it?

We thank the reviewer for their enthusiasm. Indeed, it was not our plan. We now state explicitly that the eisosome tubules were isolated serendipitously.

Most importantly, they show that the Pil1 lattice is remarkably robust and can form on various artificial membranes even without some key lipids (- sterol or -PIP2 conditions). They demonstrate a mechanism by which a protein lattice concentrates specific lipids rather than a lipid domain that preexists by virtue of preferential interaction between some lipid classes or species, which would subsequently concentrate proteins. I would suggest that the author remain more elusive in the introduction and not refer to detergent resistant domains as well as liquid order domains, which at the end are very different to what is actually shown here; the idea being to not to create shortcuts in the naive reader's reasoning.

We thank the reviewer for this insight. We had chosen this framing since MCC/eisosomes [Zahumensky&Malinsky, 2019, *Biomolecules* (PMID: 31349700); Lanze, *et al.*, 2020, *Microbiol and Mol Biol Rev* (PMID: 32938742)], as well as caveolae and t-tubules [Reeves, *et al.*, 2012, *Adv Exp Med Biol* (PMID: 22411310); Parton, 2018, *Ann Rev Cell and Dev Biol* (PMID: 22411310); Russell, *et al.*, 2017, *Cardiovasc Diabetol* (PMID: 29202762)] which share many of the eisosome properties (high stability, protein coat, implicated in mechanical signaling), are referred to throughout the literature as membrane microdomains and prime examples of membrane compartmentalization. While we were originally leaning towards the idea that our data adds additional weight to the theory that protein-mediated lipid organization underpins the general microdomain

behavior [Lu&Fairn, 2018, Crit Rev Biochem Mol Biol (PMID: 29457544); Levental, *et al.*, 2020, Trends in Cell Biology (PMID: 32302547)], we recognize that the MCC/eisosomes are quite different from the more transient and amorphous detergent-insoluble cholesterol/sphingolipid microdomains at the center of the controversy. As such, we have re-written the first paragraph, preserving the membrane compartmentalization context, but de-emphasizing references to the more controversial detergent resistant domains.

“Membrane compartmentalization enables the spatiotemporal control of a variety of signaling events at the plasma membrane. Although the biological evidence for membrane compartmentalization is overwhelming [Honigmann&Pralle, 2016, J Mol Biol; Lu&Fairn, 2018, Crit Rev Biochem Mol Biol; Shi, et al., 2018, Cell], the determinants and the physical structure of the lipid organization within the membrane remain controversial. This is because almost all tools used to study membrane lipids also risk perturbing their behavior within the membrane context [Levental, et al., 2020, Trends in Cell Biol; Sezgin, et al., 2017, Nat Rev Mol Cell Biol].” (lines 35-39).

Minor points

#5. I got sometimes confused by the words ‘mobilization’ vs ‘sequestration’ Are they antonyms or synonyms?

Mobilization and sequestration should be antonyms. We refer to the lipid coordination behavior of the Pil1 protein lattice as “sequestration” of the lipids. When the protein lattice is stretched, destabilizing the lipid binding pocket, we observe the density contributed by the lipid headgroups as blurred/lower resolution. We interpret this as more dynamic behavior of the lipids which we call “mobilization”.

#6. Figure 3. One panel should be labelled “B”. In addition, the Bar plot shows in the X axis the same labels, whereas in the legend it is written “+1% PI(4,5)P2/-sterol” and “+1% PI(4,5)P2/+sterol”. Please clarify. Thank you for pointing this out. The panel label has been added, and the X axis labels have been corrected. They were accidentally cut off during figure conversion.

#7. Figure Extended Data 9. It seems to me that the legends for panel E and D have been inverted. This has been corrected, though these panels are now relocated to Ext Data Fig 8. Thank you.

Referee #2 (Remarks to the Author):

Expertise #2: Eisosomes

General comments

In this manuscript, Kefauver and colleagues aim to understand in molecular detail membrane sculpting function of BAR-domain containing proteins and their ability to organize lipid microdomains in general. To this end, they have chosen one of the best characterized membrane microdomains, the MCC in the yeast plasma membrane, which is scaffolded on the cytosolic side by the eisosome, a hemitubular assembly of BAR domain proteins Pil1 and Lsp1.

For the first time, they isolated eisosomes in a “native-like” form, attached to tubules composed of the yeast plasma membrane, i.e. in a state very close to that in situ. Using cryoelectron microscopy, they solved the structure of these tubular eisosomes with unprecedented resolution, allowing them to identify amino acid residues in the Pil1 sequence that are responsible for binding individual Pil1 dimers to the cylindrical network whose curvature can vary over a previously predicted range depending on membrane tension. Benefiting from the extraordinary resolution they achieved and combining their cryoEM data with molecular dynamics simulations, the authors were additionally able to determine the likely binding sites of specific lipids, phosphatidylinositol 4,5-bisphosphate, phosphatidylserine and ergosterol, to the eisosome. Lipid binding was then characterized in detail on in vitro reconstituted eisosomes that tubulated liposomes prepared from lipid

mixtures of different compositions.

Finally, by combining the data describing the flexible structure of the eisosome and the way lipids bind to it, the authors were able to propose a mechanism by which lipids could be released into the surrounding membrane when the BAR domain proteins-composed lattice deforms due to changes in membrane tension. This is something that, for sterols in particular, has been suggested repeatedly before, albeit so far on the basis of only indirect evidence. And, it strongly supports speculation that it could be the release of sterols from membrane microdomains such as MCC/eisosome that underlies the observed rapid, ATP-independent adaptation of the plasma membrane to a range of external stimuli.

I believe that this is why the authors' findings will be of interest to a wide range of readers of different backgrounds and specializations. I am convinced that the data presented in the manuscript are sufficiently robust to justify the conclusions drawn by the authors. The abstract is written clearly and in a form that makes the manuscript accessible and attractive to a wide readership. Previous works are appropriately and in a balanced way cited throughout the whole text.

We thank the reviewer for their comments. We hope that our updates to the manuscript text and references and the improvement in the presentation of the figures will address their concerns.

Specific comment

It is not explained why cells expressing a TAP-tagged TORC2 subunit Bit61 were used to isolate native-like eisosomes. If the isolation of the eisosome was just a by-product obtained by serendipity, as suggested by the authors' mention of Pil1 protein as a frequent contaminant of the yeast pull-downs, this should be clearly stated in the text.

We thank the reviewer for this suggestion. We now state explicitly that the eisosome tubules were isolated serendipitously. (line 88)

Minor comments

1. lines 47-8: "The MCC microdomains, also known as eisosomes..." This formulation is misleading and should be rephrased. Many authors favor a clear distinction between the eisosome and the MCC - while the eisosome is a membrane-associated protein complex, the MCC is a membrane microdomain, i.e. a region of the plasma membrane organized and shaped by the eisosome.

We apologize for the lack of precision in our terminology. This sentence has been re-worded as follows to distinguish between the MCC membrane microdomain and the eisosome protein scaffold:

"The MCC microdomains are randomly distributed membrane furrows, about 300nm long and 50 nm deep, scaffolded by a protein coat composed of the Bin/amphiphysin/Rvs (BAR) domain family protein Pil1 and its paralog Lsp1, known as the eisosome [Malinsky, et al, 2010; Douglas&Konopka, 2014; Malinsky&Opekarova, 2016]." (line 44-47)

2. lines 56, 57: Judging by the references used, some of which rely only to integral membrane proteins accumulated within MCC, I would suggest using "MCC/eisosome" instead of "eisosome" here in the spirit of the previous comment. It is not just a matter of name: when, for example, reference #5 discusses changes in the distribution of Nce102, these are primarily related to the composition of the membrane microdomain, not to the structure of the membrane-associated complex formed by BAR domain proteins.

Throughout the text, when referring to the microdomain aspect (as opposed to the protein scaffold aspect) of the eisosomes, we have changed the terminology to "MCC/eisosomes". For all reconstituted structures, we use the term eisosome.

3. line 78: “Here we have isolated intact membrane microdomain scaffolded in helical tubules...” In this form, the statement seems to lack internal consistency. As the authors themselves state below (lines 109 and following), the intact membrane microdomain takes the form of a furrow, not a tubule. So if tubulated, the membrane microdomain does not appear to be literally intact. Tubulation, however, is a spontaneous reshaping of the MCC microdomain that can be expected after thawing of the ground portions of the membrane during the described sample preparation procedure. So I agree with the authors that the membrane microdomain is as little disrupted as possible under the given conditions. In this sense, I recommend a change of wording here.

We agree that we must clarify that we are working with isolated tubules and not *in situ* structures of the MCC/eisosome furrows, but we still want to make a distinction between our tubules which have been isolated untagged from their native source and the ones which we and others have reconstituted *in vitro* with heterologously-expressed protein and lipid mixtures. We have chosen to change the designation to “native-source” (when referring to the eisosomes) or “near-native” (when referring to the MCC microdomain) throughout the manuscript to capture this complexity.

The realization of the alternative, which the authors also mention (lines 115-18), that the eisosomes were tubulated already *in vivo*, seems rather unlikely. By the way, the first argument in the list lacks reference to the presence of tubular eisosomes in Sur7-family deletion strains. If data from a mutant strain of *Candida albicans* lacking Sur7 is cited here, this argument should be removed from the list because the aberrant invaginations observed in this mutant contain the cell wall and thus do not represent membrane tubules comparable to the structures described in this study.

We apologize for the lack of clarity in reference placement; here the reference for the Sur7 deletion strains should be the preprint from Haase, *et al.*, now published in EMBO reports [2023 (PMID: 37902009)]. The authors of this report provide convincing evidence of eisosome tubulation *in vivo* in *S. cerevisiae* strains with Sur7-family deletion using STED microscopy and freeze-fracture electron microscopy. In this final published version of the manuscript, they also show similar tubulation upon the deletion of the PI(4,5)P₂ phosphatases Inp51 and Inp52, overexpression of the PI(4,5)P₂ kinase Mss4, and treatment with the lipophilic compound palmitoylcarnitine using STED microscopy and propose a model in which Sur7 tetraspannin proteins function to prevent tubulation by the Pil1/Lsp1 BAR domain proteins.

Combining this study with the observations of *in vivo* tubulation upon Pil1 overexpression observed in Kabeche, *et al.*, 2011 [PMID: 21900489] and Kabeche, *et al.*, 2015 [PMID: 26359496], we think our speculation that we may be isolating a subset of pre-tubulated eisosomes from our Bit61-TAP strains, which could very well have a minor defect plasma membrane homeostasis, is plausible. We also think that the observation that eisosome tubulation occurs under certain circumstances *in vivo* lends support to the idea that we are observing eisosomes in a truly native-like state.

4. line 120: A wide range of readers is expected in Nature. I'm not sure if everyone has encountered the use of the term “2D classes” in this context. Neither the text here nor the Figure legend is helpful in this regard; a brief explanation would suffice.

Thank you for pointing this out. We have changed the terminology to “2D class averages” to align with the terms used in other papers in Nature journals. We have also added the following explanation to the methods section: “2D classification was run iteratively in RELION 2.1 to sort particles into clean sets of similar diameter and helical arrangement.” (line 511--512)

5. lines 293-5: A reference to “an additional small density” is missing here. Please add the reference to “Ext Data 9E, bottom panel” at the end of the sentence. Consider adding another arrow to the Fig Ext Data 9E.

Thank you for catching this. With our improved resolution at the PS binding site in this version of the manuscript, we think that the PS molecule is now better illustrated in the main Fig 3 and Ext Data Fig 2F and have eliminated this call to the Ext Data Fig 9E (which is now Ext Data Fig 8D with the new figure numbering).

6. Figure 1E: The numbers below the scale depicting the color coding of the electrostatic surface prediction should be explained or omitted.

The following text has been added the relocated Ext Data Fig 1E legend to clarify: “*Electrostatic surface prediction of Pil1 model with potentials ranging from -10 kcal* $\text{mol}^{-1}\text{e}^{-1}$ (red) to +10 kcal* $\text{mol}^{-1}\text{e}^{-1}$ (blue).*”

7. Figure 3A: Lipid headgroup density for PS marked in grey is virtually invisible unless the image is greatly magnified. Would it be possible to use another color for it?

Thank you for this suggestion. We have changed the color of PS molecule to “dodger blue” in the figures to increase its visibility.

8. Figure 3B: i) the letter B marking this image is placed incorrectly; ii) labels on the horizontal axis look incomplete.

Thank you for pointing this out. The panel label has been added, and the X axis labels have been corrected. They were accidentally cut off during figure conversion.

9. Fig Ext Data 9: The legends for images D and E are interchanged.

This has been corrected. Thank you for pointing it out. We note that the figure numbering was changed and in the new version of manuscript these panels are now in Ext Data Fig 8.

Referee #3 (Remarks to the Author):

Expertise #3: Cryo-EM, membranes

Many cell biologists and structural biologists believe that the nano-scale organization of lipids in biological membranes will play an important role in many cell membrane mechanisms. Outside of simulations, however, it been very difficult to study this organization in biologically relevant, protein-bound membranes. In this manuscript, Kefauver and colleagues have studied the organization of proteins and lipids in purified eisosomes – specialized membrane compartments in yeast that play a poorly-described role in responding to membrane stress. They have isolated eisosomes from cells and they have reconstituted eisosome-like structures in vitro from pure components. They have obtained structures of eisosomal proteins bound to membranes by cryoEM, have measured lipid mobility in reconstituted systems, and have performed molecular dynamics simulations of protein-lipid interactions. Based on these experiments they conclude that Pil1 forms a protein lattice on the eisosome membrane, and that the protein organizes and arranges PI(4,5)P₂, PS and sterols at the nano-scale within the lipid membrane, thereby altering lipid mobility and presumably the mobility of other membrane protein and lipid components.

These are detailed insights into a specialized yeast membrane domain but their implications are broader. I find this the clearest experimental demonstration that peripheral membrane proteins can organize specific lipids at the nanometer scale (beyond the well-described direct lipid binding), and that this lipid organization can be regulated by changes in the protein (here the stretching of the lattice). In my opinion this is an important step towards understanding the role of lipid organization in cellular function and should be of high and broad interest to cell biologists and structural biologists.

Overall, this is an exciting study that appears to have been well designed and executed. There some weaknesses

that should be addressed prior to publication.

We thank the reviewer for their positive feedback. We hope that this updated version of the manuscript, especially the CD spectra assay for understanding the folding of the Pil1 AH, our improved resolution of the bound PS lipid in the reconstituted structures, and our new reconstituted structures with bromosterols, as well as improved presentation of the figures will address these concerns.

1. The described loss of sterol ordering in the stretched classes is based on blurring of the voids in the most extended class. This is, however, not very convincing. Can the attempt at quantification in Extended figure 10 be improved, perhaps including intermediate stretch classes?

We thank the reviewer for this suggestion. To illustrate the gradual change in void intensity in the intermediate structures derived from the stretching component of the 3DVA, we have added panel Ext Data Fig 10F. We also used an additional 3D visualization strategy of the voids for the most compact and most stretched classes to illustrate the density within the membrane in Ext Data Fig 5G-H. By raising the threshold of the map until the membrane appears solid, applying local resolution values to the surface, and using zone maps to provide a “cut out” window into the inside of the membrane density adjacent to the amphipathic helix, we produce an inverted view of the topological features within the membrane density. We can see blobby pockets of higher resolution appearing near the amphipathic helices in the compact protein lattice class; these are absent in the stretched protein lattice class. Supplementary videos 5-6 to illustrate the phenomenon are also provided.

2. The sorting data for PS is used to support its assignment as a direct binder, but there is nothing that works as a negative control. Presumably the authors have done sorting analysis for the other top-fluor lipids they have analysed? These should be added to figure 3B and would provide such a negative control.

Thank you for pointing this out. We have now included TopFluor-PE in addition to the TopFluor-PC control that was previously shown in the Ext Data Fig 8, that indeed show no significant sorting in the +PIP2/+sterol conditions. These data are now shown in Fig 3B.

We also went back to our +PIP2/+sterol structures and reprocessed them with the newly implemented non-uniform refinement for helical reconstruction in cryoSPARC v4.4.0. Using this procedure, we were able to improve local resolution in the lipid binding pocket, enabling the visualization of the headgroup, glycerophosphate, and partial tail chains of the bound DOPS molecule. We believe that this improved resolution significantly bolsters our claim that the extra density is indeed a DOPS molecule. These data are included in Fig 3A and Ext Data Fig 2F and described in the text as follows:

“In the “+PI(4,5)P₂/+sterol” samples, we were surprised to observe an additional lipid density stabilized between the AH and the PI(4,5)P₂ headgroup. This density accommodates a phosphatidylserine (PS) lipid (DOPS is present in our lipid mixtures), with a large splay in the acyl tails. One acyl tail is visible up to C2 and the other is stabilized up to C10, including the double bond at the 9,10- position, and bent, coordinated by residues R43, K66, and R70 (Fig 3A, third panel, and Extended Data Fig 2F).” (line 246-251)

While I agree that this density is most likely to be PS in the simple in vitro system, might there be other candidates in the eisosomes? I think it is too strong language to describe the PS density as a “structural signature”.

We agree that it is important to acknowledge that in the native system, there is much more lipid diversity and the extra density observed in the binding pocket of the “compact lattice” map could be, for instance, a minor species of charged lipid. We have added the following text to our description of this density in the native-source samples to account for that possibility:

“...smaller elongated density interacting with residue K66, which we had previously assigned as a PS headgroup in the “+PI(4,5)P₂+sterol” reconstituted samples, though its identity in the native plasma membrane cannot be definitively assigned due to the complexity of its lipid composition relative to our reconstituted tubules.” (line 346-350)

3. The evidence that PIP2 is required for amphipathic helix insertion (rather than amphipathic helix immobilization), seems to be based on a difference in average lipid distance distributions of about 0.5Å in the CG MD simulations. This is a very small difference – is it enough to describe as partial detachment from the membrane? Given that the lattice remains stably membrane associated in the absence of PIP2, and that PIP2 appears to coordinate residues in both the AH and the body of the protein, is it not more likely that it instead helps immobilize the helix? The authors could also add some discussion of how PIP2 may be stabilizing the helix.

We agree that this is a small difference and have decided to remove this data. Instead, to answer this question, we performed circular dichroism experiments with a synthesized Pil1 amphipathic helix peptide and small liposomes. These assays show that in the absence of PI(4,5)P₂, the amphipathic helix does not fold. In contrast, in the presence of 10% PI(4,5)P₂, we observe a clear helical folding of the peptide. These observations support our conclusion that the amphipathic helix does not insert into the bilayer in the absence of PI(4,5)P₂. We have included these data in Fig 2E.

4. In general, there is some lack of consistency about where the authors think changes in lipid mobility reflect specific interactions with the protein and where they reflect changes in bilayer organization, eg by the sterols. One way to clarify this would be to present FRAP data for the other top-fluor lipids in the minus sterol conditions, to assess the impact of the ordered sterols on overall lipid mobility.

Thank you for this suggestion. We have repeated the FRAP assay in the +PIP2/-sterol conditions for TopFluor-PC and TopFluor-PE and see no significant differences relative to the +PIP2/+sterol condition (See Ext Data Fig 8G-H). This suggests that it is the PIP2 and the protein lattice itself, not the presence of sterols that slows the diffusion rate of the bulk lipids and creates the microdomain conditions.

While it is clear that protein and lipids work cooperatively to form a stable lipid domain (e.g. PIP2 specific interactions with R126, K130, R133; sterols interactions with bulky side chains on amphipathic helix; specific DOPS interactions with R70, K66 in the presence of sterol; amphipathic helix stabilized in the presence of PIP2), our data best support a model in which conformational changes in the protein (i.e. lattice stretching, transmitted to the amphipathic helix) produce the changes in lipid mobility we observe in the maps derived from 3D variability analysis.

To clarify these points, we have added the following to the main text:

“To investigate microdomain formation by the Pil1 lattice, we used FRAP assays with TF-PC and -PE and found that a significant portion of each of these lipids is immobilized in the presence of Pil1 (Extended Data Fig 8E-F). In the presence of both Pil1 and PI(4,5)P₂, while the immobile fraction remains similar, the dynamics of the mobile lipid fraction are decreased for TF-PC and -PE lipids (Extended Data Fig 8E-F and Supplementary Table 5). However, the immobile fraction and the dynamics of the mobile lipid fraction for TF-PC and -PE are broadly similar in the presence or absence of cholesterol (Extended Data Fig 8G-H), suggesting that it is the protein lattice itself along with the binding of PI(4,5)P₂ (and the resulting AH insertion/stabilization) that slows lipid dynamics in the membrane microdomain.” (line 261-269)

5. The CG MD simulations suggest that the sterols are positioned via interactions with bulky sidechains in the

AH. The in vitro system provides the opportunity to test this directly by repeating the reconstitution with protein containing mutations at specific amino acids in the AH. These experiments would complement the data on the mutated yeast strains. They would provide additional support for the assignment of the voids as sterols.

We thank the reviewer for this suggestion. We have expressed and purified Pil1 with 4 sterol-interacting residues mutated into alanines (Pil1^{F33A/Y40A/F42A/F50A}) and tested fluorescence recovery of TopFluor-sterol in FRAP assays with this mutant. Compared to the wild type Pil1, the immobile fraction as well as the halftime of the mobile fraction of TopFluor-sterol in nanotubes with the mutant is decreased, supporting the role of these bulky side chains in the AH in coordinating sterols. This data is presented in Ext Data Fig 9B.

6. I agree that the voids are likely to represent sterols, but the assignment is not completely conclusive since removing sterols from the bilayer may broadly alter lipid-lipid and lipid-protein interactions. I do not think further experimentation is needed here, but suggest here addition of a sentence or two to make clear that a direct assignment is not easily achievable. Alternatively, the authors could consider lipid cross-linking experiments, or experiments using labelled lipids.

To address this ambiguity, we were able to reconstitute Pil1 tubules in the presence of brominated sterols [Moss, *et al.*, 2023, NSMB (PMID: 36624348)] and solve structures of these tubules to ~3.9Å resolution. Using ergosterol as the starting product, we were able to brominate across the double bond at the 7,8- position, adding extra density to the molecule at approximately the midpoint of the sterol ring structure.

When we compare our reconstituted structures of +PIP2/+bromosterol lipid mixture with the +PIP2/-sterol and the +PIP2/+sterol reconstituted structures using the one-pixel parallel slice visualization strategy, we can see that the bromo-sterol structures resemble the +PIP2/+sterol structures in that their amphipathic helices are well-resolved, and the voids can be observed at the plane of the amphipathic helix, as well as starting at the depth of ~8Å from the bilayer midplane, corresponding approximately to the predicted location of C17 of cholesterol within the bilayer in MD simulations [Smodyrev&Berkowitz, 1999, Biophys J (PMID: 10512828)]. However, in the bromosterol structures, in the slices ranging from ~11-12Å distance from the bilayer midplane, which would correspond quite well with C7(8) where the bromination is located on the bromosterol molecule, the voids are “filled”. We think that this result strongly supports the idea that the voids represent individual (or perhaps a few) sterol molecules, while nicely controlling for the incorporation of our altered sterol molecules (amphipathic helix remains well resolved, and void pattern is retained, except at the depth of the bromination). These data have been included in Ext Data Fig 5E-F and the following text has been added to the manuscript to describe this result:

“To directly test whether the voids represent sterol molecules, we reconstituted and solved structures of Pil1 tubules with lipid mixtures containing sterols that were brominated at the 7,8- position of the steroid ring (bromosterol) to add density to these molecules that can be observed in cryoEM [Moss, 2023, NSMB] (Extended Data Figs 3E and 6E and Supplementary Data 2&3). In these +PI(4,5)P₂/+bromosterol reconstituted structures, the AHs are well-resolved, similar to the +PI(4,5)P₂/+sterol structures, suggesting that the bromosterols behave similarly to cholesterol in these structures (Extended Data Fig 5E-F). Furthermore, the voids can be observed at both the plane of the AH and starting at the depth of ~8Å from the bilayer midplane, corresponding approximately to the predicted location of C17 of cholesterol within the bilayer in MD simulations⁴⁸. However, in the +PI(4,5)P₂/+bromosterol structures, in the slices ranging from ~11-12.5Å distance from the bilayer midplane, which would correspond well with the bromination at C7(8) on the bromo-sterol molecule, the voids are interrupted by density, strongly suggesting the brominated sterols are localized to the voids, and that the void pattern represents stabilized sterol molecules (Ext Data Fig 5E-F).” (line 189-202)

7. In my opinion there should be much more prominent citation and discussion of the Moss et al paper (ref 54), and the Unwin paper (ref 55) both of which address the organization of lipids within membrane bound lipid tubes.

We agree with the reviewer, as these references guided our interpretations of the membrane features we observed in the cryoEM data. We have added a new paragraph to the intro to highlight the power of cryoEM to study membranes and lipid-protein interactions.

“Cryo-electron microscopy (cryoEM) is an emerging tool for the label-free study of membranes and protein-lipid interactions [Sharma, et al., 2023, Emerg, Top Life Sci; Levental&Lyman, 2023, Nat Rev Mol Cell Biol; Kinnun, et al.2023, Biophys J]. Beyond the wealth of data coming from new structures of transmembrane proteins with bound lipids [Levental&Lyman, 2023, Nat Rev Mol Cell Biol], recent studies have highlighted the potential of cryoEM for investigating lipids within the membrane context. Variations in membrane thickness mediated by lipid composition and/or lipid-protein interactions have been observed by cryoEM in liposomes in vitro [Heberle, et al., 2020, PNAS; Cornell, et al., 2020, PNAS], in reconstituted protein-lipid assemblies [Azad, et al., 2023, NSMB], and in in situ systems [Fischer, et al., 2018, PLOS Biol]. Moreover, perturbations in bulk membrane density mediated by lipid-protein interactions provide compelling examples of how this technique can be used to study lipids in molecular detail within their context [Moss, et al., 2023, NSMB; Unwin, 2022, PNAS]” (line 60-68).

Minor issues:

1. The abstract implies that the lipids could be assigned based on the native eisosome structure, and that this assignment was verified using the invitro reconstitutions and the MD. In fact, convincing assignments cannot be made based on the eisosome structure and require the in vitro reconstitutions. Please edit the abstract appropriately.

Thank you for pointing this out. The abstract text has been re-worded as follows: *“Our structures revealed striking organization of membrane lipids and, using in vitro reconstitutions and molecular dynamics simulations, we confirmed the positioning of individual PI(4,5)P₂, phosphatidylserine, and sterol molecules sequestered beneath the Pil1/Lsp1 coat.”* (line 25-28)

2. The “voids” assigned to sterols are illustrated by a section parallel to the lipid plane. The figure needs magnified views of these voids, and also orthogonal views, so that their positions relative to the AH could be more easily interpreted.

We have magnified the voids for the reconstituted tubules as well in Fig 2G-H. To improve the visualization of the voids relative to the amphipathic helix, we have added Ext Data Fig 10A-C,G-H. These images were produced as described above to address Major issue #1 with the compact and stretched protein lattice maps. We have also included Supplementary videos 1, 5, and 6 to illustrate the void positions in 3D.

3. Have the authors tried other methods to estimate what fraction of protein in the sample is Pil1 and what fraction Lsp1?

To estimate the ratio of Pil1 to Lsp1, we combined the total peptide intensity from each gel slice and found an average intensity ratio of 3.1:1 Pil1 to Lsp1 (See Ext Data Fig 1D), supporting our decision to model our data as Pil1. We have included the mass spectrometry analysis of the native-source eisosome purification as supplementary data and the following text in the new Supplementary methods: *“Analysis of the mass spectrometry data of our native-source preparations yielded an average intensity ratio of 3.1:1 Pil1:Lsp1 peptides (Extended Data Fig 1D and Supplementary Data 1)”*

4. Other lipid compositions were tested for reconstitution, but only a subset were used for analysis – on what basis were these subsets selected, and can anything be learned from the others?

The selection of the subset of lipid compositions was based on our ability to observe tubulation that was sufficiently robust for studies on cryoEM grids. The variations in conditions we tried were combinations of the following variables: 1) +/- DOPE, 2) 0.5% vs 2% vs 10% PI(4,5)P2, 3) brain PI(4,5)P2 vs 18:1 PI(4,5)P2, 4) cholesterol vs ergosterol, 5) combining PI(4,5)P2 with 15% vs 30% cholesterol. The vast majority of the conditions tested yielded some amount of tubulation (except DOPC:DOPS alone), but we found that the most robust tubulation occurred in mixtures containing DOPE, 10% brain PI(4,5)P2, and cholesterol. We have decided to be conservative in our interpretations of whether these conditions have some kind of physiological relevance.

We were surprised that ergosterol gave us slightly worse tubulation since it is the main sterol species in yeast (they do not produce cholesterol). Nevertheless, we were pleased that the cholesterol recapitulated the void pattern we observed in the native-source eisosomes, which almost certainly contain ergosterol, and not cholesterol. This suggests to us that the mechanism of sterol coordination by the eisosome proteins is likely conserved across species. We have now noted all of these observations in the new Supplementary methods.

5. The comment that the extra PIP2 density is not present in the -PIP2 filaments needs to be qualified by the statement that the AH which forms part of the binding pocket is also missing in the structure.

In light of our new CD spectra data that show the AH is folded in the presence of PIP2, but not without it, unless the reviewer strongly objects, we'd rather not make this qualification. The purpose of this experiment was to confirm the headgroup density we observed in the native source samples is PIP2. Consistently, in the -PIP2/+sterol samples, we see no density in this pocket and in the +PIP2/-sterol and +PIP2/+sterol samples, we see clear density at a resolution that is sufficiently high to assign it as a PIP2 headgroup.

6. 3B panel label is missing and all columns have the same label on the x axis.

Thank you. The panel label has been added, and the x-axis labels have been corrected. They were accidentally cut off during figure conversion.

7. Remarkably, the authors have used “remarkably” six times in the manuscript.

We thank the reviewer for pointing this out. The use of the word “remarkably” has been reigned in.

Referee #4 (Remarks to the Author):

Expertise #4: Membrane biology, lipid and protein interactions

Kefauver et al present structural insights into the organization of proteins associated with eisosomes in yeast. The major claim is that the reported structures represent a ‘native’ and ‘stretch-activated’ membrane domain. The basis for this claim is that the tubules analyzed by cryoEM appear in micrographs of a preparation from cells, rather than being reconstituted from isolated proteins. The structures are interesting and similar to those previously observed (and confirmed here) with isolated proteins. There are some potentially interesting speculations about the origin of unusual ‘voids’ in the membrane near amphipathic helices and possible PIP2 densities, though these are weakly supported and of unclear biological relevance.

This is a frustrating and disappointing paper. In principle, there could have been interesting insights here: the eisosome is poorly understood, membrane scaffolds have interesting organizations, and structural insights into lipid organization are few and far between. Unfortunately, the many far-reaching claims here are presented with little and weak evidence. Several are detailed below, but the most egregious is that these structures are in

no way 'native' and there is no evidence for stretch-sensitivity. Much of the data are over-interpreted, making the major conclusions largely unsupported.

Understanding lipid organization at the plasma membrane is a technically challenging topic, leading to many controversies in the field, which is exemplified in the divergent comments of the reviewers. Nevertheless, we believe our data add new and valuable insight into this discussion.

- 1) We argue that because the MCC/eisosomes are isolated from their native source without overexpression or tagging, we present here samples that retain organized plasma membrane lipids that form the basis of this study and provide unique access to a previously unattainable goal: visualizing lipidic features of the plasma membrane at the resolution achievable by helical reconstruction and single-particle cryoEM. Thus, there is value in emphasizing this aspect of how we obtained the MCC/eisosomes presented here. To prevent confusion with *in situ* structures, we have changed our terminology to "native-source" or "near-native" to reflect the importance of the retention of yeast plasma membrane in our samples, and the novel observations that has enabled.
- 2) We hope that our new explanation of the power of 3D variability analysis to reveal and visualize continuous heterogeneity in cryoEM data and our clarification of the type of stretch we observe (specifically, stretch in the protein lattice that is capable of transmitting movement to the amphipathic helix and thus destabilization of our observed lipid binding sites, bolstered by our observations of structural alterations in the bound lipids) will convince the reviewer that our evidence for stretch sensitivity is substantial and provides new mechanistic insights into this important question.

Major issues:

1. As explicitly claimed by the title, abstract, and keywords, the core advance of this manuscript is resolution of the structure of a 'native' membrane microdomain. There are several issues with this claim, but the most glaring is that the structures reported are not 'native'. This is clear from the first paragraph of the results, which makes the more reasonable – though still unsupported - claim that the tubules analyzed are 'native-like' or 'near-native'. An obvious way that the structures are not 'native' is that they are derived with detergent. Another is that they are tubes, while eisosomes *in situ* are not. The speculations connecting these tubes to cellular structures are guesses with no evidence behind them.

As described above, we believe that it is important to emphasize that these samples are isolated from their native source without overexpression or tagging and differentiate them from reconstituted samples using lipid compositions of our choosing that have been solved by us here and by others in the past. The point we want to make is that they retain plasma membrane lipids that form the basis of this study. Thus, we are changing our word usage to "native-source" (when referring to the eisosomes) or "near-native" (when referring to the MCC microdomain) which we think is justified by the following:

1. The detergent we use is below CMC, preventing membrane solubilization. We have also presented above in Rebuttal Figure 1 in response to Reviewer #1 that the inclusion of detergent is unnecessary to acquire the tubules; instead, it appears to be necessary to reduce other contaminants from being retained in our preps.
2. The tubule diameters observed in our data (~31-37nm) are comparable with the diameters of the Pil1/Lsp1 lattice invaginations measured for *in situ* samples (~32-58 nm) [Bharat, *et al.*, 2018 (PMID: 29681471)]. Furthermore, tubulation of the eisosomes at the plasma membrane has also been observed *in vivo* under certain circumstances [Haase, *et al.*, 2023 (PMID: 37902009)], suggesting that our samples could very well represent a physiological state of the MCC/eisosome.
3. Most importantly, the structural observations that we were able to make within the membranes of these native-source samples guided our choices for the composition of the lipid mixtures used in the

reconstitution studies. Inclusion of those lipids allowed us to solve structures at sufficient resolution to assign lipid identities to the structural features we observed in the “near-native” samples, as well as confirm those identities orthogonally with both *in vitro* lipid diffusion and sorting assays and *in silico* molecular dynamics simulations.

2. It is unclear to this reviewer and seemingly to the authors themselves what is meant by native, near-native, and native-like. The former has a commonly used definition that is clearly inappropriate here. The latter have no specific meaning, but it remains unclear what the authors would intend, considering they provide no evidence to support which features of the isolated structures are near-native and in which ways they may be native-like. This also extends to the claim of ‘intact’ plasma membrane, which is again obviously incorrect under the common definition of ‘intact’.

As outlined above, we have strong reason to believe that our structures reflect the endogenous protein-membrane interactions and, given the repetitive nature of the protein-membrane lattice, also reflect the endogenous lipid organization of the MCC. We believe the use of intact is appropriate here because we have preserved these protein-lipid interactions, but we have nonetheless softened the language to indicate our assumptions: “*We serendipitously isolated MCC/eisosomes from S. cerevisiae using a gentle purification procedure, preserving a lattice of untagged Pil1/Lsp1 structural proteins bound to a presumably intact plasma membrane bilayer in a near-native state, observable as a two-layer density within the protein tubule.*” (line 88-91).

3. There are also myriad issues with claimed “stretch sensitivity”, which appears closer to conjecture than data. (1) I do not know what “3D variability analysis” is, there are no references, and minimal explanation in the methods section. It is impossible to critically evaluate this approach based on the information in the manuscript.

We apologize for the lack of clarity about 3D variability analysis, which indeed forms the crux of the data supporting our interpretation of stretch-sensitivity. To improve on this, we have added the following text and reference in a new Supplementary methods section to describe 3DVA analysis. “*To identify dimensions of continuous heterogeneity in the symmetry expanded/density subtracted dataset, we employed 3D variability analysis which enables both the resolution and visualization of flexible movements within cryoEM datasets [Punjani&Fleet, 2021, J Struct Biol].*”

We hope that this explanation and reference (PMID: 33582281) clarifies that 3D variability analysis provides real 3D visual information about the direction and degree of continuous flexibility observed in cryoEM datasets.

(2) it is impossible to understand what “spring-like stretching” (line 373) or “changes in shape and size of bound lipid density” (line 375) the reader is meant to observe from SuppVideo3. The legend provides minimal explanation.

To aid in the interpretation of what is now Supplementary Video 4 in the updated figure numbering system, we have added labels and highlighting to the N-terminal contact sites (to visualize the spring-like stretching at these sites), the amphipathic helix (to visualize its movement in response to stretching), and the lipid density (to visualize the change in shape during stretching) in the video. We have also expanded the text of the video legend as follows:

“Supplementary Video 4. 3D variability visualization of native-source eisosome lattice. Volume series visualizing the “lattice-stretching” component from 3DVA using symmetry expanded/density subtracted particles from native-source eisosomes. Stretching at Nt lattice contact sites (cyan highlight) expands the Pil1/Lsp1 lattice and shifts the position of the AH (magenta highlight, connected to Nt contact sites at dotted

line), altering the structure of the putative lipid-binding pocket. This movement is correlated with changes in shape and size of the bound lipid density (green highlight) in the volume series."

(3) all isolated tubules are at equilibrium, there is no possibility for mechanical stretch, thus assigning "stretched" and "compressed" to different tubules presupposes something that doesn't exist in this experiment. It is inappropriate to use phrases like "lattice stretching" without any evidence thereof.

The fact that the isolated tubules are at equilibrium is an important point that merits clarification in the text. We are not suggesting that the tubules themselves are in the process of "stretching" and "compressing" but rather that we observe that the protein lattice itself is flexible.

Our 3D variability analysis revealed that the Nt contact sites are the protein domains which most dynamically contribute to that lattice flexibility. By solving structures derived from 10 intermediate subsets of particles extracted along the 3D variability component and refining models into these maps, we are able to make comparisons between subsets of particles exhibiting different degrees of stretch. To better illustrate this, we have added panel Ext Data Fig 10B showing the distances between different protein regions in neighboring dimers in the most stretched vs most compressed maps (with ~2.2Å of additional distance in the region near the Nt contact sites). To clarify that we are talking about protein lattice stretching rather than stretching of the tubules, we have changed our terminology throughout the text to "*stretched protein lattice*" and "*compact protein lattice*" class.

Our initial observation that our native-source samples contain a wide variety of tubule diameters already supported the notion that the Pil1 lattice exhibits sufficient flexibility to stretch to accommodate these variable tubule diameters, which was perhaps expected due to the reported function of the eisosomes in flattening in response to membrane tension (e.g. hyposhock, pipette suction, etc). However, we have not simply assigned "stretched" or "compressed" to different tubule diameters, rather we have backtraced for particles in each class (with classes ranging from most compact protein lattice to most stretched protein lattice) the diameter of the tubule of origin (Ext Data Fig 10C-D). The fact that particles from more stretched protein lattice classes are over-represented in larger diameter tubules and particles from more compact protein lattices classes are over-represented in smaller diameter tubules is empirical. Our explanation is that the protein lattice must stretch to accommodate the larger diameter tubules.

To clarify this point, we have made the following modification to the manuscript text: "*Particles from every tubule diameter are distributed across all 10 classes; however, particles from small diameter tubules are over-represented in the more compact classes, while those from large diameter tubules are over-represented in more stretched classes (Extended Data Fig 10C-D). This suggests that the flexibility we observe arises from the protein lattice stretching to accommodate for larger diameters of tubules.*" (line 338-342)

What we think is the most important and novel observation, supporting the functional relevance of the protein lattice stretching that we observe, is the Nt sites (which are experiencing this lattice stretching) are directly connected to the amphipathic helix (which is embedded in the membrane and forms a part of the lipid binding pocket) which can thus transmit that movement to the lipids stabilized within the membrane. And in fact, both the lipid density and the sterol voids transform from well-resolved in the most compressed state to low resolution and blurred in the most stretched state, demonstrating how the protein lattice stretching impacts the membrane lipid dynamics in molecular detail.

(4) the morphing between various classes is essentially a guess, it is unclear how much data or manual input goes into these types of analyses and they appear ripe for over-fitting and over-interpretation

We hope this aspect is clarified with our description and reference for 3D variability analysis in the supplementary methods that describes this method for visualizing continuous heterogeneity. All particles from the native source map are used and the only manual input is the number components (dimensions) of variability requested (all the particles are evaluated for each component). It is quite different from the use of morph map functions in structure visualization software. We also present these references as recent examples of how 3D variability analysis has been used to visualize continuous heterogeneity in cryoEM samples. [Paknejad, N, *et al.*, 2023 (PMID: 37898605); Milicevic, N, *et al.*, 2023 (PMID: 38030725); Rogala, KB, *et al.*, 2019 (PMID: 31601708)]

4. It seems odd to base the entire manuscript on a contaminant of an irrelevant pulldown with no detectable signal for the supposed proteins being resolved. Without any biochemical evidence of purity or even protein identity in the 'native' samples, the potential for artifacts (eg from contaminants) or over-fitting seems high. For example, the various tubules used to suggest 'stretch' may instead have different isoforms, interacting proteins, post-translational modifications, etc etc etc.

We have edited the manuscript text to clearly present the serendipitous nature of our finding. We also have included new supplementary data containing mass spectrometry results from the native-source eisosome purification demonstrating the presence of Pil1 and Lsp1 in the sample (Supp Data 1). At the resolution we have achieved (3.2Å) we can make an unambiguous identity assignment of the protein as either Pil1 or its paralog Lsp1 using side-chain geometry; the maps are resolved without any *a priori* information about protein identity and there is no risk, for example, that our structures represent a different contaminant protein. We have added the following text to the new Supplementary methods to indicate this: *"Ultimately, we were able to clearly assign protein identity with our structural data due to the resolution we achieved and confirm the presence of Pil1 and Lsp1 in these preps using mass spectrometry (Fig Ext Data 1D, Supplementary Data 1)."*

Rebuttal Fig 3. Result component plot. Component 1 (continuous heterogeneity) vs Component 2 ("stretching" component). The stretching component exhibits continuous heterogeneity, without obvious clusters.

It is true that for any protein isolated from its native source (as opposed to recombinantly expressed) there are potentially unknown factors like unresolved additional interacting proteins, post-translational modifications, variations in the ratio of homologs, etc. In spite of this, we are confident that the lattice stretching observed in our 3D variability analysis represents continuous, rather than discrete, heterogeneity. When we plot the reaction coordinates of this "stretching" component, we see no separated clusters of datapoints that would indicate a lack of continuity in the conformational space (e.g. a different conformation induced by a PTM or due to major differences in sidechain chemistry of the homologs) (see Rebuttal Fig 3) [Punjani&Fleet, 2021, J Struct Biol (PMID: 33582281)].

Finally, as the eisosomes have a functional role in sensing membrane perturbations, and the protein lattice stretching we observe correlates with changes in the stable localization of membrane lipids coordinated by the protein lattice, we feel that our conclusion that these observations are related to stretch-sensitivity is logical.

5. The evidence of sterol-dependence of the 'voids' is weak.

To bolster this claim, we were able to reconstitute Pil1 tubules in the presence of brominated sterols [Moss, *et al.*, 2023, NSMB (PMID: 36624348)] and solve structures of these tubules to ~3.9Å resolution. Using ergosterol as the starting product, we were able to brominate across the double bond at the 7,8- position, adding extra density to the molecule at approximately the midpoint of the sterol ring structure.

When we compare our reconstituted structures of +PIP2/+bromo-sterol lipid mixture with the +PIP2/-sterol and the +PIP2/+sterol reconstituted structures using the one-pixel parallel slice visualization strategy, we can see that the bromo-sterol structures resemble the +PIP2/+sterol structures in that their amphipathic helices are well-resolved, and the voids can be observed at the plane of the amphipathic helix, as well as starting at the depth of $\sim 8\text{\AA}$ from the bilayer midplane, corresponding approximately to the predicted location of C17 of cholesterol within the bilayer in MD simulations [Smodyrev&Berkowitz, 1999, Biophys J (PMID: 10512828)]. However, in the bromo-sterol structures, in the slices ranging from $\sim 11\text{-}12\text{\AA}$ distance from the bilayer midplane, which would correspond quite well with C7(8) where the bromination is located on the bromo-sterol molecule, the voids are “filled”. We think that this result strongly supports the idea that the voids represent individual (or perhaps a few) sterol molecules, while nicely controlling for the incorporation of our altered sterol molecules (amphipathic helix remains well resolved, and void pattern is retained, except at the depth of the bromination). These data have been included in Ext Data Fig 5E-F and the following text has been added to the manuscript to describe this result:

“To directly test whether the voids represent sterol molecules, we reconstituted and solved structures of Pil1 tubules with lipid mixtures containing sterols that were brominated at the 7,8- position of the steroid ring (bromosterol) to add density to these molecules that can be observed in cryoEM [Moss, 2023, NSMB] (Extended Data Figs 3E and 6E and Supplementary Data 2&3). In these +PI(4,5)P₂/+bromosterol reconstituted structures, the AHs are well-resolved, similar to the +PI(4,5)P₂/+sterol structures, suggesting that the bromosterols behave similarly to cholesterol in these structures (Extended Data Fig 5E-F). Furthermore, the voids can be observed at both the plane of the AH and starting at the depth of $\sim 8\text{\AA}$ from the bilayer midplane, corresponding approximately to the predicted location of C17 of cholesterol within the bilayer in MD simulations⁴⁸. However, in the +PI(4,5)P₂/+bromosterol structures, in the slices ranging from $\sim 11\text{-}12.5\text{\AA}$ distance from the bilayer midplane, which would correspond well with the bromination at C7(8) on the bromo-sterol molecule, the voids are interrupted by density, strongly suggesting the brominated sterols are localized to the voids, and that the void pattern represents stabilized sterol molecules (Ext Data Fig 5E-F).” (line 189-202)

Fig2F-H appears to show much stronger differences between the ‘native’ and the two reconstituted samples - why are reconstituted structures more poorly resolved than the isolated?

Indeed, there are differences in appearance between membrane from the native-source samples and the reconstituted ones. Our guess is that this could be related to any of the many aspects of the native membrane composition that are not recapitulated in our reconstituted samples (variations in chain length, saturation, effects of cholesterol vs ergosterol, incorporation of rare species e.g. lysolipids,). As we do not want to imply that our analysis of the membrane composition and organization is exhaustive, we have added the following text to the methods section *“While these mixtures do not capture the full complexity of the native membrane, using these reconstitutions, we were able to make several salient observations.” (line 461-463).*

The presence of ‘voids’ is not quantified in any way, preventing meaningful comparison.

To improve the visualization of the voids relative to the amphipathic helix, we have added Ext Data Fig 5A-C,G-H. These images are produced by raising the threshold of the map until the membrane appears solid, applying local resolution values to the surface, and using zone maps to provide a “cut out” window into the inside of the membrane density adjacent to the amphipathic helix. This results in an inverted view of the topological features within the membrane density, allowing us to see pockets of higher resolution appearing near the amphipathic helices in the native-source, +PIP2/+sterol, and compact protein lattice class; these are absent in the +PIP2/-sterol and stretched protein class. We have also included several Supplementary videos (1,5,&6) to illustrate the void positions in 3D. Furthermore, we have added panel F in Ext Data Fig 10 to illustrate the gradual change in void intensity in the intermediate structures derived from the stretching component of the 3DVA. These additions are further described in the response to Reviewer #3.

It is a trivial observation that the hydrophobic face of a membrane-embedded AH is interacting more frequently with sterol than are the charged residues on the other side. This is not evidence of sterol recruitment, only of those residues being in a lipid environment. A more interesting analysis would be whether sterol is enriched relative to PC.

A comparison of the dwell times of DOPC and sterol headgroups along the residues of the amphipathic helix (Ext Data Fig 7B) illustrates that the sterols do have an altered pattern of localization in general to that of the DOPC headgroup and an increased preference for bulky side chains.

6. Fig 2K purports to show that 60-70% of cholesterol is immobilized on the minute timescale in the +Pil1+PIP2 condition. This is physically implausible: e.g. the MD simulations would likely reveal that all lipids are completely mobile on the usec timescale. Same issue extends to the FRAP experiments in Fig 3.

We remind the reviewer that none of the membrane diffusion assays that we use are in free standing membranes: the tubes, because of their geometry, dramatically reduce the degrees of freedom for diffusion. Thus, the presence of immobile sterol and lipid fractions is not surprising given that FRAP experiments were performed using lipid nanotubes that have a diameter of nanometer scale [Dar, S *et al.* 2018 (PMID:28125102)]. Previous study showed that the diffusion of lipids in the nanotube compared to the relatively flat GUV-derived membrane is significantly reduced due to small surface area in the nanotubes [Raote, I *et al.* 2020 (PMID: 32452385)].

There are also issues with interpretations of the FRAP data: immobile fraction and diffusion rate are two independent parameters in FRAP experiments and it is unclear which the authors refer to when they claim that "the membrane is less diffusive".

We apologize for the lack of clarity when describing the FRAP data. Indeed, we do see that some fraction of the lipids remains immobile under the protein lattice, while there are also clear differences in diffusion rates of mobile lipid fractions. When we talk about "less diffusive" membrane we refer to the increase in the immobile fraction of TopFluor-PC and -PE in the presence of Pil1. We have now edited the description of FRAP data in the main text as follows:

" To investigate microdomain formation by the Pil1 lattice, we used FRAP assays with TF-PC and -PE and found that a significant portion of each of these lipids is immobilized in the presence of Pil1 (Extended Data Fig 8E-F). In the presence of both Pil1 and PI(4,5)P₂, while the immobile fraction remains similar, the dynamics of the mobile lipid fraction are decreased for TF-PC and -PE lipids (Extended Data Fig 8E-F and Supplementary Table 5). However, the immobile fraction and the dynamics of the mobile lipid fraction for TF-PC and -PE are broadly similar in the presence or absence of cholesterol (Extended Data Fig 8G-H)..." (lines 261-267)

7. Fig 3B is confusingly presented. First, its not easy to understand what this experiment is.

We apologize for the confusion. The lipid sorting coefficient is a comparison between the intensity of the TopFluor lipid of interest and the control lipid DOPE-Atto647N underneath the Pil1 lattice assembled on nanotubes vs the relative intensity of those lipids on the bare nanotube.

$$\text{Sorting coefficient} = \frac{(F_{\text{Tested lipid}}/F_{\text{Atto 647N DOPE}}) \text{ under Pil1}}{(F_{\text{Tested lipid}}/F_{\text{Atto 647N DOPE}}) \text{ bare lipid nanotube}}$$

Why is there no lipid under the protein in ExtFig9A?

The constriction of the nanotube by Pil1 reduces the intensity of the control lipid at sites of Pil1 lattice and gives the appearance of a loss of signal in this region. Figure legends for the new Ext Data Fig 8A-B have been updated to clarify as follows:

“Figure Extended Data 8. Lipid sorting co-efficients and +PI(4,5)P₂/-sterol FRAP assays. A. Example lipid nanotube with DOPE-Atto647N and Pil1-mCherry bound. Constriction of the nanotube reduces the intensity of the labeled lipid fluorescence where Pil1-mCherry is bound. B. Example lipid nanotube with Pil1-mCherry, fluorescent TopFluor-PI(4,5)P₂, and fluorescent DOPE-Atto647N. Constriction of the nanotube reduces the intensity of the labeled lipid fluorescence where Pil1-mCherry is bound. Increased sorting of TopFluor-PI(4,5)P₂ relative to DOPE-Atto647N is observed at sites of Pil1 nanotube constriction.”

Second, the x-axis appears to have all the same labels, so compared conditions aren't clear.

Thank you for pointing this out. The x-axis labels have been corrected. They were accidentally cut off during figure conversion.

What do the n refer to, number of nanotubes? If so, those should not be used as independent samples.

N means total independent experiments whereas n is number of tested nanotubes (see updated Figure 3B legend). We apologize for the misunderstanding, as we were not intentionally presenting n as independent experiments. However, it is standard and accepted in membrane reconstituted *in vitro* studies to pool data from independent experiments to get the number of model membranes (giant liposomes, tubes, pulled tubes from GUVs) used [Sorre, *et al*, 2009, PNAS (PMID: 19304798); Roux, *et al*, 2010, PNAS (PMID: 20160074); Heinrich, *et al*, 2010, PNAS (PMID: 20368457)].

In addition, in our case, in which we utilize *in vitro* reconstitutions in which the experimental conditions and procedures are highly controlled, N is not an essential statistical variable to probe intrinsic variability; rather n is more relevant. In this specific case, where lipid composition, protein concentration and other experimental parameters are highly controlled, the main source of variability is the distribution of tube radii, which is comparable from one experiment to another. Therefore, the total number of tubes (n) - each one being considered as an independent event - rather than the number of replicates (N) is the statistically relevant variable.

8. The logic of the myriocin result is unclear. How does it make sense that mutants with a defect in lipid binding are more myriocin resistant?

To clarify the logic of this result, we have modified the following text: *“This is in line with the previously described role of eisosomes in sphingolipid biosynthesis signaling [Frohlich, 2009, JCB; Aguilar, 2010, NSMB] and, given the mislocalization of the proposed sphingolipid biosensor, Nce102 [Zahumensky, 2022, Clinical Microbiol], we can speculate that sphingolipid biosynthesis may be mildly upregulated in these myriocin-resistant mutants.”* (line 300-303)

More generally, the physiological experiments are underwhelming, with few (if any) meaningful defects reported despite mutating quite a few of the residues supposedly important for lipid interactions.

We respectfully disagree that these mutants produce few meaningful defects; both the eisosome morphology and the physiologically relevant colocalization of tetraspannin protein Nce102 with Pil1 [Frohlich, 2009, JCB (PMID: 19564405); Zahumensky, 2022, Clinical Microbiol (PMID: 35758748)] are quite profoundly affected in several of the mutants presented.

We do concede that for the growth assays some of the phenotypes are mild. We have included a Pil1/Lsp1 deletion as a positive control in these assays (Ext Data Fig 9F) to illustrate that resistance phenotypes are similarly mild even with complete loss of the protein. In fact, for cold growth, the full deletion results in a sensitivity phenotype, suggesting that the cold resistance observed in the 130/133 mutant is a gain-of-function phenotype.

Along that line, none of the mutants described in Fig 4 should be called “lipid binding impaired” because there is no evidence provided about their lipid binding. The authors obviously know this, correctly referring to them as “predicted to affect binding” on line 323.

To confirm lipid binding impairment in our mutant strains, we have expressed and purified several of them to test in membrane nanotube-based assays. Using lipid sorting assays, a trend towards a slight decrease in PIP2 sorting is observed in PIP2-binding impaired mutant (Pil1-K130A/R133A) relative to WT, while PS sorting is significantly impaired in both the PS-binding impaired mutant (Pil1-K66A/R70A) and the PIP2-binding impaired mutant (Pil1-K130A/R133A). Using FRAP assays, we observed that the mobile fraction of sterols is significantly increased in the sterol-binding impaired mutant (Pil1-F33A/Y40A/F42A/F50A), confirming the reduction of sterol-binding by this mutant. This data is shown in Ext Data Fig 9A-B. For untested mutants, we have changed our nomenclature to “lipid binding pocket” mutants as the mutated residues are localized to the lipid pocket.

Other issues:

a. Sentence starting on line 53 is oddly phrased: the function described does not sound mysterious, but rather quite clear.

This sentence has been changed as follows: *“MCC/eisosomes...have been implicated in sensing and responding to plasma membrane stress: various stimuli including hypo-osmotic shock, heat shock, and mechanical pressure cause eisosomes to flatten and release sequestered proteins to affect signaling or transport functions.* (line 47-50)

b. Some citations are a bit sloppy for this level of journal: eg Shimshick & McConnell say nothing about the plasma membrane

We thank the reviewer for catching this error. In the end, this sentence has been removed in response to comment (c) below, but we have carefully checked all references in advance of resubmission.

c. The novelty and impact of the manuscript are framed with respect to membrane microdomains. However, eisosomes are in most ways entirely unlike the controversial PM domains referenced: they are stable, easily visualized, scaffolded by specific proteins, etc. Rather, they are effectively peripheral membrane protein assemblies, like ESCRTs, caveolae, clathrin cages. Structures of many of these have been reported. Thus, framing the novelty around lipid microdomains is misleading.

We thank the reviewer for this insight. While we recognize that the MCC/eisosomes are quite different from the more transient and amorphous detergent-insoluble cholesterol/sphingolipid microdomains that have been identified, MCC/eisosomes [Zahumensky&Malinsky, 2019, Biomolecules (PMID: 31349700); Lanze, *et al.*, 2020, Microbiol and Mol Biol Rev (PMID: 32938742)], as well as caveolae and t-tubules [Reeves, *et al.*, 2012, Adv Exp Med Biol (PMID: 22411310); Parton, 2018, Ann Rev Cell and Dev Biol (PMID: 22411310); Russell, *et al.*, 2017, Cardiovasc Diabetol (PMID: 29202762)], are referred to throughout the literature as membrane microdomains and prime examples of membrane compartmentalization. (see response to Reviewer 1 for further explanation). As such, we have re-written the first paragraph, preserving the membrane compartmentalization context, but de-emphasizing references to the more controversial detergent-resistant domains.

“Membrane compartmentalization enables the spatiotemporal control of a variety of signaling events at the plasma membrane. Although the biological evidence for membrane compartmentalization is overwhelming

[Honigmann&Pralle, 2016, J Mol Biol; Lu&Fairn, 2018, Crit Rev Biochem Mol Biol; Shi, et al., 2018, Cell], the determinants and the physical structure of the lipid organization within the membrane remain controversial. This is because almost all tools used to study membrane lipids also risk perturbing their behavior within the membrane context [Levental, et al., 2020, Trends in Cell Biol; Sezgin, et al., 2017, Nat Rev Mol Cell Biol].” (lines 35-39).

While other peripheral membrane protein lattice structures have been solved, the novelty of our study lies in our direct observations of the high degree of organization of the membrane lipids that lie beneath the Pil1 lattice, with regularly-spaced bound lipid headgroups and a sterol-mediated regular pattern of voids; this reveals that it is not just the composition, but the protein-regulated organization of the lipids in the MCC/eisosomes that facilitates their specialized function. There are a few other examples in the literature that we are aware of that describe this type of behavior of lipids scaffolded by protein arrays at molecular detail [Moss, et al, 2023, NSMB (PMID: 36624348); Unwin, 2022, PNAS (PMID: 35969788)], but our study goes further by 1) capturing the endogenous yeast membrane, and 2) by demonstrating that the molecular features we observe are replicable by the inclusion/exclusion of specific lipid species in our reconstitutions.

d. And why were the reconstitutions and simulations done with cholesterol rather than ergosterol?

We chose to use cholesterol for the reconstituted structures because its inclusion produced more robust tubulation that was necessary for solving cryoEM structures than ergosterol. We were also surprised that ergosterol gave us slightly worse tubulation since it is the main sterol species in yeast (they do not produce cholesterol). We have noted these ergosterol-related observations in the Supplementary methods. For additional info, see response to Reviewer 3.

Since cholesterol was the lipid used in the structural studies, we also used it for the lipid sorting assays, FRAP assays, and MD simulations. However, we did test PIP2 dynamics under ergosterol conditions using the FRAP assay and saw no effect relative to cholesterol. This control has been added to Ext Data Fig 8I and Supplementary Table 5.

Reviewer Reports on the First Revision:

Referees' comments:

Referee #1 (Remarks to the Author):

This is a very serious revision. The experiments with brominated sterol further reinforce one main conclusion of this study; the capacity of the BAR domain lattice to organize the distribution of specific lipids in the cytosolic leaflet, including sterols. In addition, the introduction now goes directly to the main point and stays closer to what is actually seen instead of making a link with the hypothesis of lipid-driven microdomains.

Referee #2 (Remarks to the Author):

All the comments I raised in the previous round of review have been satisfactorily addressed by the authors.

Referee #3 (Remarks to the Author):

The authors have been very thorough in responding to the issues raised by the reviewers and I congratulate them on an exciting and well-executed study. I recommend publication.

Minor: please ensure that the publication version of the extended data includes higher-resolution figures than the revised submission, in particular for Figure Extended Data 5.

Referee #4 (Remarks to the Author):

In my opinion, the manuscript has been notably improved by the revisions. I appreciate the thoughtful approach to the reviewer comments. Despite my differences of opinion about some aspects of the work, I will defer to the enthusiasm of experts in structural biology and the eisosome.

That said, there remain two major outstanding issues, where core claims are supported by poor evidence:

1. "Fig 2K purports to show that 60-70% of cholesterol is immobilized on the minute"... the authors answer here is insufficient, as neither previous literature nor their own simulation is consistent with anything like 60% of sterols being completely immobilized over minutes. The authors claim is that the majority of cholesterol molecules remain bound to a protein for minutes, which would be many orders of magnitude longer than any previously reported sterol-protein interaction. I maintain that the claim is physically implausible and has not been observed in other systems (the papers referenced also do not). The geometry of the tube may slow diffusion, but does not completely immobilize the majority of sterol. The plausibility of the claim should be directly verifiable from the

simulations: what fraction of cholesterol is fully immobile, even at microsecond timescales? how would this extend to minutes?

Rather, what is more likely is that the plateau represents bleaching of a large fraction of the fluorescent lipid molecules. This is evident from the Supp videos, which show the entire tube dimming dramatically after photobleaching a small spot. Minimally, these data must be replotted as fluorescence within the photobleached region RELATIVE to fluorescence outside the bleached region. Typically in FRAP experiments, the relevant parameter for diffusivity is the RATE at which the lipids recover. I would strongly recommend the authors to consider revising this aspect of the manuscript and removing unsupported claims.

2. "N means total independent experiments whereas n is number of tested nanotubes... it is standard and accepted in membrane reconstituted in vitro studies to pool data from independent experiments": there is such thing as field-specific standards in statistics. This approach is inappropriate. Statistical significance can only be calculated from independent replicates. Pooling individual samples from multiple replicates artificially inflates the N and gives absurdly low p-values. If there were 2 independent experiments, then N=2 and they should be shown separately. Minimally, one of these replicates could be shown as a "representative" experiment, with a statistical comparison WITHIN that experiment. It is for the editors to decide whether one repeat of such a "representative experiment" is sufficient for Nature...

Author Rebuttals to First Revision:

Referee #4 (Remarks to the Author):

In my opinion, the manuscript has been notably improved by the revisions. I appreciate the thoughtful approach to the reviewer comments. Despite my differences of opinion about some aspects of the work, I will defer to the enthusiasm of experts in structural biology and the eisosome.

That said, there remain two major outstanding issues, where core claims are supported by poor evidence:

1. "Fig 2K purports to show that 60-70% of cholesterol is immobilized on the minute"... the authors answer here is insufficient, as neither previous literature nor their own simulation is consistent with anything like 60% of sterols being completely immobilized over minutes. The authors claim is that the majority of cholesterol molecules remain bound to a protein for minutes, which would be many orders of magnitude longer than any previously reported sterol-protein interaction. I maintain that the claim is physically implausible and has not been observed in other systems (the papers referenced also do not). The geometry of the tube may slow diffusion, but does not completely immobilize the majority of sterol. The plausibility of the claim should be directly verifiable from the simulations: what fraction of cholesterol is fully immobile, even at microsecond timescales? how would this extend to minutes?

Rather, what is more likely is that the plateau represents bleaching of a large fraction of the fluorescent lipid molecules. This is evident from the Supp videos, which show the entire tube dimming dramatically after photobleaching a small spot. Minimally, these data must be replotted as fluorescence within the photobleached region RELATIVE to fluorescence outside the bleached region. Typically in FRAP experiments, the relevant parameter for diffusivity is the RATE at which the lipids recover. I would strongly recommend the authors to consider revising this aspect of the manuscript and removing unsupported claims.

The fraction of immobile lipids and their recovery rate notoriously depends on the geometry of the membrane as well as the size of the area ablated [PMID: 18214381]. Regarding the perceived long recovery times, we believe that the reviewer may be inappropriately comparing FRAP data from large, free-standing flat membrane versus our conditions in which the membrane is in a tight tube. In membrane nanotubes where a large section of the tube is photobleached – similar to our conditions – recovery times of tens of seconds for lipids are common (see for example figure 6 of (PMID: 19348750)).

Also, in FRAP experiments, recovery rate and immobile fraction depend on the time window used for observation of recovery. Therefore, the recovery rates and the immobile fraction cannot be directly translated into diffusion coefficients of molecules and fraction of bound molecules, like the reviewer seems to imply. Our use of the word "immobilized" in the manuscript text refers to the immobile fraction we measured with FRAP, but not to an absolute number of molecules bound to the coat indefinitely. In complex diffusion environments, changes in lipid diffusion can affect both the recovery rates and the immobile fraction. The fact that we do not see changes of recovery rates, but rather of the immobile fraction when Pil1 is added thus shows that lipid diffusion is reduced, most likely on longer time scales. But we cannot infer the number/fraction of bound molecules and for how long they stay bound to the Pil1 coat.

We note that in our FRAP data analyses we corrected the fluorescence recovery by measuring fluorescence intensities 1) from the photobleached region, 2) from the lipid nanotube region that was not photobleached, and 3) from background. We subtracted the background and subsequently corrected the photobleaching by calculating a relative fluorescence between the photobleached region and the neighboring nanotube region, which was not photobleached, as described in the Supplementary Methods. Thus, any unwanted photobleaching during post-photobleach acquisition does not affect measured recovery rates or mobile fraction percentages.

Additionally, it would be inappropriate to quantitatively compare lipid diffusion in the CG-MD simulations with the experimental observations: diffusion in CG simulations is much faster than what is observed in the experiments because of the intrinsically smoother CG free energy landscape, due to the loss of degrees of freedom. Therefore, there is no intrinsic timescale in simulations, which makes a quantitative comparison of rates and times with *in vitro* experiments rather arbitrary. On the other hand, a qualitative interpretation of our simulations indicates that interactions between Pil1 and lipids slow down lipid

diffusion (Rebuttal Figure 1 and Extended Data Fig 7G show 1D lipid diffusion coefficients along the axis of the tubule for consistency with the FRAP diffusion experiments; however, 2D diffusion in CG simulations of the tubules follows the same trend). This is particularly true for specific lipid classes (e.g. PIP2) and it is consistent with what was observed by others in similar systems with tubular membranes coated by proteins (PMID: 36624348).

Rebuttal Figure 1. Lipid diffusion coefficients from CG simulations. The values reported are the 1D axial diffusion of lipids in the outer leaflet of the tubes, with (green) and without (cyan) the Pil1 protein coat.

Regardless, we did not mean to state that cholesterol is immobilized in the sense implied by the reviewer, rather, in our FRAP data we observe an increase in the immobile fraction in the presence of Pil1 protein. The key message we wish to convey is that the tight interaction between Pil1 and cholesterol limits its diffusion, which supports our observations by CryoEM that sterols are stabilized beneath the AH of Pil1. This conclusion is additionally supported by our CG-MD simulations and reconstitution experiments with brominated sterols.

2. “N means total independent experiments whereas n is number of tested nanotubes... it is standard and accepted in membrane reconstituted in vitro studies to pool data from independent experiments”: there is such thing as field-specific standards in statistics. This approach is inappropriate. Statistical significance can only be calculated from independent replicates. Pooling individual samples from multiple replicates artificially inflates the N and gives absurdly low p-values. If there were 2 independent experiments, then N=2 and they should be shown separately. Minimally, one of these replicates could be shown as a “representative” experiment, with a statistical comparison WITHIN that experiment. It is for the editors to decide whether one repeat of such a “representative experiment” is sufficient for Nature...

We appreciate the feedback provided by the reviewer and apologize for any misunderstanding that may have arisen. In response to the concerns raised, we sought guidance from José Manuel Nunes, a statistics expert and lecturer in the department of Genetics and Evolution at the University of Geneva. After careful consideration, we have concluded that our analysis methodology, particularly in terms of statistical analysis, is sound. Allow us to provide further clarification and validation of our approach.

Firstly, it is crucial to clarify that each individual data point presented in the boxplots represents measurements taken from independent objects. Therefore, the data collected for each measurement correspond to independent membrane nanotubes generated on the respective day of experimentation. Additionally, these measurements are obtained from distinct constricted regions under independent Pil1 scaffolds. Given this approach, we assert that our sample means are derived from independent measured samples.

Secondly, we respectfully disagree with the assertion that experiments conducted on different days cannot be pooled together. Our consulted expert has affirmed that pooling data across different days is appropriate as long as there is no significant batch difference between the experimental days. Put simply, as long as the measured datasets from each day do not exhibit significant differences within a given experimental condition, pooling the data is justified. To provide further evidence of the validity of our approach, we conducted non-parametric Wilcoxon-Mann-Whitney tests on datasets for each

condition separately between both days (see Rebuttal Fig. 2 and Rebuttal Fig. 3 in the rebuttal letter). Note that these tests are performed due to normality rejection by one of the conditions; according to Shapiro-Wilk and Anderson-Darling normality tests, the TF-PIP2 K130A/R133A condition in the first day rejects normality. The results (exact p values shown in Rebuttal tables 1 and 2) demonstrate that there is no significant difference between experimental days in any of the conditions tested. Therefore, pooling the data points together is justified.

Finally, to enhance clarity regarding the origin of each data point, as suggested by the reviewer, we have modified panels Fig. 3C and Extended Data Fig. 9A with a new superplot in the updated version of the manuscript (Rebuttal Fig. 4 and Rebuttal Fig. 5 in the rebuttal letter). In this version, data points from day 1 and day 2 of the experiment are differentiated by shape and color (black rhombuses for day 1 and gray circles for day 2).

In addition to the reviewer's request, we have updated the statistical analysis conducted on the sorting coefficient measurements as presented in the manuscript, following guidance from our consulted expert. In the latest version, we have tested whether datasets follow a normal distribution using three different tests: 1) Shapiro-Wilk, 2) Kolmogorov-Smirnov, and 3) Anderson-Darling. Across all tested conditions, the datasets were found to adhere to a normal distribution at a confidence level of 1% (0.01). Consequently, we have now performed a two-sample t-test in the new statistical analysis, which is included in the latest version of the manuscript and in this rebuttal letter for reference (Rebuttal Fig. 4 and Rebuttal Fig. 5).

With these clarifications and updates, we trust that any potential misunderstanding regarding the clarity of our statistical analysis has been addressed.

Rebuttal Figure 2. Boxplots with individual datapoints showing sorting coefficient values obtained each day of experimentation for each tested condition shown in Figure 3C of the manuscript.

	p values
TF-PI(4,5)P2/No sterol	0.52885
TF-PI(4,5)P2/With sterol	0.35268
TF-PS/No sterol	0.37602
TF-PS/ With sterol	0.63053
TF-PC/With sterol	0.15187
TF-PE/With sterol	0.16549

Rebuttal Table 1. P values obtained using non-parametric Wilcoxon-Mann-Whitney tests for the conditions shown in Fig. 1 of the rebuttal letter. In all the cases there is no significant difference between experiments performed on day 1 and 2.

Rebuttal Figure 3. Boxplots with individual datapoints showing sorting coefficient values obtained each day of experimentation for each tested condition shown in Figure Extended Data 9A of the manuscript.

		p values
TF-PI(4,5)P2/With sterol	WT	0.35268
TF-PI(4,5)P2/With sterol	K66A R70A	0.42345
TF-PI(4,5)P2/With sterol	K130A R133A	0.21644
TF-PS/ With sterol	WT	0.63053
TF-PS/ With sterol	K66A R70A	0.5254
TF-PS/ With sterol	K130A R133A	0.34736

Rebuttal Table 2. P values obtained using non-parametric Wilcoxon-Mann-Whitney tests for the conditions shown in Fig. 2 of the rebuttal letter. In all the cases there is no significant difference between experiments performed on day 1 and 2.

Rebuttal Figure 4. Boxplots with pooled datapoints from both experimentation days (new Fig. 3C in the newest manuscript). Statistical significance obtained applying a two-sample t-test upon corroborating the normal distribution of the data at 0.01 (using Shapiro-Wilk, Kolmogorov-Smirnov, and Anderson-Darling normality tests). Black rhombuses show datapoints obtained at day 1 and gray circles show datapoints obtained for day 2.

Rebuttal Figure 5. Boxplots with pooled datapoints from both experimentation days (new Extended Data Figure 9A in the newest manuscript). Statistical significance obtained applying a two-sample t-test upon corroborating the normal distribution of the data at 0.01 (using Shapiro-Wilk, Kolmogorov-Smirnov, and Anderson-Darling normality tests). Black rhombuses show datapoints obtained at day 1 and gray circles show datapoints obtained for day 2.

The final version was seen by the referee(s).